# The double EFT expansion in quantum gravity

José Calderón-Infante[1*], Alberto Castellano[2,3†] and Alvaro Herráez[4‡]

**1** CERN, Theoretical Physics Department, 1211 Meyrin, Switzerland
**2** Enrico Fermi Institute & Kadanoff Center for Theoretical Physics,
University of Chicago, Chicago, IL 60637, USA
**3** Kavli Institute for Cosmological Physics, University of Chicago, Chicago, IL 60637, USA
**4** Max-Planck-Institut für Physik, Boltzmannstrasse 8,
85748 Garching bei München, Germany

⋆ jose.calderon-infante@cern.ch , † acastellano@uchicago.edu , ‡ aherraez@mpp.mpg.de

## Abstract

In this work, we aim to characterize the structure of higher-derivative corrections within low-energy Effective Field Theories (EFTs) arising from a UV-complete theory of quantum gravity. To this end, we use string theory as a laboratory and argue that such EFTs should exhibit a *double EFT expansion* involving higher-curvature operators. The *field-theoretic* expansion is governed by the mass of the lightest (tower of) new degrees of freedom, as expected from standard field theory considerations. Conversely, the *quantum-gravitational* expansion is suppressed relative to the Einstein-Hilbert term by the quantum gravity cutoff, $\Lambda_{QG}$, above which no local gravitational EFT description remains valid. This structure becomes manifest in the so-called *asymptotic regime,* where a hierarchy between the Planck scale and $\Lambda_{QG}$ emerges, the latter identified herein as the species scale. Most notably, we demonstrate the features of the double EFT expansion through an amplitudes-based approach in (toroidal compactifications of) ten-dimensional Type IIA string theory, and via a detailed analysis of the supersymmetric black hole entropy in 4d $\mathcal{N} = 2$ supergravities derived from Type II Calabi–Yau compactifications. We provide further evidence for our proposal across various string theory setups, including Calabi–Yau compactifications of M/F-theory and Type II string theory. Finally, we explore the implications of this framework for the Wilson coefficients of the aforementioned higher-curvature operators, revealing potentially significant constraints in the asymptotic regime and highlighting a remarkable interplay with recent results from the S-matrix bootstrap program.

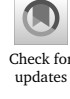

# 1 Introduction and summary

Einstein's theory of General Relativity has been extremely successful in describing gravitational interactions at low enough energies (equivalently large distance scales). Despite this success, the aforementioned theory —possibly supplemented with additional matter fields and interactions— is non-renormalizable at the quantum level. Thus, it is to be regarded as an effective field theory (EFT) valid up to a certain ultra-violet (UV) cutoff, denoted by $\Lambda_{\mathrm{UV}}$ [1, 2]. In fact, there is compelling evidence that the very framework of field theory has to break down when quantum gravitational effects become relevant. In other words, above some energy scale, $\Lambda_{\mathrm{QG}}$, the UV completion is no longer an EFT but rather some fully-fledged theory of Quantum Gravity (QG).

On the other hand, from the low-energy viewpoint, these features are naturally encoded in the form of higher-derivative and higher-dimensional corrections to the classical effective action. In particular, precisely whenever an infinite number of these operators become important and thus seem to correct the physical observables deduced from the truncated, two-derivative action, we are unavoidably lead to the conclusion that the EFT must break down. In addition to this, the higher-derivative expansion also modifies in a sensitive way the predictions made within the effective theory, and hence must be included so as to achieve sufficient precision at moderately low energies [3, 4]. For these reasons, it is crucial to understand the structure of higher-derivative corrections to gravitational EFTs arising in the low-energy limit of UV-complete theories of quantum gravity, such as string theory. This is the endeavor we aim to tackle in the present work.

In recent years, there has been a remarkable progress towards understanding certain universal features that theories of quantum gravity must possess, as well as their imprint at low energies. This is the ultimate goal of the Swampland program [5] (see [6–11] for reviews). An especially prominent role has been played by the so-called *asymptotic regimes* associated to infinite distance limits in the moduli space of the EFT, i.e., the space of vacua of the theory. The study of these asymptotic regions has led to the discovery of new several universal properties. For instance, an infinite tower of states purportedly becomes massless in Planck units along these limits, as captured by the Distance Conjecture [12]. This has been systematically studied and thoroughly tested across the string landscape [13–38], and moreover it has been argued to lead necessarily to the breaking of effective field theory as a framework at energies close to the species scale [39–42], which is parametrically below the Planck mass in this regime [14, 43, 44]. Recent works have also shown that this scale seems to appear in a universal way controlling certain higher-curvature corrections within the gravitational field theory expansion [45–53]. This paper crucially builds upon these previous efforts and extends their results with an eye to future applications, especially regarding the interplay with the S-matrix bootstrap approach.

In fact, one of our main goals is to take some steps towards bridging the gap between the Swampland and the S-matrix bootstrap approaches. The latter aims to constrain the behavior of (gravitational) amplitudes from the bottom-up by imposing first-principles such as symmetry, unitarity and causality. Hence, the connection between these two perspectives is very natural. Indeed, several interesting insights and results of relevance for crucial questions in the Swampland Program have already been produced from the S-matrix bootstrap program (see, e.g., [54–72]).

## 1.1 The structure of quantum gravitational effective actions

The main goal of this paper is to introduce the *double EFT expansion*, a proposal for the general structure of higher-derivative corrections in gravitational effective field theories. As shown in subsequent parts of this paper, this scheme is recovered in top-down string theory constructions and, furthermore, is compatible with current bottom-up S-matrix bootstrap bounds arising from imposing unitarity and causality. However, before delving into the details, let us start by giving a heuristic but physically meaningful motivation for our proposal.

Consider the low-energy expansion of a given theory of quantum gravity in $d > 3$ spacetime dimensions.[1] At the two-derivative level, this field theory includes the Einstein-Hilbert (EH) term

$$S_{\text{EFT},d} \supset \frac{M_{\text{Pl},d}^{d-2}}{2} \int \mathrm{d}^d x \sqrt{-g}\, \mathcal{R}, \tag{1}$$

---

[1]See, e.g., [73–87] for recent progress towards a definition and understanding of 3d quantum gravity in spacetimes with negative/positive cosmological constant.

and possibly a finite number of additional matter fields, as well as (gauge) interactions. Notice that, since Einstein gravity is non-renormalizable, the above effective description is expected to break down at energies close to or below the Planck scale, $M_{\text{Pl},d}$, which in fact controls the intensity of the gravitational interactions via Newton's constant, $G_N = M_{\text{Pl},d}^{2-d}/8\pi$. From the EFT perspective, this is signaled by an infinite number of higher-derivative and higher-curvature operators becoming relevant at some energy scale $\Lambda_{\text{UV}} \lesssim M_{\text{Pl},d}$. In this regard, the key idea behind the double EFT expansion is to distinguish between two qualitatively different mechanisms by which the $d$-dimensional gravitational effective field theory is expected to break down depending on the nature of the underlying UV completion.

The first type of breaking occurs whenever the theory can be completed at high energies into another *local* EFT. For instance, this happens when we encounter a finite number of new fields appearing at a given energy scale $M$. In this case, we can simply incorporate these massive degrees of freedom into our description by integrating them in, thus obtaining some new effective description involving the same number of spacetime dimensions. On top of that, let us remark that this type of breaking can also happen even if we find an *infinite* amount of new degrees of freedom with an associated mass scale $M = M_{\text{t}}$, namely an infinite tower of states of increasing masses starting at $M_{\text{t}}$.[2] In this second scenario, the theory cannot be rearranged —after integrating these objects in— into an EFT with a finite number of local fields in the same number of dimensions. Nevertheless, it is sometimes possible to uplift the latter again to a local field theory that lives instead in a higher-dimensional spacetime. This happens when the new degrees of freedom correspond to, e.g., Kaluza-Klein (KK) modes associated to some extra compact internal space.[3]

In any event, even though having a finite vs. an infinite number of new degrees of freedom makes a difference for the UV completion, these two scenarios share some common features that make them qualitatively similar from the viewpoint of the low-energy EFT expansion. Namely, in both cases the breaking of the effective description is due to the appearance of new local fields that are not included in the original theory, but need not be regarded as genuine quantum-gravitational effects. Relatedly, the fact that the EFT must break down is clear from the behavior of graviton-graviton scattering processes, which become non-unitary —due to the production of real intermediate states associated to the massive degrees of freedom— at energies close to and above $M$. In particular, since gravity couples universally to anything carrying energy and momentum, the gravitational sector of the theory necessarily requires the inclusion of these new particles.[4] Therefore, given that this EFT breakdown has a priori nothing to do with quantum gravity, the Wilsonian effective action is expected to encode the latter via the usual mechanism, i.e., the presence of higher-dimensional terms of the form

$$S_{\text{EFT},d} \supset \int \mathrm{d}^d x \sqrt{-g} \sum_{n>2} \frac{\mathcal{O}_n(\mathcal{R})}{M^{n-d}}\,. \tag{2}$$

Here we are focusing on operator-valued functions $\mathcal{O}_n(\mathcal{R})$ of classical dimension $n$ built purely out of the Riemann tensor. Notice that $M_{\text{Pl},d}$ does not appear explicitly in (2), which also suggests that the effects captured by this part of the action are not fundamentally quantum-gravitational in origin. This does not mean, however, that the Planck scale does not play any role here, since the amplitudes to which this kind of operators contribute ultimately depend on $M_{\text{Pl},d}$ through the interaction vertices, see Section 2 for details on this point.

---

[2] For clarity, we will explicitly use $M_{\text{t}}$ whenever we refer to an infinite amount of new degrees of freedom.

[3] In the context of string theory, making this extra-dimensional interpretation of the tower manifest can be rather involved, as oftentimes requires from switching to a dual local description of the physics (see, e.g., [88–90] and references therein).

[4] Similarly, above the threshold $M_{\text{t}}$ these states can appear as external legs in the scattering amplitudes, which is the ultimate cause for the divergences leading to a (perturbatively) non-unitary behavior.

Before moving on, let us remark that in general one could consider a theory exhibiting several potential UV scales, $M_i$. Hence, when some of these $M_i$ are of the same order, they should not be regarded as genuinely different cutoffs, since their contribution to higher-dimensional operators can be rearranged into a single term of the form displayed in (2). For simplicity, we will focus on the field-theoretic term related to the parametrically lowest scale, $M \equiv \min\{M_i\}$. In Section 2.3.2 below, we present a detailed top-down example featuring various cutoffs of this type. There, we will see how some contributions to a given higher-dimensional operator can encode in a non-trivial fashion the field-theoretic expansion within a decompactified theory due to a parametrically higher scale.

On the other hand, the second type of breaking in a given EFT is due to genuine quantum-gravitational phenomena. In contrast to the one discussed above, this is related not only to (possibly infinitely many) new high-energy degrees of freedom, but also to the inherent non-localities expected to arise in quantum gravity [91–94]. As such, above a certain energy scale where these effects should become apparent, henceforth denoted by $\Lambda_{\text{QG}}$, no local EFT coupled to Einstein gravity can serve as a bona-fide UV completion.[5]

The natural question that arises in this context is how this breakdown could be detected from the low-energy EFT perspective. Note that, since the ultra-violet completion is not a local field theory anymore, the intuitive argument from above does not apply herein. This point must, therefore, be addressed within the framework of a fully-fledged quantum gravity theory. Quite remarkably, the study of EFTs coming from string theory reveals that this lack of knowledge is naturally encoded into higher-derivative operators of the form [45, 47, 49–53]

$$S_{\text{EFT},d} \supset \frac{M_{\text{Pl},d}^{d-2}}{2} \int \mathrm{d}^d x \sqrt{-g} \sum_{n>2} \frac{\mathscr{O}_n(\mathcal{R})}{\Lambda_{\text{QG}}^{n-2}}. \tag{3}$$

Notice that, unlike in (2), the Planck scale does appear in this part of the action, and in fact the overall $M_{\text{Pl},d}^{d-2}$ factor ensures that these higher-derivative terms are genuine corrections to the two-derivative Einstein-Hilbert term appearing in (1). It is precisely in this sense that the EFT stops being valid due to quantum gravity effects.

The double EFT expansion, which we now introduce, naturally and elegantly encodes these two types of breakdown. It provides an organizational scheme for the higher-derivative corrections in the low-energy effective action. Upon combining eqs. (1)-(3), we get the following general structure for gravitational EFTs:

$$S_{\text{EFT},d} \supseteq \frac{M_{\text{Pl},d}^{d-2}}{2} \int \mathrm{d}^d x \sqrt{-g} \left( \mathcal{R} + \sum_{n>2} \frac{\mathscr{O}_n(\mathcal{R})}{\Lambda_{\text{QG}}^{n-2}} \right) + \int \mathrm{d}^d x \sqrt{-g} \sum_{n>2} \frac{\mathscr{O}_n(\mathcal{R})}{M^{n-d}}. \tag{4}$$

As the reader can observe, the name *double* refers to the two sums appearing in (4), which correspond to low-energy expansions around the two relevant UV scales discussed before, namely $M$ and $\Lambda_{\text{QG}}$. For the reasons detailed above, we refer to them as the *field-theoretic* and *quantum-gravitational* expansions, respectively. Notice that the EH action nicely fits as the $n = 2$ term of the quantum-gravitational piece. Before going on, let us stress that the double EFT expansion encodes the presence of an infinite number of higher-dimensional operators in the effective action of the form in (4), without necessarily implying the absence of extra terms exhibiting other structures. The possibility of having additional terms as well as their interplay with the ones shown in (2)-(3) will be discussed in more detail below.

The three scales that appear in (4) satisfy the hierarchy $M \lesssim \Lambda_{\text{QG}} \lesssim M_{\text{Pl},d}$. This is due to two simple facts. The first inequality comes from the type of breaking associated to $M$ being

---

[5]More precisely, what we mean is a local field theory containing a spin-2 massless graviton. According to the holographic principle, a possible UV completion above $\Lambda_{\text{QG}}$ could be in terms of a local *non-gravitational* quantum field theory [91–96].

milder than the one around $\Lambda_{\text{QG}}$, while the second arises from the fact that the quantum gravity breakdown ought to happen at most around the Planck scale [1]. That being said, the double EFT expansion becomes particularly meaningful in the presence of the stronger hierarchy

$$M \lesssim \Lambda_{\text{QG}} \ll M_{\text{Pl},d} \, . \tag{5}$$

In this case, the quantum-gravitational expansion is set apart from the field-theoretic one due to $\Lambda_{\text{QG}} \ll M_{\text{Pl},d}$. Otherwise, (3) looks exactly like (2) upon replacing $M \to M_{\text{Pl},d}$. In other words, we recover the usual (dimensional analysis) expectation that $M_{\text{Pl},d}$ plays the role of a cutoff for the gravitational sector. For this reason, when $\Lambda_{\text{QG}} \sim M_{\text{Pl},d}$, the double EFT expansion is not genuine and rather looks like the more familiar field theory structure with two, a priori, independent scales, $M$ and $M_{\text{Pl},d}$. In the presence of the alternative hierarchy $M \ll \Lambda_{\text{QG}} \sim M_{\text{Pl},d}$, all effects encoded in the *quantum-gravitational* expansion, (3), become subleading with respect to those in the *field-theoretic* one, (2). As we shall see later, this need not be the case when (5) holds instead.

Hereafter, we refer to (5) as the *asymptotic regime*. The reason is that, as introduced above, this hierarchy is related to infinite distance limits in quantum gravity moduli spaces, where by virtue of the Distance Conjecture [12] there is an infinite number of new degrees of freedom becoming light with respect to the Planck scale, i.e., $M_t \ll M_{\text{Pl},d}$. In addition, $\Lambda_{\text{QG}}$ can be identified with the *species scale* [39–42] in this regime [14, 43, 44]. Very nontrivially, this yields a relation between $M_t$ and $\Lambda_{\text{QG}}$ that guarantees that $\Lambda_{\text{QG}} \ll M_{\text{Pl},d}$ is also satisfied.[6] In contrast, the regime $M_t \sim \Lambda_{\text{QG}} \sim M_{\text{Pl},d}$ corresponds to the interior of the moduli space and is achieved in its most extreme form at the desert point [45, 47, 50, 97, 98]. From a bottom-up perspective, the hierarchy in eq. (5) is usually referred to as the weakly-coupled UV completion regime, since $d$-dimensional gravity —whose strength is controlled by $M_{\text{Pl},d}$— is weakly coupled when we reach either the EFT or the QG cutoff.

As discussed before, the double EFT expansion encodes that the original EFT is breaking down at energies/curvatures around $M \lesssim \Lambda_{\text{QG}}$. When $M \ll \Lambda_{\text{QG}}$, this breakdown is due to the field-theoretic expansion, which needs to be resummed (i.e., completed) at energies of the order of $M$ and above. As usual, after carrying out this procedure, the higher-dimensional operators in the field-theoretic expansion are entirely removed from the new effective action, which now incorporates additional degrees of freedom encoding the physics previously associated with those terms. On the other hand, the operators in the quantum-gravitational expansion do uplift to the UV-completed EFT, so that they keep fulfilling their role of signaling the quantum gravity breakdown at the scale $\Lambda_{\text{QG}}$. Additionally, we observe that at energies/curvatures of order $M$, each term in the field-theoretic expansion does not compete with the EH one due to the hierarchy $M \ll M_{\text{Pl},d}$. This is nothing but a manifestation that in reality this breaking is not quantum-gravitational, as already stressed. We will see all these features very explicitly both from the perspective of $2 \to 2$ graviton scattering amplitudes in Section 2, and from the viewpoint of (non-perturbative) black hole physics in Section 3.

As a summary of the discussion so far, we depict the different features of the physics encoded into the double EFT expansion in Figure 1.

So far, we have discussed the field-theoretic and quantum-gravitational expansions within the double EFT framework, which imply the existence of an infinite number of higher-dimensional terms of the form (2) and (3). For the remainder of this section, we shift our focus to a single Wilson coefficient and examine the interplay between these two expansions when taken at face value. Indeed, the same operator $\mathcal{O}_n(\mathcal{R})$ can, and generically will, appear in both expansions. Thus, the double EFT scheme predicts that the Wilson coefficient accompanying any gravitational operator of dimension $n$ typically takes the following form in Planck

---

[6]Assuming that the new degrees of freedom are weakly coupled, this also reveals that the regime $M \ll \Lambda_{\text{QG}} \sim M_{\text{Pl},d}$ discussed above can only happen when $M$ refers to a finite number of them.

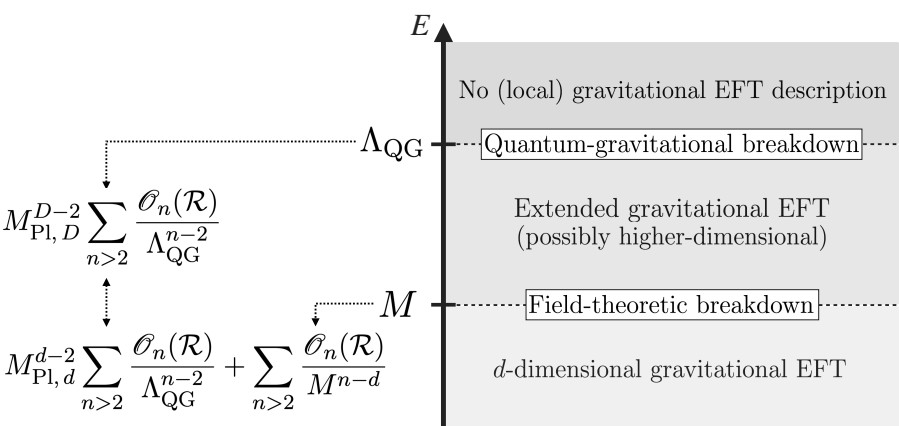

Figure 1: Schematic representation of the physics encoded into the double EFT expansion proposal. The figure shows on the vertical axis the field-theoretical and quantum-gravitational scales, $M$ and $\Lambda_{\mathrm{QG}}$, respectively. On the right, we depict the various energy regimes and clarify their associated descriptions. On the left, we indicate the various higher-dimensional operators that appear within the corresponding effective action and the scale they are related to. Below the UV cutoff, $M$, the system is described by a $d$-dimensional gravitational EFT exhibiting higher-dimensional operators suppressed by both $M$ and $\Lambda_{\mathrm{QG}}$. At energies of order $M$, this EFT suffers from a field-theoretic breakdown, such that above this scale it is possible to UV-complete the theory upon integrating in the new degrees of freedom. This leads to a (possibly higher-dimensional) extended gravitational EFT. As indicated in the figure, the field-theoretic effects disappear from the Wilsonian effective action, whilst the quantum gravity expansion still survives. At energies of the order of $\Lambda_{\mathrm{QG}}$, the latter effective description goes through a quantum-gravitational transition. Above the latter, the theory cannot be recast into another (local) gravitational EFT, but actually requires from a fully-fledged theory of quantum gravity (e.g., string theory). Notice that, whenever $\Lambda_{\mathrm{QG}} \sim M_{\mathrm{t}}$ the quantum-gravitational phase transition occurs directly at that scale, without an extended EFT description happening in between.

units (i.e., $M_{\mathrm{Pl},d} = 1$)

$$\alpha_n = \frac{a_n}{\Lambda_{\mathrm{QG}}^{n-2}} + \frac{b_n}{M^{n-d}} + \cdots. \tag{6}$$

Here, $a_n$ and $b_n$ are numerical coefficients which are expected to either be (larger than) some order one number or to vanish exactly.[7] From this perspective, the double EFT expansion predicts an infinite number of Wilson coefficients with non-vanishing $a_n$ and/or $b_n$ so that they correctly capture the cutoff scales $M$ and $\Lambda_{\mathrm{QG}}$. Additionally, the ellipsis encodes that there could be extra contributions to $\alpha_n$ that are not displayed explicitly in (4). In the following, we focus on the implications of the field-theoretic and quantum-gravitational terms —i.e., those explicitly displayed in the expansion above— and return to the additional terms at the end of this section.

An interesting feature about (6) is that having $M \ll \Lambda_{\mathrm{QG}}$ does not guarantee a priori that the Wilson coefficient will be dominated by the field-theoretic term. The reason for this is that the quantum-gravitational contribution is suppressed with a power of $\Lambda_{\mathrm{QG}}$ that is strictly smaller than that exhibited by $M$ in (2). Therefore, due to the very different nature of these two scales, $\Lambda_{\mathrm{QG}}$ might provide the leading-order contribution to some Wilson coefficients, as already observed in [45, 47, 49, 50, 52].[8] In other words, one could hope to perhaps be able

---

[7]For instance, they are absent when the operator under consideration cannot appear in the effective action due to a preserved symmetry of the system, like supersymmetry.

[8]Relatedly, the dominant corrections to other purely non-perturbative gravitational observables, such as the black hole entropy, can also be controlled by the quantum-gravitational expansion rather than the field-theoretic

to measure the quantum gravity cutoff at low energies even if $M \ll \Lambda_{\text{QG}}$ is satisfied! On the other hand, this can only happen for operators with sufficiently small dimension (i.e., not too irrelevant in the Wilsonian sense), as the difference in powers gets washed away when considering $\mathcal{O}_n(\mathcal{R})$ with $n \gg d$.

Furthermore, from a bottom-up perspective it is a non-trivial task to determine which gravitational Wilson coefficients should be dominated by what part of the double EFT expansion. Nevertheless, this question can be addressed in a rather natural way from a top-down perspective, whenever the scale $M$ corresponds to an infinite number of states (i.e., $M = M_t$). As encoded in the Emergent String Conjecture [25], the leading tower in all known string theory models is comprised either by the KK states associated to some extra dimensions or the excitation modes of a weakly coupled string. The species scale corresponding to these lightest towers therefore recovers the higher-dimensional Planck scale and the string scale, respectively. Despite this, in general, the Emergent String Conjecture does not completely fix $\Lambda_{\text{QG}}$ in terms of $M_t$ due to possible subleading towers that can lower the species scale with respect to the contribution of the leading one. Taking this into account, the Emergent String Conjecture implies the bound

$$\Lambda_{\text{QG}} \lesssim M_t^{\frac{p}{d-2+p}}, \quad \text{with} \quad p \in \mathbb{N}, \tag{7}$$

where $p$ captures the number of extra dimensions becoming large, and with emergent string limits effectively captured by taking $p \to \infty$ [99]. From a top-down perspective, this is naturally interpreted as $\Lambda_{\text{QG}}$ being always below but not necessarily at the higher-dimensional Planck scale. Given this relation, we see that the *quantum-gravitational* term dominates if $n < d + p$, i.e., for relevant operators in $(d+p)$-dimensions along decompactification limits and for all operators in weakly-coupled string limits [49, 53]. Additionally, when the species scale is determined by the leading tower —such that (7) is saturated— the *field-theoretic* piece gives the dominant contribution only if $n > p + d$, i.e., for an infinite number of irrelevant operators in $(d + p)$-dimensions along decompactification limits [49, 53].

The double EFT expansion also reveals that comparing the contribution from a certain higher-derivative operator with the EH term can sometimes yield an overestimation of the EFT cutoff. For instance, if a Wilson coefficient is dominated by the quantum-gravitational contribution, upon contrasting with the two-derivative term we would be tempted to conclude that the cutoff is set by $\Lambda_{\text{QG}}$. Nevertheless, if the scale $M \ll \Lambda_{\text{QG}}$, the actual regime of validity for the EFT turns out to be much smaller than the one suggested by the aforementioned operator. On the other hand, even if the behavior of the Wilson coefficient is controlled by the field-theoretic contribution, comparing the latter with the EH term will always yield an overestimation of the cutoff. This is related to the previously discussed fact that, at energies/curvatures of order $M$, the field-theoretic expansion does not strictly compete with the tree-level Einstein-Hilbert term. In any event, this should not be regarded as a drawback from our discussion but rather as a feature. Indeed, it signals in a very natural and clear way that the EFT breakdown is not due to purely quantum gravity effects.

Following the observations above, the following natural question arises: Given the effective action or the low-energy expansion of a given gravitational observable, how can we correctly estimate the EFT cutoff, as well as determine whether it is field-theoretic or quantum-gravitational in origin? To achieve this, we must compare each contribution in the low-energy expansion not only with the leading term, but also with all the remaining ones. Upon doing so, once energies of the order of the cutoff are reached, one would detect that several (actually infinitely many) corrections become of the same order, thus signaling the breakdown of the field theory expansion. Crucially, however, let us stress that the tree-level EH contribution need not be one of such terms. In fact, comparing directly with (1) allows us to diagnose whether

one (see Section 3 for more on this).

the EFT cutoff is of the field-theoretic or quantum-gravitational type, namely $M \ll \Lambda_{QG}$ vs. $M \sim \Lambda_{QG}$. As already discussed, in the former case the EH term is not part of the low-energy expansion that is breaking down, while in the latter case it does.

Let us finally comment on the extra terms encoded by the ellipsis in (6). As we elaborate further in Section 2.3.2 below, higher-derivative corrections that do not fit into the double EFT expansion do indeed appear in top-down string theory examples. Since we do not have a completely general characterization of their detailed form, the following relevant question arises: Could one of these extra terms yield the leading contribution to a Wilson coefficient once we sit in an asymptotic regime? Quite remarkably, we find a negative answer to this question in all the string theory examples analyzed herein. In other words, the double EFT expansion seems to possess the powerful feature of capturing the leading contribution to any non-vanishing Wilson coefficient in top-down constructions. We present some compelling arguments as to why we expect this be the case more generally in Section 2.3.2. Motivated by this observation, we will study different bounds on Wilson coefficients encoded in the double EFT expansion proposal in Section 4.

## 1.2 Summary of results and outline

The plan for the rest of the paper will be to further motivate, test and formalize this idea by studying different string theory constructions, where the structure of certain higher-curvature operators has been already analyzed in great detail. In particular, we test the double EFT expansion proposal through the genus expansion of superstring theories and maximal supergravities arising from toroidal compactifications thereof (Section 2), 4d $\mathcal{N} = 2$ models derived from Type II Calabi–Yau compactifications (Section 3.1), and 6d $\mathcal{N} = (1, 0)$ and 5d $\mathcal{N} = 1$ theories obtained from F-theory and M-theory on Calabi–Yau threefold (Appendix B).

Our aim will be to understand, from the point of view of the low energy EFT, how the gravitational sector of the theory can be sensitive to the relevant scales controlling the higher-derivative expansion, as displayed in (4). To do so, we will mainly follow two different but complementary approaches, namely we will focus on the behavior of certain gravitational observables and study their precise dependence as a function of the energies/curvatures that are probed. This is the content of Sections 2 and 3, which analyze this question from the perspective of gravitational amplitudes and black hole physics, respectively.

Subsequently, in Section 4 we will exploit the top-down lessons extracted from our previous discussions so as to motivate non-trivial upper and lower bounds on the corresponding gravitational Wilson coefficients. This, in turn, allows us to exhibit the powerful constraints that can be potentially derived from the double EFT expansion, showcasing the nice interplay with current S-matrix bootstrap bounds [54, 57–60, 63, 64, 67].

Finally, in Section 5, we discuss important questions that this work may leave open and highlight potential directions for future exploration based on our findings.

In the following we provide a brief summary where the main results of each section are highlighted, hopefully helping the reader navigate across the topics they might find more interesting.

**The amplitudes perspective**

In Section 2, we study the different kinds of breaking of gravitational EFTs through the lens of graviton scattering amplitudes, with particular emphasis on four-point graviton scattering. To this end, we make extensive use of previous results in [100–109]. This analysis highlights how the double EFT expansion organizes the higher-derivative corrections and captures the breaking scales associated to the different asymptotic regimes.

- **The breaking of gravitational EFTs:** We use four-point graviton scattering amplitudes to define the scale at which the EFT ceases to be valid. This breaking occurs when the higher-curvature operators, such as $D^{2\ell}\mathcal{R}^4$, start being comparable to the leading, two-derivative, tree-level term and/or to each other, as explained in detail in Section 2.2.3. This signals the onset of new physics, requiring a UV completion at a scale given by the smallest out of $M_\text{t}$ or $\Lambda_\text{QG}$, which encodes the kind of transition that the EFT must undergo.

- **Decompactification limits and the role of field-theoretic terms:** We analyze decompactification limits in top-down constructions of maximally supersymmetric theories in ten dimensions and toroidal compactifications thereof (Section 2). We show that for a small amount of operators in the EFT —the ones that are classically relevant or marginal in the decompactified theory— the *quantum-gravitational* contribution dominates the higher-derivative correction. Furthermore, in Sections 2.2.1 and 2.2.2 we argue that *field-theoretic* terms are always present for an infinite number of higher-curvature operators —the ones that are classically irrelevant in the decompactified theory— and dominate over putative *quantum-gravitational* ones at scales below $M_\text{t}$. This is aligned with the predictions coming from the double EFT expansion. In fact, this kind of breaking due to an arbitrarily high number of operators suppressed by the scale $M_\text{t} \ll \Lambda_\text{QG}$ is shown to correspond to a transition to the higher-dimensional theory. Indeed, the massive thresholds associated to the states of the tower can be resummed into the non-analytic massless threshold of the decompactified theory, as reviewed in Section 2.2.3. To illustrate this point, we discuss in detail the Type IIA/M-theory decompactification limit (cf. Appendix A). Hence, the *field-theoretic* terms explicitly disappear from the higher-curvature operators at scales well above $M_\text{t}$, whereas the *quantum-gravitational* contributions remain and are thus uplifted to the higher-dimensional theory, as discussed at the end of Section 2.2.3.

- **The double EFT expansion in weak coupling string limits:** From top-down constructions, limits where $M_\text{t} \sim \Lambda_\text{QG}$ are those in which the tower is given by the higher-spin excitations of a weakly coupled, critical string. We argue in Section 2.3.1 how the *quantum-gravitational* term in the double EFT expansion (4) corresponds to the tree-level term in string perturbation theory, whereas the *field-theoretic* one is given by the one-loop contribution [49]. This implies that, as long as it is present, the former will always dominate when $g_s \ll 1$, and it also provides a purely quantum-gravitational mechanism for the EFT failure. Moreover, in Section 2.3.2 we elaborate on the additional terms that naturally appear in this setup, making extensive use of the duality between M-theory and Type IIA to discuss them in the decompactification limit. In all top-down examples, these extra contributions always appear to be *subleading* with respect to those captured by the double EFT expansion. To strengthen this observation, we also present in Section 2.3.2 some compelling arguments as to why the double EFT expansion seems to capture the leading term in every non-vanishing Wilson coefficient.

### The black hole perspective

In Section 3 we investigate how certain thermodynamic properties associated to black hole solutions are able to detect the relevant UV cutoff scales of the gravitational EFT. More precisely, we focus on the supersymmetric entropy index of certain BPS black holes in 4d $\mathcal{N} = 2$ supergravities derived from Type II string compactifications. The main results of this section can be summarized as follows:

- **General agreement with the double EFT expansion proposal:** Building on earlier works [45, 47, 49, 53], we lay out in Section 3.1.2 the structure of certain protected higher-curvature and higher-derivative corrections within this class of theories. Their moduli dependence (in the vector multiplet sector) is shown to exhibit precise agreement with the expectations arising from the double EFT expansion for all different kinds of asymptotic regimes that can arise therein. More concretely, we find the corresponding Wilson coefficients to behave as $\alpha_{2k+2} = \mathcal{F}_k \sim M_{\rm t}^{2k-2}$ for $k > 1$ and $\alpha_4 = \mathcal{F}_1 \sim \Lambda_{\rm QG}^2$ for $k = 1$, in accordance with (4).

- **Transition between EFT regimes:** Using recent [110] and prior results [111–118], we illustrate in Section 3.2 how the EFT breakdown due to field-theoretic quantum effects is reflected in the black hole entropy, which is shown to behave smoothly once the higher-derivative corrections and non-local resummation effects are properly taken into account. Moreover, the leading-order quantum-gravitational correction to the black hole entropy is shown to survive the transition to the higher-dimensional theory and therefore controls the relevant thermodynamic properties of minimal-sized supersymmetric black holes [39, 40]. This illustrates our previous observation that, even though a priori $M_{\rm t} \ll \Lambda_{\rm QG}$, the quantum-gravitational piece (3) can provide the dominant contribution to certain gravitational observables, such as the black hole entropy.

**The bottom-up perspective**

Building on the general lessons drawn from top-down models, in Section 4 we use the double EFT expansion to derive various types of bounds on Wilson coefficients in the asymptotic regime. In several cases, we show how assuming the scale $M$ corresponds to an infinite tower of states —namely $M = M_{\rm t}$— and imposing the Emergent String Conjecture [25], lead to stronger constraints. Key results include:

- **Constraints on Wilson coefficients and the EFT cutoff:** We start in Section 4.1 by considering bounds on a certain Wilson coefficient in terms of the EFT cutoff, $M$. The double EFT expansion naturally encodes the upper bound in (83), which is generally expected to be implied by causality [54] (see also [58–60, 63, 67, 68]). We elaborate more on the comparison between (83) and S-matrix bootstrap results in Section 4.3, arguing that the energy scale appearing in the latter must be identified with $M$ and not with $\Lambda_{\rm QG}$. Additionally, in Section 4.1 we also derive lower bounds on non-vanishing Wilson coefficients under the assumption that the double EFT expansion captures their leading behavior, a feature that we observe in all top-down examples.

- **Scaling relations between Wilson coefficients:** Following a similar reasoning, we show in Section 4.2 how the double EFT expansion encodes bounds on scaling relations between pairs of non-trivial Wilson coefficients. In particular, we consider $\mathcal{R}^4$ and $D^4\mathcal{R}^4$ corrections to maximal supergravity theories in 10d as an illustrative example. Remarkably, the S-matrix bootstrap results of [57, 64] inform the double EFT expansion, by implying that the $\mathcal{R}^4$ Wilson coefficient should receive a non-vanishing quantum-gravitational contribution. This demonstrates how the proposed structure (4) can be used as an interpretative framework for these results.

- **Comparison with S-matrix bootstrap bounds:** The same logic can be applied to derive bounds on the dimensionless Wilson coefficients $\tilde{\alpha}_n$ defined in (100), which are the natural target for the techniques developed in [58–60, 63, 67, 68]. Again, focusing on $\mathcal{R}^4$ and $D^4\mathcal{R}^4$ corrections to 10d maximal supergravity theories, we derive such bounds in Section 4.3. We compare our results with the ones presented in [67], finding a very

nice interplay. Without any further assumption, the double EFT does not provide any lower bound on $\tilde{\alpha}_{D^4\mathcal{R}^4}$ in terms of $\tilde{\alpha}_{\mathcal{R}^4}$. Nevertheless, the constraint put forward in [67] is recovered upon assuming that the $D^4\mathcal{R}^4$ coefficient should be lower bounded by the quantum-gravitational term. This way, the S-matrix bootstrap results again inform the double EFT expansion. On the other hand, we find that the upper bounds (106) and (107) —with and without the Emergent String Conjecture— are stronger than that found in [67].

## 2 The amplitudes perspective

The purpose of this section is twofold. On the one hand, we explicitly illustrate the different features of the double EFT expansion introduced in Section 1.1 from top-down string theory constructions. On the other hand, we highlight —in a rather general and systematic way— how to determine the scale at which a given gravitational effective field theory breaks down, as identified through scattering experiments involving exclusively the gravity sector. To do so, we will focus on a particular gravitational observable, namely the $k$-point graviton scattering amplitude (with particular emphasis in the $k = 4$ case), and show that it contains all the necessary information so as to answer our original questions. Moreover, it captures the main lessons we want to draw from this work.

For concreteness, we start with a gravitational EFT coupled to a finite number of degrees of freedom, assumed to be a good description of our theory at low energies.[9] Then, the breaking scale of the EFT in the gravitational sector is identified as the one at which the exact $k$-graviton amplitude departs significantly from the one calculated with the aforementioned EFT, including also all possible higher-dimensional operators. However, since we typically have access only to the effective description, we estimate this scale by examining when these operators begin to compete with the two-derivative contribution or with one another, signaling the need for modification or (partial) UV completion of the EFT.

In the presence of towers of states, which are ubiquitous in quantum gravity —and particularly relevant near the asymptotic regime as per the Distance Conjecture [12]— there exists a dichotomy of whether the aforementioned scale at which the (gravitational sector of the) EFT breaks down is given by the characteristic mass of the tower, $M_t$, or rather by its associated species scale $\Lambda_{QG}$ [47, 49, 50, 119]. As we will illustrate below, by focusing on the four-point graviton amplitude one can argue that the EFT breaks down —in the sense specified above— at the smallest of these two, given that the UV cutoff is generically set by the mass of the first state that is not included in the original effective description. Therefore, since $M_t \lesssim \Lambda_{QG}$, this should always correspond to the mass gap of the tower, $M_t$, whenever it exists. If the tower is such that $M_t \sim \Lambda_{QG}$ (e.g., for the vibrational modes of a weakly coupled fundamental string), one can then see the breaking of the EFT happening at the species scale.

As already stressed in Section 1.1, the main difference between these two scenarios is that, when $M_t \ll \Lambda_{QG}$, the theory can be UV-completed to another local EFT in which additional fields (i.e., those included in the corresponding infinite tower of states) are incorporated. This is precisely the case whenever the tower is incarnated by the Kaluza-Klein modes associated to some extra compact dimensions. In this context, the gravitational amplitudes can be re-summed for energies greater than or equal to $M_t$, thus yielding a well-behaved observable that should be regarded as higher-dimensional in origin. The point we want to emphasize here is that the original effective field theory is sensitive —via $k$-point graviton scattering observables— to the scale $M_t$, and as such must be corrected at energies around it.

---

[9]We consider herein a gravitational EFT including a finite number of dynamical fields, which are either massless or light enough such that the gravitational amplitudes are well described by the corresponding effective action at scales above their mass. For simplicity, we loosely dub this set the *massless sector*, but in a more general context it may include any finite number of degrees of freedom with masses $m \ll M$.

On the other hand, the breaking associated to reaching the quantum gravity cutoff becomes much more severe, given the lack of existence of another local gravitational EFT description of the relevant physics occurring above that energy (cf. footnote 5). Therefore, we expect the $k$-point graviton amplitudes to require from further highly-non local ingredients so as to be corrected at energies close to and above $\Lambda_{\text{QG}}$.

## 2.1 10d type IIA string theory

In this section, we focus on 10d Type IIA string theory, since it provides an ideal arena to illustrate the points summarized above. The first few low-lying gravitational corrections to the Einstein-Hilbert action are exactly known, hence allowing us to identify the contributions encoded in the double EFT expansion in a very precise way. Accordingly, we will see how the $2 \to 2$ graviton scattering amplitude behaves when probing the different relevant scales displayed in (4). Despite focusing on this particular example for simplicity, we want to stress that essentially the same considerations translate into the more general setup of toroidal compactifications of Type II/M-theory in $d \geq 4$ [49, 53].

As we will discuss below, there are two different asymptotic regions in this scenario that can be probed, namely a weakly-coupled tensionless string limit and a decompactification limit. These are the only kinds of limits that have been observed so far from the top-down and they are in fact believed to be exhaustive, as per the Emergent String Conjecture [25]. Thus, this simple setup provides a glimpse into the type of physics that are most relevant to test the double EFT expansion within string theory. In subsequent sections, we will discuss this two classes of asymptotic regimes in more generality, also taking into account higher-curvature and higher-derivative operators beyond the first low-lying ones.

### 2.1.1 Four-point graviton scattering

Consider the four-point amplitude involving only external gravitons with polarization tensors $\zeta^{(n)}_{\mu\nu}$ and momenta $k^{(n)}_\mu$, subject to the on-shell condition $(k^{(n)})^2 = 0$. Following the notation of [103–109, 120–124], the latter reads —in the 10d Einstein frame— as

$$\mathcal{A}_4(s,t) = \mathcal{A}_4^{\text{sugra}} f(s,t), \qquad \text{with} \quad \mathcal{A}_4^{\text{sugra}} = -\hat{K} \, 128\pi^2 \ell_{\text{Pl},10}^2 (stu)^{-1}. \tag{8}$$

Here $\mathcal{A}_4^{\text{sugra}}$ is the tree-level four-graviton amplitude computed from the two-derivative supergravity Lagrangian [125, 126],[10] the quantities $s = -(k^{(1)} + k^{(2)})^2$, $t = -(k^{(1)} + k^{(4)})^2$ and $u = -(k^{(1)} + k^{(3)})^2 = -s - t$ correspond to the Mandelstam variables, whereas

$$\hat{K} = t^{\mu_1 \cdots \mu_8} t^{\nu_1 \cdots \nu_8} \prod_{n=1}^4 \zeta^{(n)}_{\mu_{2n} \nu_{2n}} k^{(n)}_{\mu_{2n-1}} k^{(n)}_{\nu_{2n-1}}, \tag{9}$$

denotes the kinematic factor of the amplitude [125, 133], with

$$\begin{aligned}
t^{\mu_1 \cdots \mu_8} = \frac{1}{5}\Big[ &-2\big(\eta^{\mu_1\mu_3}\eta^{\mu_2\mu_4}\eta^{\mu_5\mu_7}\eta^{\mu_6\mu_8} + \eta^{\mu_1\mu_5}\eta^{\mu_2\mu_6}\eta^{\mu_3\mu_7}\eta^{\mu_4\mu_8} + \eta^{\mu_1\mu_7}\eta^{\mu_2\mu_8}\eta^{\mu_3\mu_5}\eta^{\mu_4\mu_6}\big) \\
&+ 8\big(\eta^{\mu_2\mu_3}\eta^{\mu_4\mu_5}\eta^{\mu_6\mu_7}\eta^{\mu_1\mu_8} + \eta^{\mu_2\mu_5}\eta^{\mu_3\mu_6}\eta^{\mu_4\mu_7}\eta^{\mu_1\mu_8} + \eta^{\mu_2\mu_5}\eta^{\mu_6\mu_7}\eta^{\mu_3\mu_8}\eta^{\mu_1\mu_4}\big) \\
&- (\mu_1 \leftrightarrow \mu_2) - (\mu_3 \leftrightarrow \mu_4) - (\mu_5 \leftrightarrow \mu_6) - (\mu_7 \leftrightarrow \mu_8) \Big].
\end{aligned} \tag{10}$$

The function $f(s,t)$ captures additional contributions arising from an infinite number of corrections in the derivative expansion of the 10d Type IIA action and normalized with respect to the tree-level term in (8). These include higher-curvature 10d local operators of the

---

[10]See, e.g., [127, 128] for a supersymmetric generalization of the amplitude (8) using the pure spinor formalism developed in [129–132].

schematic form $D^{2\ell}\mathcal{R}^4$, with $\ell = 0, 1, \ldots, \infty$. The first three non-trivial ones yield

$$f(s,t) = 1 + \ell_{\text{Pl}, 10}^6 f_{\mathcal{R}^4}(s,t) + \ell_{\text{Pl}, 10}^{10} f_{D^4\mathcal{R}^4}(s,t) + \ell_{\text{Pl}, 10}^{12} f_{D^6\mathcal{R}^4}(s,t) + \cdots, \tag{11}$$

where the ellipsis is meant to indicate further higher-dimensional terms as well as higher-loop and non-analytic contributions at zero-momentum [109] (cf. Section 2.2.3 for more on this). In addition, we are expressing everything in terms of the 10d Planck length, which can be related to the string scale via the string coupling constant $g_s$ in the usual way, namely $\ell_{\text{Pl}, 10}^8 = \ell_s^8 g_s^2$. Hence, according to our previous discussion, we may estimate the energy scale where the gravitational sector of the original EFT breaks down as the one at which any of the different contributions in (11) —or sums thereof— naively becomes of order one, thus signaling that the amplitude starts to deviate significantly from the one provided by 10d Type IIA supergravity.

The main advantage of this setup is that the first few corrections to $\mathcal{A}_4^{\text{sugra}}$ are known exactly [100, 101, 105], since they are protected by supersymmetry, and they in fact suffice to identify the cutoff scale of our theory. They take the form[11] [103, 104, 134–137]

$$f_{\mathcal{R}^4} = \frac{stu}{(4\pi)^6} \left( 2\zeta(3) g_s^{-3/2} + \frac{2\pi^2}{3} g_s^{1/2} \right), \tag{12a}$$

$$f_{D^4\mathcal{R}^4} = \frac{stu(s^2 + t^2 + u^2)}{(4\pi)^{10}} \left( \zeta(5) g_s^{-5/2} + \frac{2\pi^4}{135} g_s^{3/2} \right), \tag{12b}$$

$$f_{D^6\mathcal{R}^4} = \frac{stu(s^3 + t^3 + u^3)}{(4\pi)^{12}} \left( \frac{2\zeta(3)^2}{3} g_s^{-3} + \frac{4\zeta(2)\zeta(3)}{3} g_s^{-1} + \frac{8\zeta(2)^3}{5} g_s + \frac{4\zeta(6)}{27} g_s^3 \right). \tag{12c}$$

In what follows, we explore the two asymptotic limits that one can take in the present example, namely $g_s \ll 1$ and $g_s \gg 1$, where the double EFT expansion presented in Section 1.1 will become manifest. Before delving into the details, we note in advance that a given term in (12) can and will play a different role depending on the asymptotic regime. This feature is closely related to the concept of dualities. Each asymptotic regime is most naturally viewed from the perspective of the weakly-coupled duality frame associated to that limit, i.e., Type IIA or M-theory in the present case. As is well known [138], a given contribution to an observable can receive different interpretations when viewed from different dual frames.

### 2.1.2 The weak coupling regime

In the weak coupling limit, namely when $g_s \ll 1$, the quantum gravity cutoff is identified with the string scale [49, 50], namely

$$\Lambda_{\text{QG}} \sim M_{\text{t}} = M_s = (4\pi)^{-1/8} g_s^{1/4} M_{\text{Pl}, 10}. \tag{13}$$

Therefore, in this regime, the first term in each expression within (12) dominates, thus leading to a suppression —with respect to the EH term— of the form $M_s^{n-2}$ for each operator individually, since their corresponding classical dimensions are $n = 8, 12, 14$, respectively. As we will discuss in Section 2.3.1, this conclusion can be easily extended to an infinite tower of operators of the form $D^{2\ell}\mathcal{R}^4$ by considering the scaling of the tree-level string scattering amplitude and the perturbative genus expansion.

This behavior indicates that, for the weakly coupled fundamental Type IIA string, the supergravity EFT breaks down at the string scale, where all the non-localities associated to its

---

[11]Note that the higher-curvature contributions to the Type IIA four-graviton amplitude shown in (12) precisely match the ones obtained in Type IIB string theory upon removing the D(-1)-instanton series [109, 134].

extended nature are made manifest through the appearance of an infinite tower of higher-spin states, which do not admit any local interacting field theory description (see, e.g., [3]).[12] Indeed, for $\sqrt{s} \sim \sqrt{t} \sim M_s$, all corrections in (11) become of $\mathcal{O}(1)$, such that one needs to resort to the full string theory framework in order to be able to correctly compute the amplitude $\mathcal{A}_4(s,t)$. From our discussion above, this embodies the paradigmatic example of the kind of breaking associated to purely *quantum-gravitational* effects, where no local quantum field theory can be used as an effective description above $\Lambda_{\mathrm{QG}} \sim M_s$.

Let us also remark that, even though the coefficients (12) are not derived through any field-theoretic computation —they are rather obtained from quantum strings propagating in spacetime— there also exist *field-theoretic* terms of the form (2) within the higher derivative expansion. Indeed, these correspond to a one-loop correction in string perturbation theory [49]. This can be explicitly seen in both $\mathcal{R}^4$ and $D^6\mathcal{R}^4$ terms in (12), whose genus-one contribution is indeed suppressed by $M_s^{n-d}$ with $n = 8, 14$ respectively. We also emphasize that the aforementioned one-loop correction exactly vanishes for $D^4\mathcal{R}^4$ [103].

Notice that the $D^4\mathcal{R}^4$ and $D^6\mathcal{R}^4$ terms contain further contributions that are not encoded in the double EFT expansion. These correspond to the extra terms captured by the ellipsis in (6). Crucially, we observe that these are always subleading with respect to the quantum-gravitational or field-theoretical ones, such that the double EFT expansion captures the leading-order contribution in the asymptotic regime.

All in all, we conclude that the expectations coming from the double EFT expansion displayed in (4) are borne out in the weak-coupling limit of 10d Type IIA string theory.

### 2.1.3 The strong coupling regime

Let us now turn to the strong coupling regime, $g_s \gg 1$, which corresponds to the decompactification limit of eleven-dimensional supergravity on a circle [89]. In this case, the quantum gravity scale is set by the 11d Planck mass [49,50,139], and the associated KK tower is given by (bound states of) D0-branes from the ten-dimensional picture. We thus have

$$\Lambda_{\mathrm{QG}} \sim M_{\mathrm{Pl},11} = (2\pi)^{-1/8} g_s^{-1/12} M_{\mathrm{Pl},10},$$
$$M_{\mathrm{t}} = M_{\mathrm{D0}} = 2\pi M_s / g_s = (4\pi)^{7/8} M_{\mathrm{Pl},10} / 2 g_s^{3/4}. \tag{14}$$

In what follows, we identify these relevant scales as captured by the three curvature corrections shown in (12) upon comparing them with the double EFT expansion presented in (4).

Let us start with the correction associated to the eight-derivative operator $t_8 t_8 \mathcal{R}^4$. In this limit, the second term in (12a) dominates. This contribution can be seen to be suppressed by $M_{\mathrm{Pl},11}^6$, as expected from the *quantum-gravitational* piece of the expansion (3) for an operator of classical dimension $n = 8$ in 10d [48–50]. Notice that the subleading *field-theoretic* contribution can now be identified with the first term in $f_{\mathcal{R}^4}(s,t)$, which is proportional to $M_{\mathrm{t}}^2 M_{\mathrm{Pl},10}^{-8}$, in agreement with (2).[13] If the series of higher-curvature operators had stopped here, one would have been tempted to conclude that the original EFT breaks down at energies of the order of $\Lambda_{\mathrm{QG}}$, since for scattering experiments with $\sqrt{s} \sim \sqrt{t} \sim M_{\mathrm{Pl},11}$ the correction term (12a) becomes comparable to the tree-level contribution. However, this is actually not the case, as can be seen by checking the remaining higher-order corrections in the derivative expansion (11), which do become of $\mathcal{O}(1)$ at a (parametrically) lower scale.

---

[12]From the bottom-up perspective, i.e., without assuming string theory, the energy scale at which the gravitational EFT must break down —sometimes due to higher-spin particles starting to dominate the amplitudes— has been studied recently in great generality in [69, 71]. This cutoff, denoted as the *higher-spin onset* scale in [69], was found to agree with the species scale in the presence of towers of light particles.

[13]The reason why this term is subleading is because of dimensional analysis, since it can be seen to be relevant (in the Wilsonian sense) both in the 10d and 11d theories, given that $n = 8 < 10 < 11$ [49,53].

In fact, $D^4\mathcal{R}^4$ and $D^6\mathcal{R}^4$ are the prototypical example that show that this can indeed happen within the gravitational sector. Their corresponding leading terms in eqs. (12b) and (12c) are easily seen to be suppressed with respect to the tree-level one by $M_t^2 M_{\mathrm{Pl},10}^8 = 4^{3/4}\pi^{7/4} g_s^{-3/2} M_{\mathrm{Pl},10}^{10}$, and $M_t^4 M_{\mathrm{Pl},10}^8 = 4^{3/2}\pi^{7/2} g_s^{-3} M_{\mathrm{Pl},10}^{12}$, respectively. This precisely matches with the prediction given by the *field-theoretic* corrections in (2). Notice that the latter provides the dominant contribution precisely for irrelevant operators in the decompactification limit, i.e., $n = 12$, $14 > d + p = 11$. Furthermore, the *quantum-gravitational* piece, which would rather correspond to a term suppressed by $M_{\mathrm{Pl},11}^{10}$ in (12b), is not present. This encapsulates the fact that the $D^4\mathcal{R}^4$ operator exactly vanishes in eleven-dimensional M-theory [100, 101, 104, 105, 123, 140, 141]. On the other hand, for $D^6\mathcal{R}^4$, the third term in (12c) can be identified with the aforementioned *quantum-gravitational* contribution, as it is precisely suppressed by $M_{\mathrm{Pl},11}^{12}$. Finally, we also note that both $D^4\mathcal{R}^4$ and $D^6\mathcal{R}^4$ receive other contributions that are nevertheless even more subleading, thus fitting with the terms encoded in the ellipsis in (6).

Focusing now on the leading-order behavior exhibited by these two corrections, one can easily bound the cutoff of the theory upon comparing each of them with the tree-level amplitude. This yields $\sqrt{s} \sim \sqrt{t} \sim M_t^{\frac{1}{5}} M_{\mathrm{Pl},10}^{\frac{4}{5}}$ and $\sqrt{s} \sim \sqrt{t} \sim M_t^{\frac{1}{3}} M_{\mathrm{Pl},10}^{\frac{2}{3}}$ as the scales around which the operators $D^4\mathcal{R}^4$ and $D^6\mathcal{R}^4$ become relevant. Let us remark that this latter observation allows us to extract two important lessons which, as we show in upcoming sections, hold more generally. First of all, both terms indicate that the gravitational EFT breaks down at scales well below the quantum gravity cutoff. Second, each term independently provides some energy scale which is parametrically smaller —in the asymptotic regime (5)— than that obtained from the previous operator.

Alternatively, and following the discussion of Section 1.1, we can also try to compare such terms among themselves, instead of stacking them up against the tree-level piece, and estimate the cutoff scale as the one at which they become of the same order. Indeed, it can be easily recognized that both corrections get comparable to each other at *lower* energies, i.e., when $\sqrt{s} \sim \sqrt{t} \sim M_t$. We generalize this result to an infinite tower of operators of the schematic form $D^{2\ell}\mathcal{R}^4$ in Section 2.2.1, hence showing that an infinite number of corrections become of the same order precisely at this scale. Thus, the natural conclusion would be that the gravitational sector of the $d$-dimensional EFT breaks down at the D0-brane scale, due to the non-convergence of the series of *field-theoretic* contributions. As argued in Section 1.1, this kind of breaking is not expected to be associated to the limit of validity of the (local) EFT framework, as opposed to the one that occurs around $\Lambda_{\mathrm{QG}}$. Let us remark that, as further explained in Section 2.2.3, one can see this very explicitly within the present setup by resumming the tower of massive threshold corrections and upgrading the original EFT to a new local field-theory, namely eleven-dimensional supergravity.

## 2.2 Decompactification limits and the EFT breaking scale

In this section, we focus on decompactification limits, which from the bottom-up are characterized by satisfying $M_t \ll \Lambda_{\mathrm{QG}}$. Above the energy scale $M_t$, the theory can be UV-completed to a new EFT in more dimensions. This suggests that (at least some of) the field-theoretic contributions in the double EFT expansion —that are sensitive precisely to this scale— should be exactly computable by considering $D$-dimensional Einstein gravity and compactifying down to $d$ dimensions. In contrast, the quantum-gravitational terms are not expected to be accessible via simple dimensional reduction, as they truly encode quantum gravitational effects.[14] Thus,

---

[14]See, however, [48, 139, 142–157] for recent discussions on the possibility to obtain these and other terms in the effective action (including those at the two-derivative level) in the context of the Emergence Proposal [7, 14, 19, 43, 44, 158].

in what follows we focus on the former field-theoretic contributions. We will say a few words about the presence of quantum-gravitational corrections at the end of the section.

In order to test this idea, let us focus on toroidal compactifications of maximal supergravity theories. In particular, we consider contributions to operators of the form $D^{2\ell}\mathcal{R}^k$ —i.e., of dimension $n = 2(\ell + k)$— appearing in the one-loop amplitude involving $k$ external gravitons and Kaluza-Klein particles running in the loop. Interestingly, we will find that the expectation above is confirmed whenever the field-theoretic contribution dominates over the quantum-gravitational one, which occurs when the operator in question is classically irrelevant in the parent $D$-dimensional theory. In contrast, whenever $D^{2\ell}\mathcal{R}^k$ is classically relevant (or marginal) in $D$ dimensions, the computation becomes UV divergent. Precisely in this case, the double EFT expansion suggests that the Wilson coefficient should instead be dominated by the quantum-gravitational contribution, as was the case for 10d Type IIA.

More ambitiously, we want to argue for the existence of an infinite number of operators of the field-theoretic form in the case of decompactification limits. The strategy we are going to employ here is to consider the maximally supersymmetric setup —where one expects the maximum number of cancellations— and show that even in this case the presence of a tower of states with a characteristic mass scale $M_t$ produces a contribution suppressed by $M_t^{n-d}$ for the corresponding dimension-$n$ operator in the effective action (cf. eq. (2)).

To this end, we recall the general form of the one-loop $k$-graviton amplitude in maximal $D$-dimensional supergravity, with $D \leq 11$, compactified on a $p$-torus down to $d = D - p$ dimensions. We will make frequent use of the resulting expression in upcoming subsections. Thus, using the worldline formalism [100, 102], the amplitude reads as

$$\mathcal{A}_k^{(p)} = \frac{1}{\mathcal{V}_{\mathbf{T}^p}} \int_0^\infty \frac{d\tau}{\tau} \int d^{D-p}p \sum_{\boldsymbol{n}\in\mathbb{Z}^p} e^{-\tau(p^2 + G^{(p)IJ}n_I n_J)} \text{Tr}\left\langle \prod_{r=1}^k \left(\int_0^\tau dt^{(r)} V^{(r)}(t^{(r)})\right)\right\rangle, \quad (15)$$

where $G_{IJ}^{(p)}$ is the torus metric (and we denote its volume by $\mathcal{V}_{\mathbf{T}^p} = \det^{1/2} G_{IJ}^{(p)}$), $\tau$ stands for the Schwinger proper time parameter, $p_\mu$ is the $d = (D-p)$-dimensional loop momentum along the non-compact directions, and the sum over $n_I$ represents the contribution of KK momenta. The $k$ external (on-shell) gravitons are implemented by the vertex operators, $V^{(r)}$,[15] inserted at $t^{(r)}$, and the trace is taken over bosonic and fermionic fields living on the worldline theory. Moreover, the particular class of contributions we are interested in here are precisely those for which the fermionic trace is saturated by the zero modes and are thus $\frac{1}{2}$-BPS, but we briefly comment on other contributions below. Hence, for such operators, the latter reduces to a simple overall kinematic factor, $\hat{K}$.

### 2.2.1 The $D^{2\ell}\mathcal{R}^4$ operators from supersymmetric loops

Let us begin by focusing on four-point amplitudes. With $k = 4$ external gravitons, the first non-vanishing contribution to this one-loop amplitude includes eight powers of the external momenta. This leads to the well-known $t_8 t_8 \mathcal{R}^4$ term in the effective action, which, for $p = 1$,

---

[15]The graviton vertex operator can be found in [102, Section 3], and it reads $V_g(t) = U_g e^{ik\cdot X}$, with $U_g$ a certain contraction involving polarization tensors, worldline fields, and external momenta. When the fermionic trace is saturated by the zero modes coming from $U_g$, the contribution associated to the exponential in $V_g$ gives rise to

$$\left\langle \prod_i e^{-ik^{(i)}X(t^{(i)})} \right\rangle = e^{-\sum_{i\neq j} k^{(i)}k^{(j)}G_B(t^{(i)}, t^{(j)})}, \quad (16)$$

with $G_B(t^{(i)}, t^{(j)}) = \frac{|t^{(i)}-t^{(j)}|}{2} + \frac{(t^{(i)}-t^{(j)})^2}{2\tau}$ the 1d (bosonic) Green's function. In the four graviton amplitude, the $U_g$ part of the vertex operators combine so as to introduce eight powers of external momenta, thus reproducing the linearized structure of $t_8 t_8 \mathcal{R}^4$. Similarly, the extra $\ell$ momentum pairs that appear at $\ell$-th order in (16) contribute to terms of the form $D^{2\ell}\mathcal{R}^4$ in the Wilsonian effective action [102].

gives precisely (12a), as can be readily identified from the kinematic factor in (9). The latter is the first of an infinite tower of contributions that arise upon expanding the exponential function that is introduced by the vertex insertions $V^{(r)}(t^{(r)})$ [100, 102]. Consequently, the $\ell$-th term in the momentum expansion includes the contribution with $2\ell$ extra powers of the external momenta, which multiply the (linearized) $t_8 t_8 \mathcal{R}^4$ structure. Thus, it is associated with local operators of the form $D^{2\ell}\mathcal{R}^4$ in the following way

$$\mathcal{A}_{4,2\ell}^{(p)} \simeq \frac{\hat{K}^{(2\ell)}}{\mathcal{V}_{\mathbf{T}^p}\,\ell!} \sum_{\boldsymbol{n}\in\mathbb{Z}^p} \int_0^\infty \mathrm{d}\tau \ \tau^{3-\frac{d}{2}+\ell}\ e^{-\tau\,G^{(p)IJ}n_I n_J}\,, \tag{17}$$

where we have explicitly integrated over the $d$-dimensional loop momenta and $\hat{K}^{(2\ell)}$ now includes the corresponding kinematic factor as well as other numerical coefficients that we do not show here explicitly. The additional powers of $\ell$ come from integrating over the positions (in proper time), $t^{(r)}$, of the four vertex insertions, for which each pair of external momenta introduces an extra factor of $\tau$.

We focus first on the ultra-violet behavior of (17), encoded in the $\tau \to 0$ integration domain. The integral over $\tau$ is convergent for $n = 8+2\ell > d$, such that upon making the change of variables $T = \tau\,G^{(p)IJ}n_I n_J$ and performing the integral over $T$, one gets

$$\mathcal{A}_{4,2\ell}^{(p)} \simeq \frac{\hat{K}^{(2\ell)}}{\mathcal{V}_{\mathbf{T}^p}\,\ell!} \Gamma\left(\frac{8+2\ell-d}{2}\right) \sum_{\boldsymbol{n}\in\mathbb{Z}^p\backslash\mathbf{0}} \left(G^{(p)IJ}n_I n_J\right)^{-\frac{8+2\ell-d}{2}}\,. \tag{18}$$

Here we used the definition $\Gamma(a) = \int_0^\infty \mathrm{d}x\, x^{a-1}e^{-x}$. Note that we are excluding by hand the $\boldsymbol{n} = 0$ contribution, which gives rise to an IR divergent piece, i.e., as $\tau \to \infty$. This corresponds to the non-analytic thresholds in the $d$-dimensional theory, arising from the massless sector running in the loop. We do not keep track of that piece in the reminder of the present discussion, as it can already be reproduced by the two-derivative EFT action. We will elaborate further on similar contributions in Section 2.2.3.

There is yet another source of possible divergences in the amplitude, namely the explicit sum over the lattice of Kaluza-Klein momenta. This computation yields an Epstein $\zeta$-function

$$\zeta\left(G^{(p)IJ}, s\right) = \sum_{\boldsymbol{n}\in\mathbb{Z}^p\backslash\mathbf{0}} \left(G^{(p)IJ}n_I n_J\right)^{-s}\,, \tag{19}$$

which converges absolutely for $\mathrm{Re}\,s > p/2$ [159–162]. Hence, the contribution to the amplitude that we are studying is UV finite if and only if $n = 2\ell + 8 > d + p$, namely whenever the operator $D^{2\ell}\mathcal{R}^4$ is irrelevant in the decompactified theory [49].[16] In order to extract the scale that suppresses the operator in the EFT, let us express everything in terms of the unimodular metric, $\tilde{G}_{IJ}^{(p)} = G_{IJ}^{(p)}\mathcal{V}_{\mathbf{T}^p}^{-2/p}$ and identify the KK scale as $M_{\mathrm{t}} = \mathcal{V}_{\mathbf{T}^p}^{-1/p}$. We obtain

$$\mathcal{A}_{4,2\ell}^{(p)} \simeq \frac{\hat{K}^{(2\ell)}}{\mathcal{V}_{\mathbf{T}^p}\,\ell!} \Gamma\left(\frac{8+2\ell-d}{2}\right) \zeta\left(\tilde{G}^{(p)IJ}, \frac{8+2\ell-d}{2}\right) \frac{1}{M_{\mathrm{t}}^{2\ell+8-d}}\,. \tag{21}$$

The term $D^{2\ell}\mathcal{R}^4$, of dimension $n = 2\ell + 8$ in the EFT, thus contains a UV convergent contribution suppressed by $M_{\mathrm{t}}^{n-d}$. This is moreover accompanied by the kinematic factor, $\hat{K}^{(2\ell)}$, the

---

[16]Equivalently, the ultra-violet convergence of the amplitude can be analyzed by performing a Poisson resummation, which trades the sum over the momentum lattice $\boldsymbol{n} \in \mathbb{Z}^p$ with an analogous one along the dual winding charges, $\boldsymbol{\omega} \in \mathbb{Z}^p$. This yields

$$\mathcal{A}_{4,2\ell}^{(p)} \simeq \frac{\hat{K}^{(2\ell)}}{\mathcal{V}_{\mathbf{T}^p}\,\ell!} \pi^{p/2}\sqrt{\det G_{IJ}^{(p)}} \sum_{\boldsymbol{\omega}} \int_0^\infty \mathrm{d}\tilde{\tau}\ \tilde{\tau}^{\frac{d+p-2\ell-10}{2}}\ e^{-\tilde{\tau}\,G_{IJ}^{(p)}\omega^I\omega^J}\,, \tag{20}$$

with $\tilde{\tau} = \tau^{-1}$ and $\det G_{IJ}^{(p)} = (\mathcal{V}_{\mathbf{T}^p})^2$. Therefore, the finiteness of the amplitude at short-distances (i.e., when $\tilde{\tau} \to \infty$) is dictated by the integrand in the zero-winding sector, which converges if and only if $n = 2\ell + 8 > d + p$.

overall volume that accounts for the change from higher- to lower-dimensional Planck units, and two numerical coefficients determined by the $\Gamma$- and Epstein $\zeta$-functions. Note that the latter includes the dependence on all the internal moduli associated to the $p$-dimensional torus except for the volume, which has already been extracted upon switching to the unimodular metric.

We note that the analysis presented so far is general in the sense that, whenever we probe a limit in which $p$-dimensions become large, we can always extract such scale. Specifically, if the $p$ dimensions decompactify at the same rate asymptotically, our considerations remain valid, and the differences between the distinct KK-mode masses are captured only by $\mathcal{O}(1)$ factors entering the $\zeta$-function. However, if several dimensions decompactify at different rates, the analysis can also be performed sequentially. In this case, one first identifies $d$ as the original dimensionality of spacetime and $p' \leq p$ as the number of dimensions that grow at the fastest rate. Then, $d' = d + p'$ is redefined as the resulting non-compact spacetime dimensionality after the first step, and $p'' \leq p - p'$ as the dimensions decompactifying at the next-fastest rate, and so on. The homogeneous and sequential decompactifications turn out to be the two limiting cases in setups with decompactification at different rates, which introduces multiple field-theoretical scales. This fits naturally within the logic of the double EFT expansion and is illustrated in a particular setup in which extra dimensions decompactify at different rates in Section 2.3.2.

In order to gain some intuition, let us recall that in the simplest possible case of a circle of size $R$ (in $D$-dimensional Planck units), the amplitude reduces to [49]

$$\mathcal{A}_{4,2\ell}^{(\mathbf{S}^1)} \simeq \frac{2\hat{K}^{(2\ell)}}{R\,\ell!}\,\Gamma\left(\frac{8+2\ell-d}{2}\right)\zeta(8+2\ell-d)R^{2\ell+8-d}\,, \tag{22}$$

with $R = M_{\mathrm{t}}^{-1}$ and $\zeta(a)$ the Riemann zeta-function. For a two-dimensional torus the story gets more interesting, since the four-point graviton amplitude now reads[17]

$$\mathcal{A}_{4,2\ell}^{(\mathbf{T}^2)} \simeq \frac{\hat{K}^{(2\ell)}}{\mathcal{V}_{\mathbf{T}^2}\,\ell!}\,\Gamma\left(\frac{8+2\ell-d}{2}\right)E_{\frac{8+2\ell-d}{2}}^{sl(2)}(\tau,\bar{\tau})\,(\mathcal{V}_{\mathbf{T}^2})^{\frac{8+2\ell-d}{2}}\,, \tag{23}$$

where we have used the definition of the non-holomorphic Eisenstein series $E_s^{sl_2}(\tau,\bar{\tau})$ (see, e.g., [49, Appendix A]), as well as $\mathcal{V}_{\mathbf{T}^2} = M_{\mathrm{t}}^{-2}$. In Section 2.3.2, we will return to this second example, which is particularly useful for illustrating how some terms that do not naively take the form of (2) still fit within the double EFT expansion proposal. In particular, these extra contributions will be naturally regarded as field-theoretic terms inherited from a higher-dimensional theory after compactification (see discussion below eq. (2)).

At this point, let us note that for a given generic operator $D^{2\ell}\mathcal{R}^4$, not only this $\frac{1}{2}$-BPS contribution is expected to arise, but also further corrections due to higher loops. However, these corrections should be subleading along the decompactification limit, as suggested by, e.g., Type IIA/M-theory duality [106].[18] In addition, one might worry about further contributions to this operator at the one-loop level beyond those considered herein, which could ultimately result in a vanishing total outcome. This scenario would arise only if non-zero fermionic modes in

---

[17]Note that this power of $\mathcal{V}_{\mathbf{T}^2}$ does not coincide with the one appearing in refs. [49, 100–108]. This difference can be accounted for by the fact that we use here a dimensionful volume, whereas in the aforementioned works $\mathcal{V}_2$ represents the volume measured in higher-dimensional Planck units. Hence, when using these units indeed, one introduces by hand an explicit dependence on $M_{\mathrm{Pl},\,d+2}^{d-2\ell-8}$ which, after re-expressed in terms of the lower-dimensional Planck mass, $M_{\mathrm{Pl},\,d}^{d-2} = M_{\mathrm{Pl},\,d+2}^{d-2}\mathcal{V}_2$, reproduces the usual power found in [49, 100–108].

[18]In fact, the terms arising from higher-loop corrections in the decompactification limit need not be subleading when probing small compactification volumes, as is evident when reducing on $\mathbf{S}^1$. In this case, the small-circle limit reproduces the Type IIA perturbative string computation, where the leading term (in the genus expansion) can often be traced back to higher-loop contributions in the 11d theory.

the worldline theory were to saturate the fermionic trace (see footnote 15), thereby producing the same tensor structure (i.e., $t_8 t_8 \mathcal{R}^4$), along with identical dependence on the internal volume and external momenta. However, this appears highly unlikely due to the complexity of the fermionic Green functions [102]. Furthermore, note that this would entail a cancellation between maximally BPS contributions and non-BPS ones, which seems unnatural. Overall, any potential extra contributions are not expected to alter the fact that a term of the form discussed above appears for an infinite number of gravitational operators in the EFT and provides the leading contribution to these operators. This is the key point for our argument and the logic behind the double EFT expansion.

To summarize, in maximally supersymmetric setups, where it is natural to expect the largest number of cancellations, there is yet a formally infinite set of higher-derivative operators of the form $D^{2\ell} \mathcal{R}^4$ (with $n = 2\ell + 8 > d + p$) which receive threshold contributions —from the tower of KK modes along the $p$ compact dimensions— that are suppressed by $M_t^{n-d}$ in the low-energy EFT. The latter can be detected through four-point graviton scattering, and moreover produce terms that naively become comparable to the two-derivative, tree-level amplitude at a scale $M_t^{\frac{n-d}{n-2}} M_{\mathrm{Pl},d}^{\frac{d-2}{n-2}} \ll \Lambda_{\mathrm{QG}}$, which is parametrically lower than $\Lambda_{\mathrm{QG}}$ and arbitrarily close to $M_t$ (for $n \gg d$). This indicates the breakdown of the original $d$-dimensional EFT at a scale well below the the quantum gravity cutoff. Let us remark that this breaking can be easily cured by going to the higher-dimensional, local EFT, and is not directly associated to the limit of validity of the local EFT framework, as opposed to the one occurring at energies close to $\Lambda_{\mathrm{QG}}$, which is due to intrinsically quantum gravity effects.

### 2.2.2 General argument from dimensional analysis

Having considered the one-loop contributions to the $D^{2\ell} \mathcal{R}^4$ terms relevant for four-point graviton amplitudes, we now focus on operators of the schematic form $D^{2\ell} \mathcal{R}^k$ with $k > 4$, which thus have classical dimension $n = 2(\ell + k)$. As opposed to the analysis in previous sections, which is based on well-established computations in highly supersymmetric setups, our arguments in the following will build on less rigorous calculations, but are nevertheless strongly supported by dimensional analysis, which is known to be particularly robust for UV convergent operators. Hence, by analogy with eq. (17), we expect the one-loop $k$-graviton amplitude, in the presence of a tower of KK-modes associated with $p$ compact dimensions, to produce a threshold contribution to operators in the gravitational EFT. These operators take the schematic form $D^{2\ell} \mathcal{R}^k$ and contribute to the ampiltide as follows

$$\mathcal{A}_{k,2\ell}^{(p)} \simeq \frac{\hat{K}^{(k,2\ell)}}{\mathcal{V}_{\mathbf{T}^p} \, \ell!} \sum_{\{n_I\}} \int_0^\infty \mathrm{d}\tau \; \tau^{k+\ell-1-\frac{d}{2}} \; e^{-\tau \, G^{(p)IJ} n_I n_J} \, . \tag{24}$$

Once again, the integral over the Schwinger parameter converges in the ultra-violet regime (i.e., for $\tau \to 0$) if and only if $n = 2(\ell + k) > d$, yielding in particular

$$\mathcal{A}_{k,2\ell}^{(p)} \simeq \frac{\hat{K}^{(k,2\ell)}}{\mathcal{V}_{\mathbf{T}^p} \, \ell!} \Gamma\left(\frac{n-d}{2}\right) \sum_{\mathbf{n} \in \mathbb{Z}^p \setminus \mathbf{0}} \left(G^{(p)IJ} n_I n_J\right)^{-\frac{n-d}{2}} \, . \tag{25}$$

There is, however, a potential second source of divergence in the UV associated to the sum over Kaluza-Klein momenta $\mathbf{n}$, which takes the form of an Epstein $\zeta$-function, hence converging whenever $n = 2(k + \ell) > d + p$. Thus, for irrelevant operators in the decompactified theory, the loop contribution due to the KK-tower is convergent and moreover gives

$$\mathcal{A}_{k,2\ell}^{(p)} \simeq \frac{\hat{K}^{(k,2\ell)}}{\mathcal{V}_{\mathbf{T}^p} \, \ell!} \Gamma\left(\frac{n-d}{2}\right) \zeta\left(\tilde{G}^{(p)IJ}, \frac{n-d}{2}\right) \frac{1}{M_t^{n-d}} \, , \tag{26}$$

where we have defined again the unimodular metric $\tilde{G}_{IJ}^{(p)} = G_{IJ}^{(p)} \mathcal{V}_{\mathbf{T}^p}^{-2/p}$ and identified the tower scale as that associated to the overall volume modulus, i.e., $M_{\mathrm{t}} = \mathcal{V}_{\mathbf{T}^p}^{-1/p}$. Let us remark though that, despite the fact that this analysis is not as thorough as the one presented in the previous section, it still provides exactly what one would expect from a series of threshold contributions associated to massive KK-modes. Furthermore, given that the computation is UV convergent and thus need not be regularized, it is sensible to obtain a result that only depends on the scale associated to the tower, $M_{\mathrm{t}}$.

### 2.2.3 The EFT breaking scale and resummation of field-theoretic terms

Consistent with the double EFT expansion, we have argued that the one-loop contribution to the graviton $k$-point function should include a term suppressed by the scale $M_{\mathrm{t}}^{n-d}$ for an *infinite* number of operators of the form $D^{2\ell} \mathcal{R}^k$, where $n = 2(k+\ell) > d+p$.[19] Consequently, when seeking for the regime of validity of the gravitational EFT in the presence of such infinite tower of operators of *field-theoretic* nature, there are two kinds of approaches or strategies that one can take.

First of all, let us compare each of the higher-derivative terms with the tree-level contribution arising from the two-derivative action. The scale at which the two become comparable is different for each such operator, since it depends on its classical dimension $n$, and yields[20]

$$\sqrt{|s^i|} \sim M_{\mathrm{t}}^{\frac{n-d}{n-2}} M_{\mathrm{Pl},d}^{\frac{d-2}{n-2}}. \tag{27}$$

Hence, we see that for any $n > d+p$ this is parametrically below the quantum gravity scale, since one finds that $\sqrt{|s^i|}/\Lambda_{\mathrm{QG}} \sim (M_{\mathrm{t}}/M_{\mathrm{Pl},d})^{\frac{(d-2)(n-d-p)}{(n-2)(d+p+2)}}$ tends to zero in the *asymptotic regime* $M_{\mathrm{t}} \ll M_{\mathrm{Pl},d}$. Furthermore, for arbitrarily large $n$, (27) can be seen to tend to $M_{\mathrm{t}}$. This, together with the fact that each contribution only provides an upper bound for the EFT cutoff, points towards $M_{\mathrm{t}}$ as being indeed the scale at which the EFT breaks down.

On the other hand, an alternative approach would consist in not only comparing each of these terms with the classical EH contribution, but also among themselves. Upon doing so, it becomes manifest that at scales around

$$\sqrt{|s^i|} \sim M_{\mathrm{t}}, \tag{28}$$

the different corrections within the *field-theoretic* expansion become of the same order. Since we have shown these to be present for an infinite number of operators with $n > d+p$, it follows immediately that the breaking of the EFT takes place at that scale. Hence, at energies close to $M_{\mathrm{t}}$, one should resum the full series of *field-theoretic* terms. As we discuss next in a simple yet illustrative example, together with the massless thresholds already present in the $d$-dimensional theory, these contributions would add up to the corresponding massless thresholds in the $(d+p)$-dimensional theory. Furthermore, if the corresponding operators additionally include a *quantum-gravitational* contribution, even if subleading, then they must also be present in the decompactified theory [50].

---

[19]Notice that for $k=4$ we have presented robust evidence supporting the presence of such terms, whereas in the case where $k>4$ our argument relies on a naive generalization of the previous calculations combined with dimensional analysis. Still, let us stress again that the infinite tower of operators $D^{2\ell}\mathcal{R}^4$ is sufficient to show the breaking of the EFT at the scale $M_{\mathrm{t}} \leq \Lambda_{\mathrm{QG}}$, and we have included the discussion for the tower of operators of the form $D^{2\ell}\mathcal{R}^k$ with $k>4$ for completeness.

[20]This set of cutoffs, which arise upon looking for energy scales at which each higher-dimensional operator becomes of the order of the tree-level, are very reminiscent of the ones recently revisited in the context of non-gravitational field theories in [163]. Those were dubbed *mirage cutoffs*, since they overestimate the true regime of validity of the underlying effective theory. Notice that, therein, the mirage cutoffs are associated to high-multiplicity processes, whereas here these can be detected from different higher-derivative terms in the four-point graviton amplitude.

All in all, the amplitudes perspective discussed herein clearly supports the general picture presented in Section 1.1, since we can unequivocally distinguish the scale $M_\mathrm{t}$ as the one at which the EFT breaks down, as detected from graviton scattering experiments. Still, this kind of breaking can generally be cured by the inclusion of the extra states in the tower, whenever they lie well below the quantum gravity cutoff (i.e., $M_\mathrm{t} \ll \Lambda_\mathrm{QG}$), into a new local EFT in more dimensions, whereas the *quantum-gravitational* contributions remain present in the UV uplifted EFT and kick in at energies around $\Lambda_\mathrm{QG}$.

**Resummation of massive thresholds**

At this point, the reader might be wondering how to reconcile the following two facts: On the one hand, we argued in Section 1.1 that the field-theoretic contributions do not actually compete with the classical EH term. On the other hand, following the analysis above, the field-theoretic correction to a dimension-$n$ operator becomes of the same order as the EH term at energies given by (27). Therefore, it would seem that this kind of contributions with arbitrarily large $n$ naively compete with the EH term at energies close to and above $M_\mathrm{t}$.

The resolution to this apparent puzzle lies on the fact that, up to now, we have focused only on corrections to the $k$-point graviton functions that can be encoded into local operators within the Wilsonian effective action, and thus we have ignored certain additional terms that account instead for the massless threshold behavior. These objects can be more generally written as

$$\mathcal{A}_k(s,t) = \mathcal{A}_k^\mathrm{an}(s,t) + \mathcal{A}_k^\mathrm{non-an}(s,t), \tag{29}$$

where the first piece corresponds to the scattering amplitude shown in eqs. (8) and (11), for the particular case $k = 4$. On the other hand, $\mathcal{A}_k^\mathrm{non-an}(s,t)$ is in fact non-analytic in the Mandelstam variables, and as such encapsulates the aforementioned extra terms.[21] Moreover, their structure can be shown to be subject to very strong constraints coming from (perturbative) unitarity, since they account for the discontinuities in the $k$-point amplitude induced by the threshold singularities (see, e.g., [164–167] and references therein). For the particular case of $2 \to 2$ graviton scattering in ten spacetime dimensions, and focusing for the moment on the $s$-channel contribution, the discontinuity ends up having the following approximate form [106]

$$\mathrm{Disc}\,\mathcal{A}_4(s,t) \propto \int \mathrm{d}^{10}p\, \mathcal{A}_4\left(k^{(1)},k^{(2)},p,-p-k^{(1)}-k^{(2)}\right) \mathcal{A}_4^\dagger\left(k^{(3)},k^{(4)},-p,p+k^{(3)}+k^{(4)}\right)$$
$$\times \delta\left(p^2\right) \delta\left(\left(p+k^{(1)}+k^{(2)}\right)^2\right) \theta(p_0)\,\theta\left(\left(p+k^{(1)}+k^{(2)}\right)_0\right). \tag{30}$$

Therefore, assuming now a low-energy expansion of the (analytic part of the) amplitude as displayed in (11), one can then see recursively from (30) how the structure of the massless thresholds is restricted, and in some cases, even uniquely fixed. For instance, in the case of 10d Type IIA supergravity one finds (in the $s$-channel) [103]

$$\mathcal{A}_4^\mathrm{non-an}(s,t) = \frac{\hat{K}\,\ell_{\mathrm{Pl},10}^8}{32\pi^2}\left(\ell_{\mathrm{Pl},10}^2\, s \ln\left(-\ell_{\mathrm{Pl},10}^2\,\frac{s}{4\pi^2}\right) + \mathcal{O}\left(\ell_{\mathrm{Pl},10}^8\right)\right), \tag{31}$$

whereas the first few contributions to the analytic part of the amplitude are shown in eqs. (8)-(12). In fact, the above logarithmic correction can be recognized to be nothing but the familiar massless supergravity threshold in ten dimensions.

Furthermore, as one may explicitly verify (see Appendix A for details), the field-theoretic corrections that seemingly dominate the amplitude in the strong coupling regime in fact resum

---

[21]The non-analytic behavior can be traced back to the long wave-length modes associated to massless degrees of freedom and can be actually included within the 1PI quantum effective action rather than its Wilsonian analogue.

—together with the first massless threshold displayed in (31)— into a formal series that gives rise to the 11d massless threshold in M-theory. Indeed, when the (dual) radius has infinite size the latter behaves as $s^{3/2}$ (cf. eq. (A.13)), whilst for finite values of the radius it admits instead a Taylor-like expansion which converges when $s M_{\text{D0}}^{-2} \ll 1$. Hence, we conclude that the apparent competition of the massive thresholds appearing in higher-curvature operators of the form $D^{2\ell} \mathcal{R}^4$ is an artifact of extending the behavior of the non-analytic series expansion beyond its actual regime of validity, and this in fact can be cured by modifying the original EFT at scales of the order $M_{\text{t}} = M_{\text{D0}}$. Once this is done, it is clear that the resummed expression of the *field-theoretic* contribution to the four-point amplitude does not really compete with the EH term.

**Quantum-gravitational terms and interplay with field-theoretic ones**

As mentioned in the introduction of this section, so far we have ignored the quantum-gravitational terms in the decompactification limits. Nevertheless, it turns out that we can say something about their existence from what we learned in the previous discussion.

As we just saw, field-theoretic terms are resummed to UV-complete the EFT above $M_{\text{t}}$, yielding the higher-dimensional massless threshold in the amplitude. This way, these contributions to higher-derivative operators disappear from the higher-dimensional effective action, as this effect is reproduced by the two-derivative part at one-loop (or higher) order. On the other hand, quantum-gravitational terms should still be encoded within the higher-dimensional effective action [50]. Turning this logic around, the lower-dimensional EFT features these quantum-gravitational corrections only if they were already present in the higher-dimensional theory. Therefore, whenever we have $M_{\text{t}} \ll \Lambda_{\text{QG}}$, we can complete the theory at energies above $M_{\text{t}}$ until we end up with an EFT satisfying $M_{\text{t}} \sim \Lambda_{\text{QG}}$. Once we reach this point, we should argue that the theory contains the quantum-gravitational piece of the double EFT expansion [53]. This will be the subject of Section 2.3.

However, before proceeding any further, and also connecting with the general discussion in Section 1.1, let us consider the case where a given dimension-$n$ operator includes both the field-theoretic and the quantum-gravitational terms. That is, in the presence of a KK-tower associated to, e.g., $p$ compact dimensions, we assume a contribution to the amplitude —when normalized with respect to the tree-level, two-derivative piece— of the form (cf. eqs. (8)-(11) for the four-point graviton example)

$$\ell_{\text{Pl},d}^{2(\ell+k)-2} f_{D^{2\ell} \mathcal{R}^k} = \tilde{f}^{(k,2\ell)}(s^i) \left( \frac{a_n}{\Lambda_{\text{QG}}^{n-2}} + \frac{b_n}{M_{\text{Pl},d}^{d-2} M_{\text{t}}^{n-d}} + \cdots \right), \qquad (32)$$

where $\tilde{f}^{(k,2\ell)}(s^i)$ represents a homogeneous function of degree $(n-2)/2$ of the Mandelstam variables, $s^i$, corresponding to the kinematic structure associated to the operator $D^{2\ell} \mathcal{R}^k$.

Given that in the *asymptotic regime* these scales $\Lambda_{\text{QG}}$ and $M_{\text{t}}$ are oftentimes correlated, it is natural to ask which of these terms gives the leading contribution for each operator. Indeed, the $(d+p)$-dimensional theory that we are compactifying in the first place should satisfy $\Lambda_{\text{QG}} \lesssim M_{\text{Pl},d+p}$. When written in terms of $M_{\text{t}}$, and using $d$-dimensional Planck units, this yields the bound (7), that we recall here for convenience

$$\Lambda_{\text{QG}} \lesssim M_{\text{t}}^{\frac{p}{d-2+p}}. \qquad (33)$$

As discussed around therein, this implies that the *quantum-gravitational* term dominates if $n < d + p$ (i.e., for relevant operators in the decompactified theory).

It is equally interesting to consider the case where $\Lambda_{\text{QG}} \sim M_{\text{Pl},d+p}$, such that the $(d+p)$-dimensional theory does not contain any additional light scales, and the bound above is saturated. In this scenario, one can readily see that the *field-theoretic* contribution will dominate

over the *quantum-gravitational* one for $n > p + d$ (i.e., for irrelevant operators in the decompactified theory), whereas both present the same asymptotic scaling behavior if $n = p+d$ (i.e., for marginal operators in $d + p$ dimensions). Thus, for the infinite tower of operators that we have shown to include a *field-theoretic* contribution, the latter turns out to be the most relevant one in the absence of any further light scale in the $(d + p)$-dimensional theory. Interestingly, this also corresponds to the case for which the one-loop computation discussed in this section converges. In other words, whenever the quantum-gravitational contribution dominates, the higher-dimensional computation becomes UV-sensitive, as it should.

As emphasized in Section 1.1, it is important to remark that, even though $M_t \lesssim \Lambda_{QG}$, the different powers appearing in each term allow for the quantum gravity piece to provide the leading contribution for a finite number of operators along decompactification limits, i.e., those with $n \leq d + p$. For instance, this precisely happens for $t_8 t_8 \mathcal{R}^4$ in ten dimensions (cf. Section 2.1). This highlights the importance of looking at the whole tower of operators, namely including also the irrelevant ones, as opposed to checking only the first few leading corrections. Otherwise, one could be tempted to identify the scale at which the gravitational sector of the EFT is breaking as the one suppressing the (leading contribution to the) first correction, which is typically given by $\Lambda_{QG}$.

## 2.3 Weakly-coupled string limits and extra terms in the EFT

Shifting focus, in this section we concentrate on limits where $\Lambda_{QG} \sim M_t$. From the top-down, all known examples of this kind correspond to weakly-coupled and tensionless string limits, as encoded in the Emergent String Conjecture [25]. Thus, we will consider next the genus expansion within string theory for the four-point graviton amplitude. In particular, we focus on 10d Type IIA string theory. This setup will suffice to illustrate the main points we want to convey, namely that the quantum-gravitational and field-theoretic terms in the double EFT expansion correspond to the genus-zero and genus-one contributions, respectively. Let us remark that, even though we will consider a theory naturally living in ten spacetime dimensions, we recall from Section 2.2.3 that the quantum-gravitational expansion in lower-dimensional theories obtained by compactification is inherited from the parent, ten-dimensional one. On the other hand, field-theoretic contributions in lower dimensions arise from threshold corrections, as discussed at length in Section 2.2.

### 2.3.1 The double EFT expansion in string perturbation theory

In Section 2.1.2, we showed that the quantum-gravitational term for the $\mathcal{R}^4$, $D^4 \mathcal{R}^4$ and $D^6 \mathcal{R}^4$ operators come from the genus-zero contribution to the four-point graviton amplitude in 10d Type IIA weakly-coupled string theory. This conclusion can be easily extended to an infinite number of similar higher-dimensional terms by considering the four-point tree-level string scattering amplitude in the perturbative genus expansion. In the string frame, the latter reads as $\mathcal{A}_4^{\text{tree}}(s,t) = -\hat{K} \ell_s^8 (4\pi g_s^2)^{-1} \mathcal{T}(s,t,u)$, with [168, 169]

$$
\begin{aligned}
\mathcal{T}(s,t,u) &= \frac{64}{\alpha'^3 stu} \frac{\Gamma\left(1 - \frac{\alpha'}{4}s\right)\Gamma\left(1 - \frac{\alpha'}{4}t\right)\Gamma\left(1 - \frac{\alpha'}{4}u\right)}{\Gamma\left(1 + \frac{\alpha'}{4}s\right)\Gamma\left(1 + \frac{\alpha'}{4}t\right)\Gamma\left(1 + \frac{\alpha'}{4}u\right)} \\
&= \frac{64}{\alpha'^3 stu} \exp\left[\sum_{k=1}^{\infty} \frac{2\zeta(2k+1)}{2k+1}\left(\frac{\alpha'}{4}\right)^{2k+1}\left(s^{2k+1} + t^{2k+1} + u^{2k+1}\right)\right].
\end{aligned}
\tag{34}
$$

To arrive at the second line, we have used repeatedly the series expansion

$$
\ln(\Gamma(1-x)) = \gamma x + \sum_{n=2}^{\infty} \zeta(n) x^n / n,
$$

where $\gamma$ denotes the Euler-Mascheroni constant. Notice that equation (34) precisely reproduces the first coefficients in (12a)-(12c).[22] Moreover, it captures the leading-order term for an arbitrarily high number of higher-derivative and higher-curvature operators of the form $D^{2\ell}\mathcal{R}^4$, each of them suppressed by $M_s^{(2\ell+8)-2}$ with respect to the Einstein-Hilbert term. Note that this indeed agrees with (3) and therefore tells us that from this point onward it becomes necessary to include the full tower of higher-spin excitations into the theory, since they start dominating the relevant physical observables such as the scattering amplitudes considered herein (cf. footnote 12).

This illustrates that, as befits the double EFT expansion, there is an infinite tower of higher-dimensional operators in the quantum-gravitational expansion (3) for Type IIA/B weakly coupled string theories. Let us also remark, in passing, that a similar behavior can be found in the tree-level closed-string amplitudes of heterotic, Type I and bosonic string theories (see, e.g., [170] and [59, Appendix B]).

Similarly, we showed in Section 2.1.2 that the $\mathcal{R}^4$, $D^4\mathcal{R}^4$ and $D^6\mathcal{R}^4$ operators feature field-theoretic terms, which arise from the genus-one contribution to the amplitude. In general, this genus-one correction to a dimension-$n$ operator is of order $(g_s)^0$, when measured in the string frame. Switching to the Einstein frame introduces an extra $g_s^{\frac{10-n}{4}}$ factor, such that upon using the relation between the string scale and the Planck mass in (13), it follows that this term is suppressed by $M_s^{10-n}$. This yields, as advertised, a field-theoretic contribution to an infinite tower of higher-dimensional operators.

### 2.3.2 Extra terms and the surprising power of the double EFT expansion

Let us now generalize the previous results to the full genus expansion of the four-point graviton amplitude in weakly-coupled Type IIA string theory. This will allow us to discuss the extra contributions encoded in the ellipsis in (6). Moreover, using the duality with M-theory in the strong coupling limit, we will also be able to comment about the presence of these additional terms along limits of the decompactification type.

From the weakly-coupled 10d Type IIA point of view, the genus expansion in string perturbation theory provides different contributions to each higher-dimensional term that scale with the corresponding power of $g_s$. In particular, an $h$-loop correction to a dimension $n = 2\ell + 8$ operator in 10d Planck units comes with a power of $g_s^{2h-2}$ from the genus counting, times an extra factor of $g_s^{2(d-n)/d-2}$ due to the relation between string and 10d Planck units. Furthermore, it was argued from M-theory/Type IIA duality that such $D^{2\ell}\mathcal{R}^4$ operators only receive non-vanishing contributions up to $h = \ell$ loops (for $\ell > 1$) [106].[23] Thus, we conclude that the part in the four-point graviton amplitude (8) arising from operators of the schematic form $D^{2\ell}\mathcal{R}^4$ can be compactly expressed as

$$\ell_{\text{Pl},10}^{2\ell+6} f_{D^{2\ell}\mathcal{R}^4} = \tilde{f}^{(2\ell)}(s,t,u)\left(\sum_{h=0}^{\ell}\frac{c_h}{M_{\text{Pl},10}^{2\ell+6}}\, g_s^{\frac{4h-\ell-3}{2}}\right), \tag{35}$$

where $\tilde{f}^{(2\ell)}(s,t,u)$ is a homogeneous function of degree $\ell + 3$ of the Mandelstam invariants.

For the $g_s \ll 1$ regime, $\Lambda_{\text{QG}}$ and $M_{\text{t}}$ are both set by the string mass (cf. eq. (13)). In such a limit —satisfying $M_{\text{t}} \sim \Lambda_{\text{QG}}$— the quantum-gravitational term always dominates over

---

[22]One can see this explicitly upon using the relation [103]

$$\left(\frac{\alpha'}{4}\right)^n (s^n + t^n + u^n) = n \sum_{2p+3q=n}\frac{(p+q-1)!}{p!q!}\left(\frac{(s^2+t^2+u^2)}{2}\right)^p\left(\frac{(s^3+t^3+u^3)}{2}\right)^q,$$

and subsequently expanding the exponential in (34).

[23]Note that this has been rigorously proven for $\ell < 6$ using the pure spinor formalism [134].

the field-theoretic one. As discussed in the previous section, these correspond to the tree-level and one-loop contributions in the genus expansion, respectively. The presence of additional corrections in (35) is due to higher-genus terms, which are subleading with respect to the *field-theoretic* and, in this case, also relative to the *quantum-gravitational* ones. This supports the general picture that, even though extra terms might be present, they will always be subleading with respect to the field-theoretic and/or quantum-gravitational piece whenever the Wilson coefficient does not vanish identically. Otherwise, we would find ourselves in a situation in which $c_0 = c_1 = 0$ (for some of the aforementioned higher-curvature operators) in string perturbation theory, but with non-vanishing higher-genus contributions. This seems extremely unlikely or fine-tuned and turns out to never be the case in known top-down examples (to the best of our knowledge).

Similar conclusions can be drawn for the decompactification limit to M-theory, namely when $g_s \gg 1$. In this regime, it is transparent from (35) that the dominant contribution —for a given operator $D^{2\ell}\mathcal{R}^4$— comes from the highest power of $g_s$. Such term corresponds to the largest possible value of $h$ contributing to the relevant Wilson coefficient, which is given by $h = \ell$. The latter has been argued to always come from the one-loop threshold of Kaluza-Klein modes in 11d M-theory compactified on $\mathbf{S}^1$ [106] (cf. Section 2.2). Furthermore, this correction can be seen to be suppressed precisely by $M_{\mathrm{Pl},10}^8 M_t^{2\ell-2}$, as befits the *field-theoretic* piece in the double EFT expansion (4), with the mass of the tower given by that of D0-brane states in Type IIA string theory, see eq. (14). From this observation, it is clear that any further contributions will be subleading with respect to the *field-theoretic* one. On the other hand, the *quantum-gravitational* piece, suppressed by $\Lambda_{\mathrm{QG}}^{2\ell+6}$, turns out to arise from the genus $h = \frac{\ell+3}{3}$ correction [106]. Since $h$ is an integer, one expects this term to be present only for $D^{2\ell}\mathcal{R}^4$ with $\ell \in 3\mathbb{Z}_{>0}$. This is consistent with known results regarding, e.g., $D^4\mathcal{R}^4$ and $D^6\mathcal{R}^4$ operators in 11d M-theory, as discussed in Section 2.1 (see also [100, 101, 104, 105, 123, 140, 141]). Notice that for sufficiently large $\ell$, there can be additional terms that dominate over the quantum-gravitational one. In that case, the double EFT expansion will capture the leading behavior only if the field-theoretic contribution is non-vanishing. However, as we argued in Section 2.2, this term does not cancel in maximal supergravity. We take this as evidence that this will not occur in less supersymmetric setups, unless the full Wilson coefficient vanishes (due to, for example, other symmetry reasons).

**Extra terms from different field-theoretic scales**

Before concluding this section, we briefly discuss a class of extra terms coming from the presence of different field-theoretic scales with a parametric hierarchy, and discuss how they nicely fit within the logic underpinning the double EFT expansion.

To this end, let us start by revisiting the $D^{2\ell}\mathcal{R}^4$ Wilson coefficient in toroidal compactifications of $d$-dimensional maximally supersymmetric theories, as presented in Section 2.2.1. Specifically, we focus on the $\mathbf{T}^2$-example (cf. (23)), which is particularly useful for illustrating what happens in the presence of two distinct field-theoretic UV cutoffs. The ratio of these scales, namely $M_1 = 1/R_1$ and $M_2 = 1/R_2$, appears in the amplitude through the complex structure of the torus, i.e., $\mathrm{Im}\,\tau = M_2/M_1$. Consequently, when $M_1 \sim M_2$, the $D^{2\ell}\mathcal{R}^4$ operator receives the expected field-theoretic contribution, (2). Conversely, for $M_1 \ll M_2$, the non-holomorphic Eisenstein series can be expanded as [49, Appendix A][24]

$$E_s^{sl(2)}(\tau,\bar{\tau}) = 2\zeta(2s)(\mathrm{Im}\,\tau)^s + 2\pi^{1/2}\frac{\Gamma(s-1/2)}{\Gamma(s)}\zeta(2s-1)(\mathrm{Im}\,\tau)^{1-s} + \cdots, \qquad (36)$$

---

[24]Notice that the expansion (36) naively spoils the $\mathbb{Z}_2$-symmetry exchanging $R_1 \leftrightarrow R_2$. This gauge invariance is, however, restored if one realizes that the amplitude (23) is preserved under the $SL(2,\mathbb{Z})$ group. Hence, the opposite behavior $M_2 \ll M_1$ indeed arises when taking the limit $\mathrm{Im}\,\tau \to 0$ instead, which exhibits the same dependence as in (37) with $M_1 \leftrightarrow M_2$.

where the ellipsis denotes exponentially suppressed corrections along the $\operatorname{Im}\tau \to \infty$ limit. Plugging this into (23), and taking into account that $\mathcal{V}_{\mathbf{T}^2} = R_1 R_2$, we obtain

$$
\begin{aligned}
\mathcal{A}_{4,2\ell}^{(\mathbf{T}^2)} &\simeq \frac{2\hat{K}^{(2\ell)}}{\mathcal{V}_{\mathbf{T}^2}\,\ell!}\,\Gamma\left(\frac{8+2\ell-d}{2}\right)\zeta(8+2\ell-d)\,\frac{1}{M_1^{8+2\ell-d}} \\
&\quad + \frac{2\hat{K}^{(2\ell)}}{\mathcal{V}_{\mathbf{T}^2}\,\ell!}\,\pi^{1/2}\Gamma\left(\frac{7+2\ell-d}{2}\right)\zeta(7+2\ell-d)\,R_1\frac{1}{M_2^{7+2\ell-d}} + \cdots .
\end{aligned}
\tag{37}
$$

As expected, the leading term precisely corresponds to the field-theoretic contribution due to the lowest KK scale $M_1$, such that it can be associated with a single extra dimension (cf. (22)). Interestingly, the second summand is not suppressed by $M_2^{8+2\ell-d}$, as one could have naively guessed. In fact, the asymptotic hierarchy $M_1 \ll M_2$ implies that the field-theoretic correction associated to $M_2$ is actually $(d+1)$-dimensional in origin. Therefore, we see that this second term in fact recovers (22) in $d+1$ dimensions. The extra factor of $R_1$ is responsible for changing from $d$- to $(d+1)$-dimensional Planck units. The physical interpretation is thus clear: upon re-integrating in the tower of states with masses of order $M_1$, the first term disappears from the $D^{2\ell}\mathcal{R}^4$ Wilson coefficient —it becomes part of the non-analytic structure of the two-derivative amplitude— while the second one is identified with the field-theoretic contribution controlled by $M_2 \sim M_{\mathrm{t}}$ that one expects to arise in the uplifted $(d+1)$-dimensional theory.

In general, we can imagine a $d$-dimensional theory with several distinct scales, $M_i$, possibly related to the presence of different towers of states. Following the logic behind the double EFT expansion, the field-theoretic piece would include various summations related to the different field-theoretic cutoffs, while the quantum-gravitational piece ought to be controlled by the single scale $\Lambda_{\mathrm{QG}}$. For illustrative purposes, let us focus on a setup with two field-theoretic scales satisfying $M_1 \ll M_2 \lesssim \Lambda_{\mathrm{QG}}$ and in which $M_1$ is associated to a $p$-dimensional Kaluza-Klein tower. Motivated by our toroidal example, we would expect the following structure for the Wilson coefficients:

$$
\alpha_n = a_n \Lambda_{\mathrm{QG}}^{2-n} + b_n M_1^{d-n} + c_n M_1^{-p} M_2^{d+p-n} + \cdots .
\tag{38}
$$

As usual, $a_n$, $b_n$ and $c_n$ are numerical coefficients that either become larger than or equal to some $\mathcal{O}(1)$ number, or they vanish exactly. The first and second terms correspond to the familiar quantum-gravitational and field-theoretic contributions (cf. (6) with $M \to M_1$). The third summand, on the other hand, is inherited from the $(d+p)$-dimensional field-theoretic contribution controlled by the second cutoff $M_2$, after reducing it to $d$ dimensions.

Interestingly, one can check that the third term in (38) is subleading with respect to the second one in the asymptotic regime $M_1 \ll M_2$ if and only if

$$
n > D = d + p ,
\tag{39}
$$

i.e., precisely when the underlying operator is irrelevant in the $(d+p)$-dimensional theory. Let us recall that this is precisely the case for which the computations in Section 2.2.1 converge. Nevertheless, on physical grounds one still expects these terms to arise whenever $n \leq D$. As a simple example, we consider again M-theory on $\mathbf{T}^2$, which corresponds to $p = 1$ in (38), and study the first higher-curvature operator $t_8 t_8 \mathcal{R}^4$. The latter is relevant in nine dimensions, and its contribution to the four-point graviton amplitude reads as [100, 109, 171]

$$
\mathcal{A}_{\mathcal{R}^4}^{(\mathbf{T}^2)} = \frac{\hat{K}}{\mathcal{V}_{\mathbf{T}^2}}\left(\frac{2\pi^2}{3}\mathcal{V}_{\mathbf{T}^2}^{2/3}M_{\mathrm{Pl},9}^{-14/3} + \mathcal{V}_{\mathbf{T}^2}^{1/2}E_{3/2}^{sl_2}(\tau,\bar{\tau})\right),
\tag{40}
$$

where $\hat{K}$ is displayed explicitly in (9). Thus, let us take an inhomogeneous decompactification limit back to 11d, with both $\mathcal{V}_{\mathbf{T}^2}M_{\mathrm{Pl},9}^2$ and $\operatorname{Im}\tau$ blowing up. The first term can be easily argued

to correspond to the *quantum-gravitational* contribution [49,50]. As for the second, using the the expansion (36) for the series $E_{3/2}^{sl_2}(\tau, \bar\tau)$, we find

$$
\begin{aligned}
\mathcal{A}_{\mathcal{R}^4}^{(\mathbf{T}^2)} &\supset \frac{\hat{K}}{\mathcal{V}_{\mathbf{T}^2}} \mathcal{V}_{\mathbf{T}^2}^{1/2} \left( 2\zeta(3)\operatorname{Im}\tau^{3/2} + 4\zeta(2)\operatorname{Im}\tau^{-1/2} + \mathcal{O}\left(e^{-2\pi\operatorname{Im}\tau}\right)\right) \\
&= \frac{\hat{K}}{\mathcal{V}_{\mathbf{T}^2}} \left( 2\zeta(3)M_1^{-1}M_2^2 + 4\zeta(2)M_1 + \mathcal{O}\left(e^{-2\pi\operatorname{Im}\tau}\right)\right),
\end{aligned}
\tag{41}
$$

where in the second step we have re-expressed everything in terms of the two distinct Kaluza-Klein masses $M_1, M_2$. Thus, by comparison with (38), it is easy to see that the first and second terms correspond to the nine- and ten-dimensional *field-theoretic* contributions. Notice, however, that when $n < D$ (as in the present example) the quantum-gravitational piece dominates over the $M_1$ and $M_2$ field-theoretic ones [49,53].[25] Hence, if the latter is non-vanishing, then the Wilson coefficient will be controlled by either one of the two familiar contributions encoded in the double EFT expansion (4).

To test this from the top-down, we consider the case of inhomogeneous decompactification limits to eleven-dimensional M-theory. Given that the higher the number of dimensions, the more operators satisfy $n < D$, this is arguably the most dangerous scenario one could envision. Furthermore, as explained above, M-theory has a plethora of higher-derivative terms in the Lagrangian without quantum-gravitational contribution. Any such operator with $n < 11$ would provide an example where the double EFT expansion does not encode the leading contribution to the corresponding Wilson coefficient. However, notice that the first such term without the quantum-gravitational piece in 11d is the $D^4\mathcal{R}^4$ one, which has $n = 12$. This way, we see how the double EFT expansion yields the leading contribution to the gravitational Wilson coefficients in a non-trivial way within this quantum gravity setup.

In summary, for all the cases presented herein, there are strong indications supporting the claim that the double EFT expansion (4) always captures the leading term of a given non-vanishing gravitational Wilson coefficient in any asymptotic regime. Even though other terms are generically present —and some of them even fit in the logic behind the double EFT expansion, they are always found to be subleading with respect to the field-theoretic/quantum-gravitational piece. It would be interesting to further support (or disprove) this remarkable and surprising power of the double EFT expansion. In Section 4, we will invoke this feature to derive bottom-up constraints on Wilson coefficients.

## 3 The black hole perspective

Let us now try to address the same question but from a completely different point of view. Namely, we seek to understand how purely non-perturbative gravitational physics, as described by the low energy EFT, can probe the relevant scales appearing in the effective Lagrangian (4). To do so, we consider black hole solutions and study the way in which their thermodynamic properties get modified upon including quantum corrections associated to the aforementioned

---

[25]This is easily seen by comparing the quantum-gravitational term with both the $M_1$ and $M_2$ field-theoretic ones in $d$- and $D$-dimensions, respectively. First, notice that consistency requires $\Lambda_{\mathrm{QG}} \lesssim M_{\mathrm{Pl,D}}$. Thus, the quantum-gravitational contribution dominates over the field-theoretic one associated to $M_1$ due to $n < D$ (cf. Section 2.2.3). Next, upon integrating in the extra dimensions relative to $M_1$, we land into a $D$-dimensional theory exhibiting the structure (4) with $M \sim M_2 \lesssim \Lambda_{\mathrm{QG}}$. If $M_2 \sim \Lambda_{\mathrm{QG}}$, then the quantum gravity term is automatically the leading one. Conversely, if $M_2 \ll \Lambda_{\mathrm{QG}}$, we expect that $M_2$ captures the KK scale of $p'$ extra dimensions. In this case, the quantum-gravitational piece still dominates due to $n < D + p'$. Since the quantum-gravitational and field-theoretic expansions of the $D$-dimensional EFT precisely yield the first and third terms in (38), this conclusion holds in the lower $d$-dimensional theory as well.

higher-dimensional and higher-curvature operators. Notice that this question nicely connects with recent results and observations made in several works [46, 48, 51, 172–176], where the relation between the minimal black hole entropy and the quantum gravity cutoff has been greatly explored. Additional complementary ideas have also been put forward in [119], which argues that a subtle phase transition for small enough uncharged black holes must always exist in quantum gravity due to the appearance of a more stable saddle in the gravitational path integral. Examples of this would be, e.g., the Gregory-Laflamme [177, 178] or Horowitz-Polchinski [179, 180] transitions. Our aim here will be to revisit these issues in the particular context of 4d $\mathcal{N} = 2$ theories obtained from compactifying Type II string theory on a Calabi–Yau threefold. We moreover focus on supersymmetric charged black hole solutions, whose macroscopic properties are captured by the effective field theory, and study in detail their entropy index as a function of the moduli fields evaluated at the horizon. Note that the latter are uniquely fixed by the black hole charges due to the attractor mechanism [181–184]. Furthermore, let us remark that, similarly to the four-point graviton scattering studied in Section 2, the black hole entropy corresponds to a purely gravitational observable and may, as such, be non-trivially affected by the EFT expansion displayed in eq. (4).

In this context, based on the recent analysis performed in [110], we argue that when restricting ourselves to the large volume regime, there seems to be a sharp transition which is attained precisely when the black hole reaches a size of the order of the Kaluza-Klein scale in the dual 5d M-theory picture. This is signaled by an apparently divergent contribution to the entropy index induced by certain local higher-dimensional chiral operators involving the graviton and graviphoton field strengths. In addition, we show that upon including further non-local effects associated to the D0-brane tower one is able to resum their contribution, yielding the correct, finite answer from the higher-dimensional perspective.

## 3.1 4d $\mathcal{N} = 2$ from type IIA Calabi–Yau compactifications

### 3.1.1 The two-derivative action

Let us consider 4d $\mathcal{N} = 2$ supersymmetric EFTs, as derived from Type IIA string theory compactified on a Calabi–Yau threefold $X_3$. The resulting EFT is characterized by the following two-derivative (bosonic) Lagrangian [185–188]

$$S_{4d} = \frac{1}{2\kappa_4^2} \int \mathcal{R} \star 1 - 2G_{a\bar{b}}\, dz^a \wedge \star d\bar{z}^b + \frac{1}{2} \mathrm{Re}\, \mathcal{N}_{AB} F^A \wedge F^B + \frac{1}{2} \mathrm{Im}\, \mathcal{N}_{AB} F^A \wedge \star F^B, \qquad (42)$$

where we have focused on the gravity and vector multiplet sector, which is the relevant one for our work. Here $z^a = b^a + it^a$, $a = 1, \ldots, h^{1,1}(X_3)$, denote the complex scalar fields describing the Kähler deformations of the threefold, whereas the vector fields descend from the dimensional reduction of the 10d Ramond-Ramond 1-form and 3-form potentials, yielding a total of $h^{1,1}(X_3) + 1$ abelian gauge bosons with field strengths $F^B = dA^B$, $B = 0, 1, \ldots, h^{1,1}(X_3)$.

Regarding their kinematics, the former scalar fields are described by a non-linear sigma model with a metric $G_{a\bar{b}} = \partial_a \partial_{\bar{b}} K$ that can be derived from the following Kähler potential

$$K = -\log i \left( \bar{X}^A \mathcal{F}_{0,A} - X^A \bar{\mathcal{F}}_{0,A} \right), \qquad (43)$$

whose explicit dependence on the moduli $z^a$ is determined via the holomorphic periods

$$X^0, \qquad X^a = X^0 z^a, \qquad \mathcal{F}_{0,A} = \frac{\partial \mathcal{F}_0}{\partial X^A}. \qquad (44)$$

Here $\mathcal{F}_0 = \mathcal{F}_0(X^A)$ is a holomorphic, homogeneous degree-two function of the electric periods $X^A$, usually referred to as the prepotential. Close to the large volume point, the latter can be

written as

$$\frac{\mathcal{F}_0(X^A)}{(X^0)^2} = -\frac{1}{6}\mathcal{K}_{abc}z^a z^b z^c + \mathbb{A}_{ab}z^a z^b + \frac{1}{24}c_{2,a}z^a + \frac{\zeta(3)\chi(X_3)}{2(2\pi i)^3} - \frac{1}{(2\pi i)^3}\sum_{k>0}n_k^{(0)}\text{Li}_3\left(e^{2\pi i k_i z^i}\right),$$
(45)

with the different coefficients in (45) depending on certain topological data of the threefold, see, e.g., [189] and references therein. In particular, what will be important for us is that $\mathcal{K}_{abc} = \int_{X_3}\omega_a \wedge \omega_b \wedge \omega_c$ are the triple intersection numbers, $c_{2,a}$ denote the expansion coefficients of the second Chern class —in a certain integral basis of $H^2(X_3)$, $\chi_E(X_3)$ is the Euler characteristic, and $n_k^{(0)}$ correspond to the genus-zero Gopakumar-Vafa invariants of the Calabi–Yau [112, 113].

Moreover, due to $\mathcal{N} = 2$ supersymmetry, the (complex) gauge kinetic matrix $\mathcal{N}_{AB}$ appearing in the action (42) can be also computed in terms of the Kähler structure deformations, see, e.g., [187] and references therein.

### 3.1.2 Higher-derivative operators and the double EFT expansion

Beyond the two-derivative level, there are in fact many possible higher-dimensional and higher-curvature operators that can arise after compactifying Type IIA on the threefold $X_3$. Among them, a particularly interesting subset of interactions are those that are moreover protected by supersymmetry, such that their field-dependence can be determined exactly in terms of the vector multiplet moduli. These corrections, involving the gravity and vector multiplet fields, include terms which may be written schematically as follows [190–193]

$$S_{4d} \supset -\frac{i}{2}\int d^4x \sqrt{-g}\sum_{k\geq 1}\mathcal{F}_k(z)\mathcal{R}_-^2(W_-^2)^{k-1} + \text{h.c.},$$
(46)

and supersymmetric partners thereof [114, 194]. Here $\mathcal{F}_k(z)$ denotes the genus-$k$ topological free energy, whilst $\mathcal{R}_-$ and $W_-$ are the anti-self-dual components of the Riemann and graviphoton field strengths,[26] respectively. Note that we have deliberately excluded the $k = 0$ term in the above series of local interactions, since it would rather account for the two-derivative action in the vector multiplet sector, such that one may identify $\mathcal{F}_0$ with the prepotential displayed in (45). In addition, it turns out that the moduli-dependent coefficients appearing in (46) can be equivalently computed using the duality between Type IIA string theory on $X_3$ and M-theory compactified on the same threefold times a circle [112, 113]. The generating function for these corrections is obtained via a one-loop calculation for each supersymmetry-preserving particle in the 5d theory,[27] which in the case of a hypermultiplet would read as

$$\sum_{k\geq 0}\mathcal{F}_k(z^a)W_-^{2k-2} = \frac{1}{4}\int_0^\infty \frac{d\tau}{\tau}\sum_{k\geq 0}\frac{(-1)^g 2^{2k}(2k-1)\mathcal{B}_{2k}}{(2k)!}\left(\frac{\tau^2 W_-^2}{4}\right)^{k-1}e^{-\tau Z}.$$
(47)

Here, $Z(q,p) = e^{K/2}\left(p^A\mathcal{F}_{0,A} - q_A X^A\right)$ denotes the central charge of the BPS state, which depends on the vector moduli via the holomorphic periods (44) and moreover determines the mass of the corresponding particle in 4d Planck units. We have also introduced above the Bernoulli numbers, namely $\mathcal{B}_{2k} = (-1)^{k+1}2(2k)!(2\pi)^{-2k}\zeta(2k)$.

In what follows, we will need the explicit expression for the different coefficients $\mathcal{F}_k(z^a)$ accompanying the higher-derivative operators. In this respect, a particularly interesting regime

---

[26]The graviphoton field strength $W_{\mu\nu}(x)$ is defined as the moduli-dependent combination of $U(1)$ gauge fields whose anti-self dual component appears in the gravitino supersymmetry variation, namely $W_- = 2ie^{K/2}\text{Im}\mathcal{N}_{AB}X^A F_-^B$ [195].

[27]Notice that all the non-trivial moduli dependence of the Wilson coefficients in (47) arises from the central charge of the supersymmetric particles running in the loop.

that will prove to be useful is the large volume patch. There, the prepotential (45) may be approximated by the leading-order cubic piece, whereas the genus-one topological string amplitude simplifies to [190, 191]

$$\mathcal{F}_1(z^a) = \frac{1}{24} \int_{X_3} J_{\mathbb{C}} \wedge c_2(TX_3) + \cdots = \frac{1}{24} c_{2,a} z^a + \cdots, \tag{48}$$

with $J_{\mathbb{C}}$ the complexified Kähler form and $c_2(TX_3)$ the second Chern class of the tangent bundle of $X_3$. Similarly, higher-genus corrections obtained from D0-brane states in (47) take the following simple form [112, 196]

$$\mathcal{F}_{k>1}(z^a) = \chi(X_3) \frac{2(2k-1)\zeta(2k)\zeta(2k-2)\Gamma(2k-2)}{(2\pi)^{2k}} \left( \frac{2\pi M_s}{g_s} \right)^{2-2k} + \cdots, \tag{49}$$

where $\chi_E(X_3) = 2\left(h^{1,1}(X_3) - h^{2,1}(X_3)\right)$ is the Euler characteristic of the threefold $X_3$. Notice that, contrary to the $k = 1$ case, the quantities $\mathcal{F}_{k>1}$ are dimensionful and in fact can be seen to be explicitly controlled by the D0-brane mass [49].

Finally, let us take the opportunity to connect the present setup with our general discussion from Section 1.1. Notice that, from the expression (46) and the large volume analysis presented herein, it is clear that the double EFT expansion holds in these kind of theories as well. The reason being that the set of higher-curvature operators of the form $\mathcal{O}_{2k+2}(\mathcal{R}, W) = \mathcal{R}_-^2 W_-^{2k-2}$, which are gravitational in origin, are indeed suppressed either by the tower mass scale $M_{\text{D}0} = 2\pi M_s/g_s$ to the power $n - d$ (for $n > d$), where $n = 2k + 2$ is the classical dimension associated to the corresponding $\mathcal{O}_{2k+2}(\mathcal{R}, W)$ operator; or rather present a species scale suppression of the form specified by (3) for the case $k = 1$ [45, 47, 49].

More generally, one can similarly show that upon exploring other possible infinite distance limits lying purely within the vector multiplet sector, which end up being either partial decompactifications to 6d F-theory or weakly-coupled limits to a dual heterotic 4d compactification [25], the expectations coming from the double EFT expansion are also borne out [49, 53]. Indeed, in the former case the relevant tower of asymptotically massless particles is comprised by bound states of D0- and D2-branes wrapping the elliptic fibre of the Calabi–Yau, whereas for the latter the object setting both the tower mass and the quantum gravity cutoff corresponds to an NS5-brane wrapping the $K3$ fibre.

## 3.2 BPS black holes and the entropy index

### 3.2.1 The attractor mechanism

Within this kind of theories, there exist certain supersymmetric black hole solutions exhibiting various interesting properties. These configurations can be moreover regarded as interpolating solitons between two maximally symmetric backgrounds: a Minkowski vacuum at infinity, and a Bertotti-Robinson Universe of topology $\text{AdS}_2 \times \mathbf{S}^2$ [182, 197]. The latter occurs close to the horizon, whose metric reads —in isotropic coordinates— as [198, 199]

$$ds^2 = -e^{2U(r)}dt^2 + e^{-2U(r)}\left(dr^2 + r^2 d\Omega_2^2\right), \qquad \text{with} \ \ e^{-2U(r)} = \frac{|Z|^2 \kappa_4^2}{8\pi r^2}. \tag{50}$$

Here, $Z$ denotes the central charge of the black hole, and it depends on the Kähler moduli evaluated at the horizon locus at $r = 0$. These are completely determined, in turn, by the charges $(p^A, q_A)$ of the black hole through the attractor (or stabilization) equations [181–184]

$$ip^A = C X^A - \bar{C}\bar{X}^A, \qquad iq_A = C\mathcal{F}_{0,A} - \bar{C}\bar{\mathcal{F}}_{0,A}, \tag{51}$$

which relate the latter to rescaled —by a compensator field $C = e^{K/2}\bar{Z}$— versions of the holomorphic periods introduced in (44). Notice that the black hole entropy, which is computed at leading order by the horizon area divided by $\kappa_4^2/2\pi$ [200, 201], can be seen to be given by $S_{BH} = \pi|Z|^2$, as per (50).

Interestingly, it turns out that one can extend the previous analysis so as to include higher-curvature and higher-derivative operators, which should not only change the explicit form of the solution, but also its associated thermodynamic properties. Here we will only briefly outline the most relevant modifications, which will be important for us in what follows. A more detailed account of the procedure can be found in [202] as well as in the original works, see, e.g., [111, 114–118] for an incomplete list of references.

First of all, the analogous quantity determining the area of the black hole becomes a symplectic extension of the aforementioned central charge, which we denote by $\mathscr{Z}$. It is defined as follows:

$$|\mathscr{Z}|^2 = p^A F_A(Y, \Upsilon) - q_A Y^A = e^{\mathscr{K}} \left| p^A F_A(X, W_-^2) - q_A X^A \right|^2 , \tag{52}$$

where $\mathscr{K}$ corresponds to the symplectic combination

$$\mathscr{K} = -\log\left(i\bar{X}^A F_A(X, W_-^2) - iX^A \bar{F}_A(\bar{X}, \bar{W}_-^2)\right) , \tag{53}$$

which is nothing but a generalization of the usual Kähler potential. We have also introduced the following rescaled variables [111]

$$Y^A = e^{\mathscr{K}/2}\bar{\mathscr{Z}}X^A , \qquad \Upsilon = e^{\mathscr{K}}\bar{\mathscr{Z}}^2 W_-^2 , \tag{54}$$

thus generalizing the compensator field defined above, and with $W_-^2$ being the (anti-self-dual) graviphoton field strength squared. Furthermore, the quantities $F_A(X, W_-^2)$ are the first derivatives with respect to $X^A$ of the generalized holomorphic prepotential

$$F(Y, \Upsilon) = \sum_{g=0}^{\infty} F_g(Y^A)\Upsilon^g , \tag{55}$$

whose coefficients $F_g(X^A)$ can be essentially identified with the topological string amplitudes appearing in (46) (see, e.g., [110] for details).

Crucially, in terms of these, the stabilization equations maintain their form, namely

$$ip^A = Y^A - \bar{Y}^A , \qquad iq_A = F_A(Y, \Upsilon) - \bar{F}_A(\bar{Y}, \bar{\Upsilon}) , \tag{56}$$

whilst $\Upsilon$ is set to $-64$. On the other hand, the BPS black hole entropy is given by the expression [114]

$$\mathcal{S}_{BH} = \pi\left[|\mathscr{Z}|^2 + 4\text{Im}\left(\Upsilon\partial_\Upsilon F(Y, \Upsilon)\right)\right] . \tag{57}$$

This formula generalizes the Bekenstein-Hawking area law by incorporating quantum corrections arising from higher-genus Gopakumar-Vafa terms, and in fact corresponds to an index-like quantity that is protected by supersymmetry and thus only receives contributions from supersymmetry-preserving operators. Therefore, it is widely believed (see, e.g., [202, 203]) that all such contributions are already captured by the infinite series of terms shown in (46).

Lastly, and following the discussion of the previous section, let us specialize the above formulae to the large volume regime, which is where our analysis will be placed. There, the generalized prepotential introduced in (55) can be written as [117]

$$F(Y, \Upsilon) = \frac{D_{abc}Y^a Y^b Y^c}{Y^0} + d_a \frac{Y^a}{Y^0}\Upsilon + G(Y^0, \Upsilon) + \cdots , \tag{58}$$

where $D_{abc} = -\frac{1}{6}\mathcal{K}_{abc}$ and $d_a = \frac{1}{24}\frac{1}{64}c_{2,a}$ are determined by the topological data of the Calabi–Yau threefold, namely the triple intersection numbers $\mathcal{K}_{abc}$ as well as the second Chern class (cf. eq. (48)), and the ellipsis are meant to indicate further subleading worldsheet instanton contributions. The first two terms in (58) determine the leading-order contribution to $F(Y,\Upsilon)$ at genus zero and one, respectively, whereas the function $G(Y^0,\Upsilon)$ is related instead to the one-loop determinant (47), and reads

$$G(Y^0,\Upsilon) = -\frac{i}{2(2\pi)^3}\chi_E(X_3)(Y^0)^2 \sum_{k=0,2,3,\dots} c^3_{k-1}\alpha^{2k} + \cdots, \tag{59}$$

with

$$c^3_{k-1} = (-1)^{k-1}2(2k-1)\frac{\zeta(2k)\zeta(3-2k)}{(2\pi)^{2k}}, \qquad \alpha^2 = -\frac{1}{64}\frac{\Upsilon}{(Y^0)^2}. \tag{60}$$

### 3.2.2 Reaching the UV cutoff scale

Up to now we have reviewed the possibility of constructing supersymmetric black hole solutions in 4d $\mathcal{N}=2$ theories arising as low energy limits of certain string theory constructions. We moreover discussed that, within the large volume regime, these objects are completely characterized to leading order by their gauge charges as well as the topology of the compactification space. However, we also pointed out that there are further local higher-curvature operators involving the graviton and graviphoton field strengths that modify the black hole entropy through a perturbative series of corrections. Our aim here will be to investigate more closely the physical meaning of the expansion parameter $\alpha$, and try to elucidate whether the EFT transition occurring at the Kaluza-Klein scale could be detected through those as well.

To begin with, let us notice that the coefficients appearing in the higher-derivative terms controlling the quantum deformations of the black hole solutions grow in a factorial way, namely $c^3_{k-1} \sim (2k-3)!$ (cf. eq. (49)). This means, in turn, that the series expansion

$$G(Y^0,\Upsilon) \sim -\frac{i}{2(2\pi)^3}\chi_E(X_3)(Y^0)^2 \sum_{k=0}^{\infty} c^3_{k-1}\alpha^{2k}, \tag{61}$$

has zero radius of convergence and can be treated at best as an *asymptotic* approximation [204, 205], which is valid for $|\alpha| \ll 1$. Hence, for any fixed order $N$ in the sum (61), the truncated series —up to and including $k = N$— provides a better estimate for $G(\alpha,\Upsilon)$ the smaller $|\alpha|$ is. Reciprocally, the larger we take the expansion parameter, the more it deviates from the correct resummed result (see Figure 2), leading ultimately to a seemingly divergent-like behavior. Therefore, it is natural to conclude that the regime $|\alpha| \gtrsim \mathcal{O}(1)$ becomes pathological from the 4d EFT perspective,[28] since the quantum corrections to, e.g., the black hole entropy provided by the latter get unreliable from that point onwards.

In order to understand what this is telling us, let us recall here the defining relation of the expansion parameter

$$|\alpha| = \frac{1}{8}|\Upsilon|^{1/2}|Y^0|^{-1} = \frac{1}{8}\frac{|\Upsilon|^{1/2}}{|X^0 e^{\mathcal{K}/2}\mathcal{Z}|}, \tag{62}$$

where these quantities should be evaluated at the attractor point solving the algebraic equations (56). Furthermore, upon substituting the mass of the D0-brane into (62), the above

---

[28]This expectation is confirmed by looking at the optimal truncation in the series (61), whose maximum order can be seen to be $k_\star \sim \frac{1}{2}\left(1 + \frac{4\pi^2}{|\alpha|}\right)$, thus essentially collapsing to the first term whenever $|\alpha| \gtrsim 1$ [110].

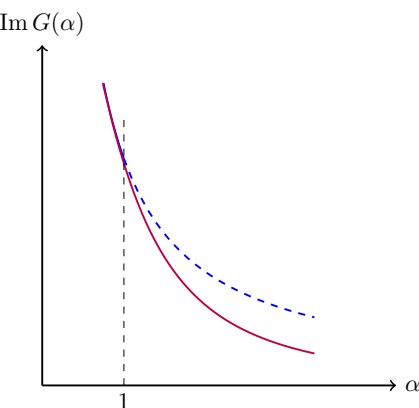

Figure 2: Profile of the quantity $\operatorname{Im} G(\alpha, \Upsilon)$ capturing the relevant quantum corrections to the generalized holomorphic prepotential $F(Y^A, \Upsilon)$ at large volume, as a function of the expansion parameter $\alpha$ evaluated at the horizon locus. For illustration, we have taken $\chi_E(X_3)$ positive and $\alpha$ to be real-valued (cf. Section 3.2.3). The blue (dashed) line shows the optimal truncation provided by the asymptotic series (61), whereas the purple (solid) curve corresponds to the exact resummed function displayed in (79). Both curves agree to very high precision when $\alpha \ll 1$, whilst for $\alpha \gtrsim 1$ they start departing from each other in an exponential manner [110].

expression simplifies to[29]

$$\left| \alpha(q_A, p^B) \right| = \frac{\sqrt{8 \mathcal{V}_h}}{|\mathcal{Z}(q_A, p^B)|} = \frac{r_5}{r_h} \bigg|_{\text{hor}} , \tag{63}$$

with $\mathcal{V}_h$ denoting the physical volume of the Calabi–Yau threefold measured in string units within the near-horizon geometry [110]. This implies then that the modulus of $\alpha$, when evaluated at the attractor locus, represents the ratio between the black hole radius $r_h$ and that of the M-theory circle, denoted here by $r_5$. Incidentally, this allows us to interpret in very simple physical terms the transition point occurring at $|\alpha| \sim \mathcal{O}(1)$, where the local 4d EFT starts producing unphysical results. Indeed, the failure to describe these black holes arises because the compactified extra dimension is no longer small relative to their horizon. Therefore, local fluctuations in the classical geometry can now easily excite massive Kaluza-Klein replica on the circle, such that a purely 4d description becomes inadequate. At this point, a higher-dimensional theory is needed, which should moreover be able to resolve these issues and provide a better account of the physical properties of the aforementioned black hole solutions. This, in fact, can be seen to be what happens after performing a resummation of the series (61), whose details will mostly depend on the particular black hole solution under consideration, as we show next.

### 3.2.3 An explicit example: Black holes with vanishing D6-brane charge

In order to illustrate the main points of our perhaps abstract discussion above, let us focus on a particular class of 4d black hole solutions that are characterized by having vanishing D6-brane charge. The reason for selecting those is twofold. Firstly, from the mathematical point of view, the algebraic system derived from the attractor equations (51) can be shown

---

[29]The identity (63) follows from the value of the horizon radius, i.e., $r_h = |\mathcal{Z}| \kappa_4 / \sqrt{8\pi}$ [114] (cf. eq. (50)), as well as the Kaluza-Klein length-scale $r_5 = M_{\text{D0}}^{-1}$, where the D0-brane mass is computed to be $M_{\text{D0}} = \sqrt{8\pi} |X^0| e^{\mathcal{K}/2} / \kappa_4$.

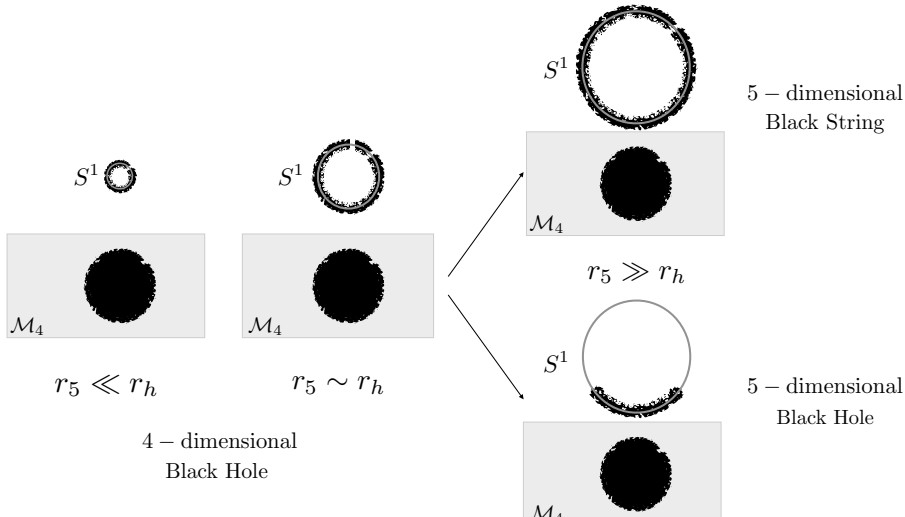

Figure 3: Schematic depiction of the 4d black hole solutions as a function of the parameter $|\alpha| = r_5/r_h$. For a horizon radius much larger than the radius of the M-theory circle (left), the 4d EFT gives the correct result. As $|\alpha|$ becomes $\mathcal{O}(1)$ (middle), the 4d EFT solution starts to break down, and the series of corrections must be completed to provide the 5d picture. Finally, in the limit where the radius of the M-theory circle is much larger than that of the black hole along the four non-compact dimensions, the full solution must be treated in the 5d EFT, and can *a priori* yield either a 5d black string wrapping the extra circle (top right) or a 5d black hole (bottom right). The BPS solution that we explicitly consider, with vanishing D6 charge, uplifts to a 5d supersymmetric black string.

to (essentially) always admit a well-behaved solution [110, 206], regardless of the choice for the remaining gauge charges. Secondly, from the physical perspective, they can be interpreted as 5d black strings wrapped on the M-theory circle, as schematically displayed in Figure 3.[30] Therefore, since in this case the horizon becomes transverse to the circular direction, they are able to probe both the four- and five-dimensional realms, giving us some valuable insight into how the entropy index interpolates between these two regimes [110]. This is particularly important in light of recent discussions involving the question as to what is the size/entropy of the minimal black hole that can be described within a given gravitational EFT, and its relation to the quantum gravity cutoff [46, 48, 51, 172–176].[31]

**The two-derivative analysis**

Following the original references [111, 206], we first analyze the aforementioned physical system at the two-derivative level and in the large volume approximation. Thus, having no D6-brane charge translates mathematically into the condition $p^0 = 0$. This implies, in turn, that $CX^0$ is purely real, such that the algebraic system (51) becomes linear for the remaining (rescaled) variables $CX^a$, allowing us to solve for them directly as follows

$$CX^a = \frac{1}{6} CX^0 D^{ab} q_b + \frac{i}{2} p^a \,, \tag{64}$$

---

[30]Let us emphasize here that not every 4d BPS black hole uplifts to some (combination of) 5d wrapped BPS string(s) [207], since there is also the possibility of placing a putative 5d black hole at the center of a Taub-NUT geometry [208, 209]. The latter are, however, not able to probe the regime $|\alpha| \gg 1$ [110].

[31]The idea that along decompactification limits the (neutral) *minimal black holes* in the EFT —i.e., those probing the species scale and whose entropy is proportional to the number of light species— actually correspond to black branes/strings wrapping the large extra dimensions has been recently proposed in [175].

where $D^{ab}$ is the inverse matrix of $D_{ab} \equiv D_{abc}p^c$, which we assume to exist. On the other hand, from the $q_0$-equation, we also obtain

$$(CX^0)^2 = \frac{1}{4} \frac{D_{abc}p^a p^b p^c}{\hat{q}_0} \equiv (x^0)^2, \tag{65}$$

with $\hat{q}_0 = q_0 + \frac{1}{12}D^{ab}q_a q_b$. Note that we take $p^a > 0$ and $\hat{q}^0 < 0$ so as to have positive definite Kähler volumes (cf. eqs. (67) and (68) below). Hence, as advertised, one finds all moduli fields becoming functions that, when evaluated at the horizon locus, depend solely on the black hole charges.

Similarly, one can obtain analogous expressions both for the extremized central charge of the black hole $Z$ and its entropy $S_{BH}$, once the attractor values for $CX^A$ are known. These read as

$$|Z(q_A, p^B)|^2 = -\frac{D_{abc}p^a p^b p^c}{CX^0} = 2\sqrt{\frac{1}{6}|\hat{q}_0|\mathcal{K}_{abc}p^a p^b p^c}, \tag{66a}$$

$$S_{BH}(q_A, p^B) = -4\pi CX^0 \hat{q}_0 = 2\pi\sqrt{\frac{1}{6}|\hat{q}_0|(\mathcal{K}_{abc}p^a p^b p^c)}, \tag{66b}$$

in agreement with the Bekenstein-Hawking area law. Finally, in order to convince ourselves that the solution just found actually belongs to the large volume regime, we must ensure that the moduli profile induced by the black hole background remains therein. This requires from having both *i)* the overall threefold volume $\mathcal{V}$ and *ii)* the individual 2-cycle volumes large all along the BPS flow. Therefore, assuming that these two conditions are met by the boundary values of the fields in the Minkowski vacuum implies, per the monotonicity of the solution [181], that we only need to care about what happens at the horizon locus. Upon computing the overall Calabi–Yau volume

$$\mathcal{V}_h = \frac{1}{8}e^{-K}|X^0|^{-2}\Big|_{\text{hor}} = \frac{1}{8}\frac{|Z|^2}{|CX^0|^2} = \sqrt{\frac{6|\hat{q}_0|^3}{\mathcal{K}_{abc}p^a p^b p^c}}, \tag{67}$$

as well as that associated to the minimal-size 2-cycles in the geometry

$$t_h^a = \text{Im}\left(\frac{CX^a}{CX^0}\right)\Big|_{\text{hor}} = p^a\sqrt{\frac{6|\hat{q}_0|}{\mathcal{K}_{abc}p^a p^b p^c}}, \tag{68}$$

we deduce the following charge hierarchy

$$(\hat{q}^0)^2, (p^a)^2 \gg \left|\frac{D_{abc}p^a p^b p^c}{\hat{q}_0}\right|, \qquad \text{with} \quad |\hat{q}_0|, p^a \gg 1, \tag{69}$$

to be necessary for the large radius approximation to apply herein.[32] Note, however, that we have not specified the behavior of the numerical coefficient $(\hat{q}_0)^{-1}D_{abc}p^a p^b p^c$, which we can readily identify with $(2x^0)^2$ —namely with the value of the $CX^0$ variable evaluated at the attractor point. As we will see, this quantity will play a crucial role in what follows.

---

[32]Large individual charges are generically required for the black hole to have a mass way above the 4d Planck scale, since the former is determined by the central charge evaluated in the asymptotically flat $r \to \infty$ regime. Still, it might be possible to have species-scale-sized black holes with $|\hat{q}^0| \gg 1$ and $p^a \sim \mathcal{O}(1)$ which are much heavier than the 4d Planck scale if $\Lambda_{QG} \ll M_{Pl,4}$ [172, 176].

**Including higher-derivative corrections**

The previous solution gets significantly modified once we allow for higher-derivative terms in the 4d action (42) to be present [114, 116, 210]. More precisely, one finds that, even though the functional dependence of the rescaled variables $Y^a$ remains of the form specified by (64), the new value for the stabilized $Y^0$ is determined by the implicit equation [117]

$$(Y^0)^2 = \frac{\frac{1}{4}D_{abc}p^a p^b p^c - d_a p^a \Upsilon}{\hat{q}_0 + i(G_0 - \bar{G}_0)}, \tag{70}$$

where $G_0 \equiv \partial G(Y^0, \Upsilon)/\partial Y^0$, cf. eq. (58). On the other hand, the analogous physical quantities which are relevant to describe the thermodynamic properties of the black hole solution get deformed as explained in Section 3.2.1. In particular, the generalized central charge $\mathscr{Z}$ and the entropy index $\mathcal{S}_{\text{BH}}$ now read

$$|\mathscr{Z}(q_A, p^B)|^2 = -\frac{D_{abc}p^a p^b p^c - 2d_a p^a \Upsilon}{Y^0} + iY^0\left(G_0 - \bar{G}_0\right), \tag{71a}$$

$$\mathcal{S}_{\text{BH}}(q_A, p^B) = -4\pi Y^0 \hat{q}_0 - i\pi\left(3Y^0 G_0 + 2\Upsilon G_\Upsilon - \text{h.c.}\right), \tag{71b}$$

which are distinct from those computed in (66) due to both the different value for $CX^0$ and $Y^0$ at the attractor point, as well as the presence of additional corrections depending on the higher-derivative terms in the prepotential $F(Y, \Upsilon)$.

It is now easy to see that, for us to recover the same results as in the previous two-derivative analysis, we need to choose the charges such that not only (69) is satisfied but also $|\hat{q}_0| \gg |i(G_0 - \bar{G}_0)|$ holds at the horizon. This restriction, in turn, allows us to solve eq. (70) iteratively as follows

$$(Y^0)^2 = (y^0)^2\left(1 + \frac{i(G_0(y^0, \Upsilon) - \bar{G}_0(\bar{y}^0, \bar{\Upsilon}))}{|\hat{q}_0|} + \cdots\right), \tag{72}$$

where $(y^0)^2 = (x^0)^2\left(1 - 4d_a p^a \Upsilon/D_{bce}p^b p^c p^e\right)$ and the ellipsis denote higher-order terms in the expansion. In this case, the generalized central charge and black hole entropy index defined in eqs. (52) and (57) simplify to

$$|\mathscr{Z}|^2 = 2\sqrt{\frac{1}{6}|\hat{q}_0|\mathcal{K}_{abc}p^a p^b p^c} + \cdots, \tag{73a}$$

$$\mathcal{S}_{\text{BH}} = 2\pi\sqrt{\frac{1}{6}|\hat{q}_0|\left(\mathcal{K}_{abc}p^a p^b p^c + c_{2,a}p^a\right)} + \cdots. \tag{73b}$$

Note that the expression for the corrected entropy in (73b) agrees with the leading-order microscopic counting result [211, 212]. Now, to properly understand what the additional constraint imposed above is telling us, let us compute the imaginary part of $G_0(Y^0, \Upsilon)$. From the local 4d EFT one obtains the asymptotic series [110]

$$i\left(\bar{G}_0 - G_0\right) = -\frac{\chi_E(X_3)}{8(2\pi)^3}|\Upsilon|^{1/2}\sum_{k=0,2,3,\dots}(2 - 2k)c_{k-1}^3 \alpha^{2k-1} + \cdots, \tag{74}$$

which grows like $\alpha^{-1}$ for $\alpha \ll 1$. Hence, we conclude that the consistency of the solution requires from having $Y^0 \gg 1$ at the attractor locus, which implies the following refined charge hierarchy

$$(\hat{q}^0)^2, (p^a)^2 \gg \left|\frac{D_{abc}p^a p^b p^c}{\hat{q}_0}\right| \gg 1. \tag{75}$$

From here, it becomes straightforward to deduce the contribution of $G$-dependent terms to the entropy index, yielding a final result of the approximate form

$$\mathcal{S}_{\text{BH}} = 2\pi \sqrt{\frac{1}{6}|\hat{q}_0| \left( \mathcal{K}_{abc} p^a p^b p^c + c_{2,a} p^a \right) - \frac{\chi_E(X_3)}{4\pi^2} \sum_k c_{k-1}^3 (y^0)^{2-2k}} + \cdots, \qquad (76)$$

where the ellipsis account for further subleading terms in $1/|\hat{q}^0|$ (cf. eq. (72)), and once again the sum runs over the genus expansion in the topological string theory or, equivalently, over the higher-derivative operators shown in (46).

Crucially, and following the general discussion of Section 3.2.2, one can easily get convinced that precisely when we try to look for solutions where the parameter $\alpha$ —which is determined in turn by the classical value $x^0$— controlling the asymptotic expansions (74) and (76) becomes of order one or larger, our scheme breaks down completely [110]. Indeed, the correction terms induced by the higher-derivative chiral operators (cf. (46)) start giving misleading results for any truncated expression of the series (61) and can readily overcome the tree-level piece, thus effectively invalidating our 4d analysis from eq. (72) onwards.

**Completing the series in the ultra-violet**

As we argued in Section 3.2.2, the physical interpretation of the failure by the four-dimensional EFT to account for the thermodynamic properties of black holes with an $\alpha \gtrsim \mathcal{O}(1)$ has to do with the fact that for those values of the expansion parameter, the relative size of the horizon and extra hidden circle become comparable to each other. Hence, at this point one should not expect the EFT to fully capture the physics associated to these objects, given that they could easily excite heavier Kaluza-Klein states. On the other hand, from the naive two-derivative analysis performed in Section 3.2.3 it is reasonable to expect that, upon choosing appropriately the gauge charges, one could reach a regime where the size of the horizon transverse to the 5d black string wrapping the circle gets parametrically lowered with respect to its radius, as depicted in Figure 3. In the following we would like to entertain ourselves with the question of whether and how the 5d uplifted theory resolves all these issues, ultimately providing for a meaningful physical answer. We follow closely the treatment in [110].

To do so, we must revisit the one-loop calculation that yields the perturbative corrections to the generalized prepotential. For the D0-brane tower, one may write

$$G(Y^0, \Upsilon) = \frac{i}{2(2\pi)^3} \chi_E(X_3)(Y^0)^2 \mathcal{I}(\alpha), \qquad (77)$$

where $\mathcal{I}(\alpha)$ is defined as follows [112, 113]

$$\mathcal{I}(\alpha) = \frac{\alpha^2}{4} \sum_{n \in \mathbb{Z}} \int_0^\infty \frac{ds}{s} \frac{1}{\sinh^2(\pi n \alpha s)} e^{-4\pi^2 n^2 i s}. \qquad (78)$$

Notice that, if we expand the $\sinh^2(x)$ appearing in the denominator above using its Laurent series around the origin, we recover the same formula as in (47). However, we can do better and use the identity $\sum_{n \in \mathbb{Z}} e^{2\pi i n \theta} = \sum_{k \in \mathbb{Z}} \delta(\theta - k)$ so as to compute exactly the above integral. Upon doing so and substituting back in (77), one obtains [110]

$$G(Y^0, \Upsilon) = \frac{i}{2(2\pi)^3} \chi_E(X_3)(Y^0)^2 \alpha^2 \sum_{n=1}^\infty n \operatorname{Li}_1 \left( e^{-\alpha n} \right), \qquad (79)$$

where $\operatorname{Li}_k(x) = \sum_{n=1}^\infty \frac{x^n}{n^k}$ denotes the $k$-th polylogarithm function, which converges whenever $|x| < 1$. Let us stress that (79) is well-behaved for all $\alpha \geq 0$. In fact, for small values of the

expansion parameter, one can easily show that $G(Y^0, \Upsilon)$ behaves as

$$G(Y^0, \Upsilon) \sim \frac{i}{2(2\pi)^3} \chi_E(X_3) \zeta(3) \left( \frac{-\Upsilon}{64} \right) \alpha^{-2}, \tag{80}$$

in agreement with the asymptotic estimation given by the 4d EFT. On the contrary, when $\alpha \gtrsim \mathcal{O}(1)$ the non-analyticities around $\alpha = 0$ signal the necessity of introducing further non-local corrections beyond the series (58), and the 5d picture indeed becomes essential. In any event, what remains true is that, even in this regime, the resummed expression for $G(Y^0, \Upsilon)$ is such that $|\hat{q}_0| \gg |i(G_0 - \bar{G}_0)|$ always holds at the attractor point —when assuming the hierarchy (69). Hence, the iterative solution described around eq. (72) is still valid, yielding a final result for the resummed entropy index within the large volume approximation that reads as [110]

$$
\begin{aligned}
\mathcal{S}_{\mathrm{BH}} = {}& 2\pi \sqrt{\frac{1}{6} |\hat{q}_0| \left( \mathcal{K}_{abc} p^a p^b p^c + c_{2,a} p^a \right)} \left( 1 - \frac{\chi_E(X_3) Y^0 \alpha^2}{(2\pi)^3 |\hat{q}_0|} \sum_{n=1}^{\infty} n^2 \mathrm{Li}_0 \left( e^{-\alpha n} \right) \right)^{-1/2} \\
& + \frac{\chi_E(X_3)}{4\pi^2} (Y^0)^2 \alpha^2 \left( \sum_{n=1}^{\infty} n \, \mathrm{Li}_1 \left( e^{-\alpha n} \right) + (Y^0)^{-1} \sum_{n=1}^{\infty} n^2 \mathrm{Li}_0 \left( e^{-\alpha n} \right) \right),
\end{aligned}
\tag{81}
$$

where one should substitute above the particular value of $Y^0$ that solves the implicit equation (70). Interestingly, it can be shown that upon taking the 5d limit (i.e., $\alpha \to \infty$), the quantity $G(Y^0, \Upsilon)$ and, consequently, the one-loop corrections to $\mathcal{S}_{\mathrm{BH}}$ appearing in (81), vanish in an exponential fashion. The only piece surviving corresponds to the $\mathcal{R}^2$-operator, which also appears in the five-dimensional action of M-theory compactified on the same Calabi–Yau [211]. In addition, one can show that the entropy non-trivially coincides with the classical and one-loop exact analogue in the uplifted theory [110], as it should be.

To close this section, let us make two important remarks. Firstly, one could have wondered about whether non-perturbative stringy corrections to the generalized prepotential (58) would spoil the present analysis, rendering the 5d wrapped string solution inconsistent or, to the very least, unstable. However, it turns out that one can take these effects into account —which from the dual 5d M-theory perspective would correspond to D0-brane pair production in the black hole background, and show that in fact they do not play any role herein [110]. This also matches very nicely with the results of [213], where these solutions are embedded (at the two-derivative order) within 5d supergravity and are moreover argued to exist for all sizes of the asymptotic compactification circle. Secondly, let us also stress that our analysis reinforces the idea that the most relevant corrections to the black hole entropy arise from higher-curvature operators suppressed by $\Lambda_{\mathrm{QG}}$, such that the minimal black hole size is precisely attained when the linear term in the prepotential starts to dominate over the cubic piece, thus corresponding to the 5d Planck scale (i.e., the quantum gravity cutoff), in agreement with other recent studies [46, 48, 110, 176].

## 4 The bottom-up perspective

In this last section, we illustrate how the double EFT expansion can be exploited so as to derive various kinds of interesting constraints on the gravitational Wilson coefficients, especially in the asymptotic regime (5). Due to the inherent multi-scale structure of the setup, these bounds will go beyond those suggested by naive dimensional analysis. First, in Section 4.1, we discuss lower and upper bounds on a given Wilson coefficient with respect to the EFT cutoff. Subsequently, we derive in Section 4.2 certain non-trivial relations between different Wilson coefficients as they become large in Planck units. Finally, we also discuss bounds relating different combinations of Wilson coefficients and the EFT cutoff in Section 4.3, as well as

compare our findings with S-matrix bootstrap results. In order to illustrate how the last two types of relations appear, we focus on ten-dimensional maximal supergravity theories, where we comment on the nice interplay between the constraints coming from (4) with existing S-matrix bootstrap analyses.

Placing these bounds will require focusing on certain Wilson coefficients, instead of considering —as done in previous sections— the quantum-gravitational and field-theoretic expansions as a whole. As discussed at the end of Section 1.1, this raises the question of whether the double EFT expansion should capture the leading contribution to a given gravitational Wilson coefficient in the asymptotic regime. Remarkably, this seems to be always the case when inspecting top-down models (see Section 2.3.2 for general arguments in this direction). In what follows, we will assume this feature to hold and explore its consequences.

The motivation for this section is twofold. On one hand, we would like to stress that the double EFT expansion potentially encodes non-trivial bounds on Wilson coefficients which can serve as interesting targets for future bottom-up/bootstraps searches. On the other hand, adopting a perhaps complementary point of view, we will see how S-matrix bounds may inform the precise structure of the double EFT expansion. For instance, they will sometimes imply a non-trivial quantum-gravitational contribution to some Wilson coefficients.

As we will see, oftentimes it is convenient to assume that the scale $M$ corresponds to that of an infinite tower of states. Since this distinction matters from the bottom-up perspective —namely, considering a given effective theory vs. any EFT with finite number of degrees of freedom— we will explicitly indicate when this assumption is required by writing $M_{\mathrm{t}}$ instead of $M$. In particular, the latter becomes useful when combined with the Emergent String Conjecture [25]. Remarkably, this criterion leaves a very precise imprint in some of the derived bounds, thus opening up the possibility of testing this quantum gravity/Swampland constraint with S-matrix bootstrap techniques.

## 4.1 Wilson coefficients and the EFT cutoff

Let us first explore bounds on the Wilson coefficients in terms of the EFT cutoff, which as stressed in Section 1.1 is identified with the scale of the new degrees of freedom, $M$. Once again, we stress that we will always consider in what follows the asymptotic regime (5). Therefore, all the bounds will be strictly valid in the limit $M \to 0$ in Planck units.

According to the double EFT expansion, the Wilson coefficient accompanying a certain dimension-$n$ operator in the effective action takes the generic form (6) ($M_{\mathrm{Pl},d} = 1$), which we recall here for the comfort of the reader

$$\alpha_n = \frac{a_n}{\Lambda_{\mathrm{QG}}^{n-2}} + \frac{b_n}{M^{n-d}} + \cdots. \tag{82}$$

As we have seen in previous sections, depending on the rate at which both scales $\Lambda_{\mathrm{QG}}$ and $M$ go to zero asymptotically, a given Wilson coefficient can be controlled by one scale or the other. Despite this fact, the consistency condition $M \lesssim \Lambda_{\mathrm{QG}}$ implies a general upper bound on both terms. Indeed, for $M$ fixed, (82) is maximized when the inequality above is saturated, i.e $M \sim \Lambda_{\mathrm{QG}}$. This yields

$$|\alpha_n| \lesssim \frac{1}{M^{n-2}}. \tag{83}$$

Crucially, we have used that the extra terms denoted by the ellipsis in (82) are always subleading with respect to the field-theoretic and/or quantum-gravitational contribution.

Before moving on, let us stress that bounds of this form on Wilson coefficients are not strictly new. For instance, they are expected to hold from causality in the EFT [54]. Furthermore, sharp bounds along these lines have been found in [58–60, 63, 67, 68] using S-matrix

bootstrap techniques. It is appealing that the double EFT expansion proposal captures this general expectation. However, a proper comparison between (83) and the S-matrix bootstrap approach requires to carefully identify the meaning of the energy scale appearing in the latter. We postpone a more detailed account of these issues to Section 4.3.

Let us now discuss lower bounds on Wilson coefficients in terms of the EFT cutoff. First, notice that such a constraint can only be formulated if the Wilson coefficient is *assumed to be non-vanishing*. Indeed, in general we cannot exclude the possibility of having an exactly zero coefficient, perhaps due to some symmetry forbidding its presence in the Lagrangian. Moreover, even if this is the case and $\alpha_n \neq 0$, the double EFT expansion can provide a lower bound only if the quantum-gravitational or the field-theoretic expansions are assumed to contribute. Notice that we have described examples of Wilson coefficients with vanishing quantum-gravitational or field-theoretic contribution in Section 2.1. However, we have not been able to find a non-vanishing Wilson coefficient for which *both* the quantum-gravitational and the field-theoretic contribution are not present. Even though we do not have a clear-cut argument as to why this should always be the case, some compelling arguments on its favor were already presented in Section 2.3.2. This suggests that any non-trivial Wilson coefficient should be bounded by the smallest term within the double EFT expansion. Imposing the asymptotic regime (5) — $M \lesssim \Lambda_{\mathrm{QG}} \ll \mathcal{O}(1)$ in Planck units— this yields

$$\begin{aligned}
|\alpha_n| &\gtrsim \frac{1}{M^{n-d}}, &&\text{for} \quad n \leq d, \\
|\alpha_n| &\gg \mathcal{O}(1), &&\text{for} \quad n > d.
\end{aligned} \tag{84}$$

The first line comes from the field-theoretic term, which gives the weakest bound for $n \leq d$. On the other hand, for $n > d$, both terms are bounded from below by $(\Lambda_{\mathrm{QG}})^{d-n}$, thus yielding the second line upon imposing $\Lambda_{\mathrm{QG}} \ll \mathcal{O}(1)$.

Notice that, so far, we have not used any particular relation between $\Lambda_{\mathrm{QG}}$ and $M$ apart from the natural bound $\Lambda_{\mathrm{QG}} \gtrsim M$. Such an extra interdependence can be obtained when the scale $M$ is assumed to correspond to an infinite amount of new degrees of freedom (i.e., $M = M_{\mathrm{t}}$) and by imposing the Emergent String Conjecture [25]. As discussed in Section 1.1, this leads to the bound (7). Taking $p \to \infty$ (i.e., for a weakly-coupled string limit) yields the strongest condition, in fact saturating the complementary bound $\Lambda_{\mathrm{QG}} \gtrsim M_{\mathrm{t}}$. On the other hand, the weakest constraint is obtained for $p = 1$, i.e., when a single large extra dimension decompactifies in the asymptotic limit. From a bottom-up perspective, we should remain agnostic about which kind of limit we are exploring. Thus, we take the weakest condition implied by the Emergent String Conjecture, again given by $p = 1$. All in all, we obtain the following absolute bound on the species scale in terms of the mass scale of the lightest tower:

$$\Lambda_{\mathrm{QG}} \lesssim M_{\mathrm{t}}^{\frac{1}{d-1}}. \tag{85}$$

Imposing this, the lower bounds for non-vanishing Wilson coefficients coming from having non-zero quantum-gravitational and/or field-theoretic contributions become

$$\begin{aligned}
|\alpha_n| &\gtrsim \frac{1}{M_{\mathrm{t}}^{n-d}}, &&\text{if} \quad n \leq d+1, \\
|\alpha_n| &\gtrsim \frac{1}{M_{\mathrm{t}}^{\frac{n-2}{d-1}}}, &&\text{if} \quad n > d+1.
\end{aligned} \tag{86}$$

When compared to (84), we see that the inequality becomes stronger when $n > d$, i.e., for irrelevant operators. As advertised, the Emergent String Conjecture thus allows us to strengthen our bounds. In upcoming sections, we will also find other cases where this happens.

Before proceeding with our discussion, we should first clarify that (7) —and thus (85)— is known to hold for decompactifications to a flat space background and for tensionless string limits in moduli spaces of flat space vacua. On the other hand, the behavior of the tower of states can drastically change for decompactifications to running solutions [214] and for tensionless string limits in AdS spacetimes [38] (see [35–37,215] for further results on asymptotic regimes in AdS/CFT). A satisfactory working definition of $\Lambda_{\text{QG}}$ in terms of $M_t$ in those cases is still lacking. Nevertheless, a similar power-like bound as that shown in (85) is expected to hold in general. Our main goal is to show that the Emergent String Conjecture indeed leaves an imprint in our bounds. Should the power in (85) be modified in the future, it would be straightforward to update such a modification in our results.

Finally, notice that all the non-vanishing Wilson coefficients from top-down models discussed in previous sections satisfy the lower bound

$$|\alpha_n| \gtrsim \frac{1}{M_t^{n-d}}, \tag{87}$$

even when $n > d + 1$. For decompactification limits ($\Lambda_{\text{QG}} \gg M_t$), the reason is that the field-theoretic contribution turns out to be there for any non-vanishing Wilson coefficient. Some general arguments on favor of this observation were given in Section 2.2. On the other hand, we found examples of emergent string limits for which some Wilson coefficients did not have a field-theoretic contribution. Nevertheless, all those cases exhibit a non-vanishing quantum-gravitational contribution. Given that $\Lambda_{\text{QG}} \sim M_t$ for this type of limit, the Wilson coefficient still satisfies the bound above, and in fact it saturates the upper bound in (83).

Finally, let us notice that having an infinite number of Wilson coefficients satisfying (87) essentially encodes the fact that $M_t$ is a valid *EFT cutoff*. However, this does not imply that it should hold for any non-vanishing gravitational Wilson coefficient individually. It is thus remarkable that (87) is observed in all the examples considered so far. It would be important to clarify whether this is a general behavior in theories of quantum gravity or rather a lamppost effect.

## 4.2 Scaling relations between Wilson coefficients

In the previous section, we derived upper and lower (asymptotic) bounds on a given Wilson coefficient $\alpha_n$ as a function of the EFT cutoff $M$. However, the double EFT expansion actually contains more information than this. Since different Wilson coefficients should take the form in (82), it is possible to find scaling relations among them (possibly also including the EFT cutoff).

Due to the fact that we have a priori two scales in the expansion, it will be useful to use the parametrization

$$\Lambda_{\text{QG}} \sim M^\gamma, \qquad \text{with} \quad 0 < \gamma \leq 1. \tag{88}$$

The upper bound on $\gamma$ comes from the natural ordering of scales $M \lesssim \Lambda_{\text{QG}}$, while the lower bound is a consequence of working in the asymptotic regime, where both $\Lambda_{\text{QG}}, M \to 0$. As we saw in the previous section, these bounds can be strengthened when assuming $M = M_t$ and upon imposing the Emergent String Conjecture. In this case we can write down

$$\Lambda_{\text{QG}} \sim M_t^\gamma, \qquad \text{with} \quad \frac{1}{d-1} \leq \gamma \leq 1, \tag{89}$$

where the Emergent String Conjecture is encoded in the lower bound on $\gamma$. By plugging this parametrization into (82), one can trade the EFT cutoff scale by some non-vanishing Wilson coefficient. It is then possible to plug this back into combinations of other Wilson coefficients

(and possibly the EFT cutoff) to find scaling relations among each other. The latter will generically depend on $\gamma$, such that using the constraints in (88) or (89) one may get bounds on it.

A particularly simple class of constraints involve scaling relations between two different Wilson coefficients in Planck units in which the dependence on the EFT cutoff is completely eliminated. Since the cutoff does not appear in these bounds, they are well-suited for the primal S-matrix bootstrap approach employed in [57, 64]. We explore this type of bounds in this section, taking ten-dimensional maximal supergravity as an illustrative example.

In fact, restricting to maximally supersymmetric theories brings various advantages for us. First, it is the setup in which we tested explicitly our proposal in Section 2. Additionally, since both amplitudes and higher-curvature operators are highly constrained, it is the easiest one to further analyze using S-matrix bootstrap techniques. For instance, there are no operators allowed at the four and six derivatives level. Moreover, the only eight-derivative higher-curvature operator that one can write down is the famous $t_8 t_8 \mathcal{R}^4$. Remarkably, the S-matrix bootstrap results of [57, 64] imply that its Wilson coefficient, $\alpha_{\mathcal{R}^4}$, must be non-vanishing in $d = 9, 10, 11$! The next possible operator is the $D^4 \mathcal{R}^4$ that we discussed in Section 2.1. In what follows, we will obtain bounds on scaling relations among these two Wilson coefficients as outlined above.

Let us start with the $\mathcal{R}^4$ operator. Inserting $n = 8$ and $d = 10$ in (82), it should take the form

$$\alpha_{\mathcal{R}^4} = a_{\mathcal{R}^4} M^{-6\gamma} + b_{\mathcal{R}^4} M^2 + \cdots . \tag{90}$$

Notice that the field-theoretic term goes to zero as $M \to 0$. Thus, should the double EFT expansion capture the leading contribution to this Wilson coefficient in the asymptotic regime, having $a_{\mathcal{R}^4} = 0$ would be in contradiction with the order one lower bound for this Wilson coefficient found in [57, 64]. In other words, this Wilson coefficient *must* receive a quantum-gravitational contribution! This is a nice example of how S-matrix bootstrap results can inform the structure studied in this paper. Taking this into account, we have

$$\alpha_{\mathcal{R}^4} \sim M^{-6\gamma} , \tag{91}$$

where we are ignoring $a_{\mathcal{R}^4} \neq 0$ as an order one factor.[33] Notice that we must also have $\gamma \geq 0$ to be compatible with the bound in [57, 64], thus recovering the natural ordering of scales $\Lambda_{\text{QG}} \lesssim M_{\text{Pl},d}$ from the bottom-up.

The relation in (91) is key for the procedure outlined above. It allows us to trade $M$ for $\alpha_{\mathcal{R}^4}$ in the expression for another Wilson coefficient. For instance, consider the next higher-curvature correction in 10d maximal supergravity, given by the $D^4 \mathcal{R}^4$ operator discussed in Section 2.1. Using again equation (82), this time for $n = 12$ and $d = 10$, its Wilson coefficient satisfies

$$\alpha_{D^4 \mathcal{R}^4} = a_{D^4 \mathcal{R}^4} M^{-10\gamma} + b_{D^4 \mathcal{R}^4} M^{-2} + \cdots . \tag{92}$$

Following the same logic as when deriving (83), this is upper bounded by the largest of these two terms, thus yielding

$$\begin{aligned} |\alpha_{D^4 \mathcal{R}^4}| &\lesssim \alpha_{\mathcal{R}^4}^{5/3} , && \text{for} \quad \gamma \geq \frac{1}{5} , \\ |\alpha_{D^4 \mathcal{R}^4}| &\lesssim \alpha_{\mathcal{R}^4}^{1/(3\gamma)} , && \text{for} \quad \gamma < \frac{1}{5} . \end{aligned} \tag{93}$$

Therefore, imposing $\gamma \geq 0$ as in (88) leads to no bound at all. On the other hand, assuming $M = M_{\text{t}}$ and imposing the Emergent String Conjecture as in (89), we get

$$|\alpha_{D^4 \mathcal{R}^4}| \lesssim \alpha_{\mathcal{R}^4}^3 , \quad \text{as} \quad \alpha_{\mathcal{R}^4} \to \infty . \tag{94}$$

---

[33]We are also taking into account that this Wilson coefficient is non-negative (see, e.g., [57, 58, 67]).

Notice that the asymptotic regime, $M_t \to 0$, now corresponds to $\alpha_{\mathcal{R}^4} \to \infty$ as indicated. Finding this bound from bottom-up using S-matrix bootstrap would give model-independent evidence for the Emergent String Conjecture. As expected by having used the Emergent String Conjecture, this bound is saturated by the strong coupling limit of 10d Type IIA string theory, which leads to a decompactification to one dimension more (see Section 2.1).

As in Section 4.1, to derive lower bounds we need to assume the $D^4\mathcal{R}^4$ Wilson coefficient to be non-vanishing due to either the field-theoretic or the quantum-gravitational contribution. Let us study the bounds placed by these two terms separately. Upon substituting (91), the field-theoretic term satisfies

$$|\alpha_{D^4\mathcal{R}^4}| \gtrsim \alpha_{\mathcal{R}^4}^{1/(3\gamma)}, \quad \text{as} \quad \alpha_{\mathcal{R}^4} \to \infty. \tag{95}$$

Given $\gamma \leq 1$ as in both equations (88) and (89), we then get

$$|\alpha_{D^4\mathcal{R}^4}| \gtrsim \alpha_{\mathcal{R}^4}^{1/3}, \quad \text{as} \quad \alpha_{\mathcal{R}^4} \to \infty. \tag{96}$$

Interestingly, this bound can be derived without assuming $M = M_t$, since it rather comes from the natural ordering of scales $M \lesssim \Lambda_{QG}$. On the other hand, the quantum-gravitational term yields the stronger bound

$$|\alpha_{D^4\mathcal{R}^4}| \gtrsim \alpha_{\mathcal{R}^4}^{5/3}, \quad \text{as} \quad \alpha_{\mathcal{R}^4} \to \infty. \tag{97}$$

From the viewpoint of the double EFT expansion, we cannot predict which one of the two contributions should be non-vanishing. Hence, the best bound we can give is the weakest one, i.e., the one in (96). Nevertheless, we will see in the next section that another type of S-matrix bootstrap bound obtained in [67] implies that the $D^4\mathcal{R}^4$ Wilson coefficient should be lower-bounded by the quantum-gravitational contribution.[34] This is another example of how S-matrix bootstrap results can inform the double EFT expansion. Remarkably, in this case it even allows us to strengthen our bounds, leading to the stronger one displayed in eq. (97).

**Recovering string universality**

As we just discussed, the lower and upper bounds in (94) and (97) are saturated by the two different asymptotic limits in Type II string theory. Notice however that the double EFT expansion allows for everything in between. It is then interesting to ask under which assumptions we can recover string universality from the bottom-up. As we show now, there is indeed a natural way of achieving this by imposing the Emergent String Conjecture in a stronger way.

Recall that we are using the Emergent String Conjecture to place a lower bound on the species scale in terms of the mass of the lightest tower. As discussed below (7), this is due to possible subleading towers contributing to the species scale. In string theory examples, this happens along "mixed" limits that interpolate between the "pure" ones, for which the leading tower is the only one contributing to the species scale. Nevertheless, the moduli space of 10d maximal supergravity exhibits two disconnected infinite distance limits.[35] It is then natural to assume that these two limits should be "pure". Then, we can use the Emergent String Conjecture to fix

$$\gamma = \frac{p}{8+p}, \tag{98}$$

---

[34]Notice that this does not always imply $a_{D^4\mathcal{R}^4} \neq 0$. Indeed, this contribution vanishes in the strong coupling limit of Type IIA string theory (see Section 2.1). The correct statement would be that the quantum-gravitational contribution should be there whenever the field-theoretic is subleading with respect to it.

[35]Recall that the moduli space of these theories is completely fixed by supersymmetry.

in (89). Imposing that this Wilson coefficient should be lower-bounded by the quantum-gravitational contribution then yields

$$
\begin{aligned}
\alpha_{D^4\mathcal{R}^4} &\sim \alpha_{\mathcal{R}^4}^{5/3}\,, && \text{for} \quad p \geq 2\,, \\
\alpha_{D^4\mathcal{R}^4} &\sim \alpha_{\mathcal{R}^4}^{3}\,, && \text{for} \quad p = 1\,,
\end{aligned}
\tag{99}
$$

thus recovering the two behaviors that can be found in Type II string theory.

Notice however that, from the bottom-up perspective, the first one could equally correspond to a decompactification to 12 or more dimensions or to an emergent string limit. The reason is that the $D^4\mathcal{R}^4$ Wilson coefficient is dominated by the first term, and thus controlled by the species scale, for any $p \geq 2$. As we increase $p$, one needs to consider higher and higher-dimensional operators to be sensitive to the scale of the tower. More concretely, only if $n > 10 + p$ the Wilson coefficient is dominated by the term controlled by $M_\text{t}$. From an S-matrix bootstrap perspective, it thus seems more and more costly to tell apart emergent string limits from decompactifications of a high number of extra dimensions. On the other hand, it can be used to rule out different numbers of extra dimensions that can be decompactified starting from 10d maximal supergravity theories.

Finally, let us stress that the assumption of only having "pure" infinite distance limits only applies to a few setups. In most examples in string theory, the moduli space is multi-dimensional and contain several "mixed" infinite distance limits interpolating between the various pure ones. Thus, it seems hard to place bounds on the number of extra dimensions from bottom-up by constraining Wilson coefficients.

## 4.3   Comparison with S-matrix bootstrap bounds

In this section, we compare bounds derived from the double EFT expansion with those obtained with S-matrix bootstrap techniques in [58–60, 63, 67, 68]. As explained in these works, their methods naturally place bounds on the dimensionless Wilson coefficients[36]

$$
\tilde{\alpha}_n = \alpha_n M^{n-2}\,.
\tag{100}
$$

Ignoring order one factors, one can find two types of bounds in [58–60, 63, 67, 68]: order one upper bounds on a given $\tilde{\alpha}_n$, and relative bounds on two dimensionless Wilson coefficients as both of them go to zero. As we will see next, we are able to recover both kinds of bounds from the double EFT expansion.

Before proceeding, it is crucial to identify the scale appearing in S-matrix bootstrap studies in our language. As the notation in (100) suggests, we identify it with the scale of new degrees of freedom, $M$. For the S-matrix bootstrap, we consider a $d$-dimensional EFT with finite number of low-spin fields and valid up to some energy scale. As we discussed in Section 1.1, this energy scale corresponds to $M$ in the double EFT expansion. On the other hand, the scale appearing in the S-matrix bootstrap is also sometimes said to be the scale of the first higher-spin state. From this viewpoint, it is tempting to identify the latter with $\Lambda_\text{QG}$ instead. This is only valid if $M \sim \Lambda_\text{QG}$, otherwise the EFT breaks down well before reaching $\Lambda_\text{QG}$. One can UV-complete this theory by integrating in the new degrees of freedom but, crucially, this will affect the higher-curvature terms. In particular, the field-theoretic contributions controlled by $M$ will disappear. Furthermore, when the scale $M = M_\text{t}$ is associated with an infinite tower of states, this requires integrating in an infinite amount of degrees of freedom, thus falling outside of the assumption above of having a finite number of fields. As remarked, e.g., in [216, 217], integrating in a finite number of states in the tower is inconsistent Wilsonian sense due to the lack of scale separation among them.

---

[36]For comparison with their definitions, recall that we are working in Planck units where we have set $8\pi G_N = 1$.

We are now ready to derive bounds on $\tilde{\alpha}_n$ from the double EFT expansion. Let us start with the order one upper bounds on $|\tilde{\alpha}_n|$. These are the most generic kind of bounds that appear in S-matrix bootstrap studies. They have been obtained for any Wilson coefficient under consideration and in various spacetime dimensions. As we already noted in Section 4.1, our upper bounds in (83) precisely recover this type of constraints upon identifying the S-matrix bootstrap scale with $M$. Thus, the double EFT expansion correctly implements this general expectation for Wilson coefficients.

Naively identifying the scale in the S-matrix bootstrap bounds to be $\Lambda_{\text{QG}}$ leads to a parametric violation whenever a Wilson coefficient is dominated by the field-theoretic contribution, which can happen when $M \ll \Lambda_{\text{QG}}$. Nevertheless, if we correctly UV-complete the theory until its cutoff is of order $\Lambda_{\text{QG}}$, then the bound is no longer violated. As explained in Section 2, the field-theoretic term comes from integrating out the (towers of) states with mass of order $M$ at one-loop. Thus, these terms disappear from the Wilson coefficients when integrating in the (towers of) states. This leads to a new (possibly higher-dimensional) EFT for which the double EFT expansion has $M \sim \Lambda_{\text{QG}}$, such that the order one upper bound is satisfied. By this natural mechanism, the order one upper bounds on $|\tilde{\alpha}_n|$ are protected against integrating states in or out!

Let us now move to the relative bounds on two dimensionless Wilson coefficients as they go to zero. For concreteness, we consider ten-dimensional maximal supegravities as in Section 4.2. In this setup, this type of bounds were recently derived from S-matrix bootstrap for the $\mathcal{R}^4$ and $D^4\mathcal{R}^4$ Wilson coefficients [67]. In what follows, we study the latter from the perspective of the double EFT expansion and compare with the results obtained in [67].

The procedure is similar to that performed in Section 4.2, but considering now the dimensionless Wilson coefficients. For $\mathcal{R}^4$ we found the scaling in (91). Recalling the definition shown in (100), for the dimensionless counterpart we then have

$$\tilde{\alpha}_{\mathcal{R}^4} \sim M^{6(1-\gamma)}. \tag{101}$$

Similarly, for the $D^4\mathcal{R}^4$ Wilson coefficient we found (92), which yields

$$\tilde{\alpha}_{D^4\mathcal{R}^4} = a_{D^4\mathcal{R}^4} M^{10(1-\gamma)} + b_{D^4\mathcal{R}^4} M^8 + \cdots. \tag{102}$$

Notice that, for $\gamma < 1$, both dimensionless Wilson coefficients go to zero as $M \to 0$. As advertised, we are then able to put relative bounds in this limit.

As in previous sections, we can place lower bounds on $\tilde{\alpha}_{D^4\mathcal{R}^4}$ by assuming that it is non-vanishing due to the field-theoretic and/or the quantum-gravitational contribution. In particular the Wilson coefficient is lower bounded by the smallest of the two contributions. Trading $M$ by $\tilde{\alpha}_{\mathcal{R}^4}$ using (101), the field-theoretic term yields

$$|\tilde{\alpha}_{D^4\mathcal{R}^4}| \gtrsim \tilde{\alpha}_{\mathcal{R}^4}^{4/(3(1-\gamma))}, \quad \text{as} \quad \tilde{\alpha}_{\mathcal{R}^4} \to 0. \tag{103}$$

Since the above constraint is trivialized as $\gamma \to 1$, we find no bound using the double EFT expansion. This situation is improved if $\tilde{\alpha}_{\mathcal{R}^4}$ is lower bounded by the quantum-gravitational contribution. Indeed, this leads to

$$|\tilde{\alpha}_{D^4\mathcal{R}^4}| \gtrsim \tilde{\alpha}_{\mathcal{R}^4}^{5/3}, \quad \text{as} \quad \tilde{\alpha}_{\mathcal{R}^4} \to 0. \tag{104}$$

Remarkably, we recover the S-matrix bootstrap bound in [67]. Thus, as advertised in Section 4.2, the results of [67] imply that the $D^4\mathcal{R}^4$ Wilson coefficient should be lower-bounded by the quantum-gravitational term. Similarly to what happened with the $\mathcal{R}^4$ Wilson coefficient and the bounds in [57,64], S-matrix bootstrap informs the double EFT expansion and allows us to strengthen our bounds.

Let us now move to the upper bounds on $\tilde{\alpha}_{D^4\mathcal{R}^4}$ in terms of $\tilde{\alpha}_{\mathcal{R}^4}$. Noting that (102) is bounded from above by the largest of the two terms and trading $M$ by $\tilde{\alpha}_{\mathcal{R}^4}$ using (101), we find

$$
\begin{aligned}
|\tilde{\alpha}_{D^4\mathcal{R}^4}| &\lesssim \tilde{\alpha}_{\mathcal{R}^4}^{5/3}\,, && \text{for} \quad \gamma \geq \frac{1}{5}\,, \\
|\tilde{\alpha}_{D^4\mathcal{R}^4}| &\lesssim \tilde{\alpha}_{\mathcal{R}^4}^{4/(3(1-\gamma))}\,, && \text{for} \quad \gamma < \frac{1}{5}\,.
\end{aligned}
\tag{105}
$$

Unlike for the dimensionful Wilson coefficients, since the dimensionless ones are going to zero in the limit, $\gamma \geq 0$ yields the non-trivial bound

$$
|\tilde{\alpha}_{D^4\mathcal{R}^4}| \lesssim \tilde{\alpha}_{\mathcal{R}^4}^{4/3}\,, \quad \text{as} \quad \tilde{\alpha}_{\mathcal{R}^4} \to 0\,.
\tag{106}
$$

This bound can be however strengthened by assuming $M = M_{\text{t}}$ and using the Emergent String Conjecture, which imposes $\gamma > 1/9$. Indeed, we obtain

$$
|\tilde{\alpha}_{D^4\mathcal{R}^4}| \lesssim \tilde{\alpha}_{\mathcal{R}^4}^{3/2}\,, \quad \text{as} \quad \tilde{\alpha}_{\mathcal{R}^4} \to 0\,.
\tag{107}
$$

In comparison to [67], their S-matrix bootstrap approach provided the weaker bound

$$
|\tilde{\alpha}_{D^4\mathcal{R}^4}| \lesssim \tilde{\alpha}_{\mathcal{R}^4}\,, \quad \text{as} \quad \tilde{\alpha}_{\mathcal{R}^4} \to 0\,.
\tag{108}
$$

Thus, we see that the double EFT expansion can impose stronger constraints than what have been found so far using S-matrix bootstrap. As discussed in [67], there is a particular reason why their analysis cannot yield stronger bounds than this one. Given two consistent values for $\tilde{\alpha}_{\mathcal{R}^4}$ and $\tilde{\alpha}_{D^4\mathcal{R}^4}$, any (positive) linear combination of them gives a four-point supergraviton scattering that satisfies their constraints. Going beyond this bound seems like an outstanding challenge for the S-matrix bootstrap program, which will probably require exploring higher-point scattering amplitudes or perhaps mixed scattering systems. We hope that the bounds we found above can provide further motivation to tackle this problem, since improving these S-matrix bootstrap bounds could be used to test both the double EFT expansion and the Emergent String Conjecture from bottom-up.

## 5 Outlook

In this work, we have motivated and introduced the *double EFT Expansion*, an organizational framework for higher-derivative corrections in gravitational effective field theories. The proposal sorts them into two different low-energy expansions: the *field-theoretic expansion*, controlled by the mass scale $M$ of the lightest (tower of) particles, and the *quantum-gravitational expansion*, governed by the species scale $\Lambda_{\text{QG}}$, which acts as a quantum gravity cutoff. This dual structure provides a systematic way to distinguish quantum gravitational effects from those arising due to intermediate massive field-theoretic states, offering novel insights into the interplay of UV and IR physics in quantum gravity.

Building on previous efforts [45–53], we have provided top-down evidence for the validity of this structure by inspecting various string theory models. These include toroidal compactifications of M-theory (Section 2) and Calabi–Yau compatifications of F-theory (Appendix B.1), M-theory (Appendix B.2), and Type II string theories (Section 3.1). Furthermore, in Section 4 we derived bounds on Wilson coefficients using the double EFT expansion and verified their compatibility with current bottom-up S-matrix bootstrap results [54, 57–60, 63, 64, 67]. From the top-down, it would be interesting to further test our proposal across the quantum gravity landscape, including other types of higher-derivative corrections and string theory models with less amount of supersymmetry. Similarly, it would be valuable to improve on our comparison with S-matrix bootstrap bounds, as well as to include other bottom-up approaches.

Apart from this, our work leaves several other interesting open questions. First of all, Section 2 was devoted to an amplitudes perspective. In particular, we focused on four-point graviton scattering in ten-dimensional string theory and toroidal compactifications of maximal supergravity theories. For all the Wilson coefficients considered in Section 2.2.1, we found the field-theoretic term to be non-vanishing. Hence, even though less supersymmetric models are expected to lead to fewer cancellations, it would be relevant to verify or disprove this observation in more general setups. Moreover, in order to complement the arguments presented in Section 2.2.2, it would be valuable to get more insights from studying higher-point scattering amplitudes. In Section 2.3.2, we discussed the presence of extra terms that do not fit into the double EFT expansion but still seem to never yield the leading-order behavior of any Wilson coefficient in the asymptotic regime. Understanding their underlying structure is yet another interesting direction for future research. In particular, it would be important to rigorously confirm that the double EFT expansion contains the leading contribution to any non-vanishing Wilson coefficient in the asymptotic regime, as observed in all top-down examples analyzed herein.

We changed gears in Section 3 and considered another rich gravitational observable, namely the black hole entropy. In particular, we studied the effect of the higher-derivative corrections encoded in the double EFT expansion to BPS extremal black holes in 4d $\mathcal{N} = 2$ theories arising from Calabi–Yau compactifications of Type II string theory. In Section 3.2, we analyzed in detail these effects on black holes exploring the large volume limit, which leads to a decompactification to M-theory on the same threefold, via the attractor mechanism. It would be interesting to extend our results to other (infinite distance) limits within the vector multiplet moduli space of these theories, such as decompactifications to F-theory or weakly-coupled string limits. This will be partially studied in the upcoming work [218]. Furthermore, notice that the higher-derivative corrections that played a most prominent role in our analysis vanish exactly in more supersymmetric setups. Thus, it seems that the entropy index of BPS black holes would not be sensitive to the tower/Kaluza-Klein scale whenever more supersymmetry is present. Finding another related black hole observable that could capture the presence of extra dimensions is also left for future work.

Finally, in Section 4 we adopted a bottom-up perspective. We studied some of the bounds on gravitational Wilson coefficients that naturally follow from the double EFT expansion — whenever it encodes the leading contribution in the asymptotic limit— and compared them with current S-matrix bootstrap constraints. Remarkably, some of these bounds are strengthened by imposing the Emergent String Conjecture [25], which opens up the exciting possibility of testing this conjecture using the S-matrix bootstrap approach. The upper bounds discussed in Section 4.1 are generally expected to be implied by causality and unitarity, while the lower bounds are more elusive for S-matrix bootstrap studies. In this regard, it is interesting to observe that the lower bound displayed in (87) reads

$$|\tilde{\alpha}_n| = |\alpha_n| M^{n-2} \gtrsim G_N M^{d-2} \,, \tag{109}$$

upon re-expressing it for the dimensionless Wilson coefficient and restoring Newton's constant. Notice that the right hand side looks precisely like a one-loop gravitational effect. This might suggest that, perhaps under particular circumstances, the strategy followed by [58–60, 63, 67, 68] could recover this type of bounds when considering the amplitude at the one-loop level. In Section 4.2, we constrained scaling relations between the $\mathcal{R}^4$ and $D^4\mathcal{R}^4$ Wilson coefficients in 10d maximally supersymmetric theories. From the S-matrix bootstrap perspective, the approach of [57, 64] seems particularly well-suited to explore this kind of relations. Finally, in Section 4.3 we discussed similar bounds on the dimensionless Wilson coefficients associated to $\mathcal{R}^4$ and $D^4\mathcal{R}^4$, and we contrasted them with the results recently obtained in [67]. Despite the advanced technical difficulties, it would be valuable to improve

on the bound in (108), as it could lead to a very non-trivial bottom-up test of the double EFT expansion and the Emergent String Conjecture (cf. eqs. (106) and (107)).

On a perhaps more broader context, this work aims to improve our understanding about the space of consistent, UV-complete theories of quantum gravity by providing valuable insights on the structure of higher-derivative corrections to generic gravitational EFTs. We hope to report on the open questions raised above in the future and that our work motivates others to explore these and other related research directions.

# Acknowledgments

We are indebted to Ivano Basile, Matilda Delgado, Damian van de Heisteeg, Luis Ibáñez, Christian Kneißl, Dieter Lüst, Fernando Marchesano, Carmine Montella, Miguel Montero, Antonia Paraskevopoulou, Ignacio Ruiz, Savdeep Sethi, Ángel Uranga, Cumrun Vafa, Irene Valenzuela, Max Wiesner and Alexander Zhiboedov for illuminating discussions and very useful feedback. A.C. thanks Matteo Zatti for collaboration on related topics. The authors would like to acknowledge IFT-Madrid for hospitality and support during the different stages of this work. A.C. is also thankful to the Department of Physics at Harvard University for hospitality during the early stages of this work. J.C. and A.H. wish to thank the Erwin Schrödinger International Institute for Mathematics and Physics for their hospitality during the programme "The Landscape vs. the Swampland". J.C., A.C. and A.H. are also grateful to Marta Igarzabal, Teresa Lobo and Raquel Santos for continuous encouragement and support.

**Funding information**   The work of A.C. is supported by a Kadanoff and an Associate KICP fellowships, as well as through the NSF grants PHY-2014195 and PHY-2412985.

# A   Details on the 10d massive threshold resummation

The purpose of this appendix is to illustrate in a explicit example how the field-theoretic terms of the form (2) associated to an infinite tower of Kaluza-Klein modes can be resummed for energies well above the characteristic mass $M_t$, and in fact disappear from the Wilsonian effective action whilst becoming part of the massless thresholds in the decompactified theory. We follow closely the treatment in [104, 106].

To start with, and building on the discussion in Section 2, we consider 10d Type IIA string theory and study the four-point graviton amplitude (8) in the strong coupling regime, i.e., for $g_s \gg 1$. The latter can be effectively computed from 11d M-theory, where the contribution we are interested in here arises from a one-loop calculation that yields [100, 102]

$$\mathcal{A}_4(s,t) = \hat{K}\left[\mathcal{I}(S,T) + \mathcal{I}(S,U) + \mathcal{I}(U,T)\right], \tag{A.1}$$

with $\hat{K}$ being the kinematical factor associated to a specific contraction of four Weyl tensors, cf. eq. (9). We have chosen the notation such that $\{S, T, U\}$ are the Mandelstam invariants of eleven-dimensional supergravity,[37] whereas $\mathcal{I}(S,T)$ denotes the Feynman integral associated

---

[37] The relation between the 11d Mandelstam variables and the analogous quantities in 10d Type IIA string theory (cf. discussion below (8)) reads as follows

$$\ell_s^2 s = S\,\frac{\ell_{\text{Pl},11}^2}{2\pi R_{11}}\,, \qquad \ell_s^2 t = T\,\frac{\ell_{\text{Pl},11}^2}{2\pi R_{11}}\,, \qquad \ell_s^2 u = U\,\frac{\ell_{\text{Pl},11}^2}{2\pi R_{11}}\,, \tag{A.2}$$

where $R_{11}$ denotes the radius of the M-theory circle in units of $\ell_{\text{Pl},11}$.

to a box diagram in an auxiliary (massless) scalar $\varphi^3$ theory, of the form

$$\mathcal{I}(S,T) = \int d^{11}p \, \frac{1}{p^2} \frac{1}{\left(p + k^{(1)}\right)^2} \frac{1}{\left(p + k^{(1)} + k^{(2)}\right)^2} \frac{1}{\left(p + k^{(1)} + k^{(2)} + k^{(3)}\right)^2} \,, \qquad (A.3)$$

where the external momenta are given by $k^{(r)}$, $r = 1, \ldots, 4$. These are moreover subject to the on-shell and momentum conservation conditions $(k^{(r)})^2 = 0$ and $\sum_r k^{(r)} = 0$, respectively. Therefore, in order to make contact with the 10d theory, we need, first of all, to consider the spacetime background $\mathbb{R}^{1,9} \times \mathbf{S}^1$, such that the loop momentum along the compact direction is now quantized in terms of (integer) Kaluza-Klein charges. Additionally, we must take the external gravitons to be massless and completely polarized along the non-compact $\mathbb{R}^{1,9}$ component. This leads to the following Feynman integral

$$\mathcal{I}(S,T) = \frac{1}{2\pi R_{11} \ell_{\text{Pl},11}} \int \prod_{r=1}^{4} d\tau_r \int d^{10}p \sum_{n \in \mathbb{Z}} e^{-(R_{11}\ell_{\text{Pl},11})^{-2} n^2 \tau - \sum_{r=1}^{4} p_r^2 \tau} \,, \qquad (A.4)$$

with $\ell_{\text{Pl},11}^9 = 4\pi \kappa_{11}^2$ being the 11d Planck length and $\tau_r$ denote the corresponding Schwinger parameters. Note that we have defined the quantities $\tau = \sum_{r=1}^{4} \tau_r$, and $p_r = p + \sum_{s=1}^{r} k^{(s)}$. Furthermore, following the analysis in refs. [100,101] one can show that eq. (A.4) reduces to

$$\mathcal{I}(S,T) = \frac{2\pi^5}{2\pi R_{11} \ell_{\text{Pl},11}} \int \frac{d\tau}{\tau^2} \int_{\mathscr{T}_{ST}} \prod_{r=1}^{3} d\omega_r \sum_{n \in \mathbb{Z}} e^{-(R_{11}\ell_{\text{Pl},11})^{-2} n^2 \tau - Q(S,T; \omega_r) \tau} \,, \qquad (A.5)$$

where $Q(S,T; \omega_r) = -S\omega_1(\omega_3 - \omega_2) - T(\omega_2 - \omega_1)(1 - \omega_3)$ and the domain of integration is defined to be $\mathscr{T}_{ST} = \{0 \leq \omega_1 \leq \omega_2 \leq \omega_3 \leq 1\}$. Interestingly, it turns out that the remaining contributions appearing in (A.3) may be written similarly but with a different integration domain, namely $\mathscr{T}_{TU} = \{0 \leq \omega_3 \leq \omega_2 \leq \omega_1 \leq 1\}$ and $\mathscr{T}_{SU} = \{0 \leq \omega_2 \leq \omega_1 \leq \omega_3 \leq 1\}$, respectively.[38]

Crucially, the integral (A.5) can be separated into a momentum-independent contribution, which arises from the constant term in $\exp(-Q(S,T; \omega_r)\tau)$ and is ultimately related to the $t_8 t_8 \mathcal{R}^4$ operator (cf. eq. (12a)), as well as a second piece that captures the corrections to the 10d effective action of the form $D^{2\ell} \mathcal{R}^4$. The latter reads formally as

$$\mathcal{I}'(S,T) \equiv \frac{2\pi^5}{2\pi R_{11} \ell_{\text{Pl},11}} \int \frac{d\tau}{\tau^2} \int_{\mathscr{T}_{ST}} \prod_{r=1}^{3} d\omega_r \left(e^{-Q(S,T; \omega_r)\tau} - 1\right) \sum_{n \in \mathbb{Z}} e^{-(R_{11}\ell_{\text{Pl},11})^{-2} n^2 \tau} \,. \qquad (A.6)$$

In what follows, we will focus on the subset of corrections determined by (A.6), since those are the ones we are most interested in here. Notice that this includes both the non-analytic contribution to the amplitude due to intermediate massless states running in the loop (i.e., the terms with zero Kaluza-Klein momentum), as well as the massive thresholds. The former can be identified with the general expression for the massless one-loop correction in 11d supergravity compactified in $\mathbf{T}^{11-d}$, and takes the form [104]

$$\mathcal{I}'_{(n=0)}(S,T) = \frac{2\pi^{d/2}}{\ell_{\text{Pl},11}^{11-d} \mathcal{V}_{11-d}} \Gamma(4 - d/2) \int_{\mathscr{T}_{ST}} \prod_{r=1}^{3} d\omega_r \, [Q(S,T; \omega_r)]^{\frac{d-8}{2}} \,, \qquad (A.7)$$

where $\mathcal{V}_{11-d}$ denotes the overall internal volume in units of $\ell_{\text{Pl},11}$. Notice that the $d \to 10$ limit is logarithmically divergent due to the pole in the $\Gamma$-function. Thus, after regularization (and

---

[38]Note that, strictly speaking, the integral (A.5) converges only in the unphysical region $S, T < 0$, where it can be readily evaluated. This allow us to obtain the physical amplitude by analytic continuation from the latter result, which is also a familiar trick frequently used when computing string theory amplitudes [103].

upon appropriately choosing the scale inside the logarithm) it can be written as [103, 104, 107]

$$\mathcal{I}'_{(n=0)}(S,T) = \frac{2\pi^5}{2\pi R_{11}\ell_{\mathrm{Pl},11}}\left(-\mathscr{G}(S,T)\right)\left(\ln(-\mathscr{G}(S,T))-2\right), \tag{A.8}$$

where [104]

$$\mathscr{G}^n(S,T) = \int_{\mathscr{T}_{ST}} \prod_{r=1}^{3} \mathrm{d}\omega_r \; [-Q(S,T;\omega_r)]^n . \tag{A.9}$$

Notice that this has precisely the same structure as the leading, non-analytic, massless threshold displayed in (31).

The massive thresholds, on the other hand, are easily seen to be convergent both in the UV and the IR regimes. In fact, upon expanding the exponential in (A.6) one arrives at[39]

$$\mathcal{I}'_{(n\neq0)}(S,T) = \sum_{\ell=2} \mathcal{I}'_{\ell}(S,T), \tag{A.10}$$

with

$$\begin{aligned}\mathcal{I}'_{\ell}(S,T) &= \frac{2\pi^5}{2\pi R_{11}\ell_{\mathrm{Pl},11}} \frac{\mathscr{G}^{\ell}(S,T)}{\ell!} \int_0^\infty \frac{\mathrm{d}\tau}{\tau} \tau^{\ell-1} \sum_{n\neq0} e^{-(R_{11}\ell_{\mathrm{Pl},11})^{-2}n^2\tau} \\ &= \frac{2\pi^5}{2\pi R_{11}\ell_{\mathrm{Pl},11}} \frac{\mathscr{G}^{\ell}(S,T)}{\ell!} 2\Gamma(\ell-1)\zeta(2\ell-2) M_{\mathrm{D0}}^{2-2\ell} ,\end{aligned} \tag{A.11}$$

where $M_{\mathrm{D0}} = (R_{11}\ell_{\mathrm{Pl},11})^{-1} = 2\pi(g_s\ell_s)^{-1}$ [89]. Note that upon summing over the three different contributions in the box diagram (A.3), the Kaluza-Klein thresholds are given by symmetric homogeneous polynomials of order $\ell$ in the variables $\{S,T,U\}$, cf. eq. (12).

With this, we are now ready to discuss how the field-theoretic corrections to the graviton four-point amplitude —as computed through (A.6)— resum in the strong coupling limit to give the massless contribution of the 11d superparticle displayed in (A.7). Indeed, upon adding together (A.8) and (A.11), using the defining series for the zeta function, and after performing a resummation of the index $\ell$, one finally arrives at

$$\mathcal{I}'(S,T) = \frac{2\pi^5}{2\pi R_{11}\ell_{\mathrm{Pl},11}^3} \sum_{k\in\mathbb{Z}} \left(\frac{k^2}{R_{11}^2} - \ell_{\mathrm{Pl},11}^2 \mathscr{G}(S,T)\right)\left[\ln\left(1 - \frac{R_{11}^2}{k^2}\ell_{\mathrm{Pl},11}^2 \mathscr{G}(S,T)\right) - 2\right], \tag{A.12}$$

which for the limit $R_{11} \to \infty$ can be traded by an integral over the continuous variable $x = k/R_{11}$ as follows

$$\begin{aligned}\mathcal{I}'(S,T) &= \frac{4\pi^5}{2\pi\ell_{\mathrm{Pl},11}^3} \int_0^\infty \mathrm{d}x \left(x^2 - \ell_{\mathrm{Pl},11}^2 \mathscr{G}(S,T)\right)\left[\ln\left(1 - \frac{\ell_{\mathrm{Pl},11}^2}{x^2}\mathscr{G}(S,T)\right) - 2\right] \\ &= \frac{4\pi^5}{3}\left(-\mathscr{G}(S,T)\right)^{3/2} .\end{aligned} \tag{A.13}$$

In order to obtain the final, finite result we have appropriately regularized the infrared regime $x \to \infty$. Notice that this indeed agrees with the massless threshold correction computed directly via eq. (A.7) for the particular case of $d = 11$.

---

[39] Notice that the $\ell = 1$ term in (A.10) is absent since it does not contribute to $\mathcal{A}_4(s,t)$ when added together with the analogous contributions coming from $\mathcal{I}(T,U)$ and $\mathcal{I}(U,T)$, as per the on-shell condition $S + T + U = 0$. This also explains, from the dual 11d perspective, why the series of corrections of the form $D^{2\ell}\mathcal{R}^4$ starts directly from $\ell = 2$, cf. eq. (11).

# B  The double EFT expansion in theories with eight supercharges

In this appendix, we consider 6d $\mathcal{N} = (1,0)$ and 5d $\mathcal{N} = 1$ supergravity theories arising from Calabi–Yau threefold compactifications of F-theory and M-theory, respectively. Our discussion builds on the analysis performed in [50, Section 5], where it was shown that the leading-order contribution to the first non-trivial (supersymmetry-preserving) higher-curvature correction follows the behavior predicted by (3). The aim here will consist in supplementing these results by keeping track of subleading terms as well, which as we will see can be identified with the field-theoretic contribution displayed in (2). Therefore, we argue in the following that the aforementioned setups also provide further evidence to the double EFT expansion proposal presented herein.

## B.1  6d $\mathcal{N} = (1,0)$ supergravity

Let us start with minimal supersymmetric models in six dimensions, which can be obtained, e.g., from compactifying F-theory on an elliptically fibered Calabi–Yau threefold $\pi : X_3 \to B_2$ [219, 220]. Similarly to what happens in the 4d $\mathcal{N} = 2$ theories discussed in Section 3.1, the moduli space of the resulting 6d EFT factorizes —at the two-derivative level— between that corresponding to the tensor and hypermultiplets. In what follows we will focus on the former, since their scalar fields appear to control certain protected, higher-curvature operators appearing in the gravitational sector.

Hence, let us recall that a tensor multiplet in 6d $\mathcal{N} = (1,0)$ supergravity is comprised by a dynamical anti-self-dual 2-form field $\mathsf{B}_2$, with field strength[40]

$$\mathsf{G}_3 = d\mathsf{B}_2 + (\text{Chern-Simons terms}), \tag{B.1}$$

together with one Weyl fermion and a real scalar. Upon compactifying F-theory on an elliptic threefold, one obtains a $n_T = h^{1,1}(B_2) - 1$ dimensional tensor multiplet moduli space, which can be locally parametrized by a set of inhomogeneous coordinates $j^\alpha$, $\alpha = 1, \dots, h^{1,1}(B_2)$. The latter moreover describe the scalar coset [222]

$$\mathcal{M}_{\mathrm{T}} \cong \frac{SO(1, n_T)}{SO(n_T)}, \tag{B.2}$$

with $SO(1, n_T)$ having mostly minus signature. As a consequence, its geometric data can be completely encoded into two symmetric, rank-2 tensors, namely

$$\Omega_{\alpha\beta}, \qquad g_{\alpha\beta} = j_\alpha j_\beta - \Omega_{\alpha\beta}, \tag{B.3}$$

where $\Omega_{\alpha\beta}$ is a constant tensor with Lorentzian signature $(1, n_T)$ whilst $g_{\alpha\beta}$ is moduli-dependent and positive definite, with $j_\alpha = \Omega_{\alpha\beta} j^\beta$. Furthermore, the scalars are subject to the restriction[41]

$$\frac{1}{2} \Omega_{\alpha\beta} j^\alpha j^\beta \overset{!}{=} 1. \tag{B.5}$$

---

[40]The precise form of the Chern-Simons terms appearing in (B.1) can be deduced directly from anomaly cancellation in 6d $\mathcal{N} = 1$ supergravity, see, e.g., [221, Appendix B].

[41]Geometrically, the constraint (B.5) may be interpreted as the constant (unit) volume hypersurface, since the $j^\alpha$ are related to the expansion coefficients of the Kähler form of the twofold base $B_2$ —in an appropriate integral basis $\{\omega_\alpha\} \in H^{1,1}(B_2)$— as follows

$$J_{B_2} = X^\alpha \omega_\alpha = (\mathcal{V}_{B_2}^{1/2} j^\alpha) \omega_\alpha, \tag{B.4}$$

with $\mathcal{V}_{B_2} = \frac{1}{2} \Omega_{\alpha\beta} X^\alpha X^\beta$.

All in all, the relevant bosonic (pseudo-)action for the 6d theory under consideration can be written as follows [221–223]

$$S_{6d} \supset \frac{1}{2\kappa_6^2} \int \mathcal{R} \star 1 - \frac{1}{2} \left( g_{\alpha\beta} \, dj^\alpha \wedge \star dj^\beta + g_{\alpha\beta} \, \mathsf{G}^\alpha \wedge \star \mathsf{G}^\beta \right), \tag{B.6}$$

where it should be understood that the scalar fields satisfy the quadratic constraint (B.5), whilst the 2-forms $\mathsf{B}^\alpha$ must be supplemented with the (anti-)self-duality conditions

$$g_{\alpha\beta} \star \mathsf{G}^\beta = \Omega_{\alpha\beta} \mathsf{G}^\beta. \tag{B.7}$$

Notice that, strictly speaking, the set of 2-forms $\mathsf{B}^\alpha$ contains one additional potential which actually belongs to the gravity multiplet and moreover satisfies a self-duality restriction, as per (B.7).

Beyond two-derivatives, there are multiple terms that can enter the effective supergravity action. Particularly interesting are those which can be argued to be present via anomaly cancellation, and supersymmetric partners thereof. Among those, the most relevant one for us will be the following four-derivative, higher-curvature operator (see, e.g., [224] for details and conventions)

$$S_{6d} \supset \frac{1}{2\kappa_6} \int d^6x \left( \frac{1}{32(2\pi)^{10/3}} c_{1,\alpha} j^\alpha \right) \mathrm{Tr} \, \mathcal{R}_2 \wedge \mathcal{R}_2, \tag{B.8}$$

where

$$(\mathcal{R}_2)^a{}_b = \frac{1}{2} e^a{}_\sigma e_b{}^\rho \mathcal{R}^\sigma{}_{\rho\mu\nu} dx^\mu \wedge dx^\nu, \tag{B.9}$$

denotes the curvature 2-form, $e^a{}_\mu$ is the spacetime vielbein, and $c_{1,\alpha}$ are the components of the first Chern class of the base $B_2$ in a certain basis of $H^{2,2}(B_2)$.

Additionally, as shown in [15, 16, 24] there exists just one possible kind of infinite distance/weak coupling point in the tensor multiplet moduli space. Notice that this also follows from the coset structure (B.2). As demonstrated in the aforementioned references, these limits are characterized by having some effective divisor class $[D_0]$ with vanishing self-intersection, such that the Kähler form of the base $B_2$ can always be written as [15, 16]

$$J_{B_2} = X^0[D_0] + X^i[D_i], \qquad i = 1, \dots, h^{1,1}(B_2) - 1, \tag{B.10}$$

where the $X^\alpha$ scale with the infinite distance parameter $\lambda \to \infty$ in a way such that

$$X^0 = \lambda, \qquad X^i = \frac{x^i}{\lambda}, \qquad \mathcal{V}_{B_2} = x^i[D_0] \cdot [D_i] + \mathcal{O}(\lambda^{-2}), \tag{B.11}$$

with $x^i$ some finite, positive definite constants. Thus, as already mentioned, this implies that the divisor dual to $[D_0]$ admits a holomorphic representative whose volume vanishes along the limit as follows

$$\mathcal{V}_{D_0} = \frac{x^i}{\lambda}[D_0] \cdot [D_i] \sim \frac{\mathcal{V}_{B_2}}{\lambda}, \qquad \text{as } \lambda \to \infty. \tag{B.12}$$

Consequently, there exists a dual emergent, weakly coupled string that arises upon wrapping a D3-brane on the shrinking cycle, and whose microscopic nature depends on whether the intersection product between $[D_0]$ and $c_1(B_2)$ is equal to 0 or 2. In the former case, the resulting extended object is of type II whereas for the latter one obtains an heterotic fundamental string. In the sequel, we discuss each scenario in turn and use the resulting moduli dependence of the higher-curvature operator (B.8) to infer properties about the double EFT expansion (4) in the corresponding theory.

Therefore, let us consider first the case where $c_1(B_2) \cdot [D_0] = 2$. Upon using the moduli identification (B.4) and setting $M_{\text{Pl},6} = 1$, we find (cf. eq. (6))

$$\alpha_4 = \frac{a_4}{M_s^2} + \frac{b_4}{M_s^{-2}}, \qquad \text{with } a_4 = \frac{1}{32(2\pi)^{10/3}}, \quad b_4 = \frac{c_{1,i}\, x^i}{64(2\pi)^{10/3}\mathcal{V}_{B_2}}, \qquad \text{(B.13)}$$

for the associated Wilson coefficient, where $\Lambda_{\text{QG}} = M_{\text{t}} = M_s \sim \mathcal{V}_{B_2}^{1/4}/\lambda^{1/2}$ denotes the mass scale of the emergent heterotic string in 6d Planck units. Hence, the first term in (B.13) can be identified with a tree-level contribution in the dual heterotic frame, whereas the second can be seen to arise at one-loop order. Notice that this indeed agrees with the general expectation coming from the double EFT expansion, where we see explicitly that both the *quantum-gravitational* and the *field-theoretic* corrections arise for the particular operator under consideration.

On the other hand, in the alternative case of an emergent dual Type II string, where $c_1(B_2) \cdot [D_0]$ vanishes, one realizes that now $a_4 = 0$, hence implying the absence of the quantum-gravitational piece [50]. This does not mean, necessarily, that the Wilson coefficient $\alpha_4$ is absent of content, since a priori the field-theoretic contribution would be non-zero unless the first Chern class of the base vanishes identically, which only happens when the fibration is trivial, such that the theory preserves a higher amount of supersymmetry.

## B.2   5d $\mathcal{N} = 1$ supergravity

We now move to consider 5d $\mathcal{N} = 1$ supergravity theories arising from M-theory compactified on a Calabi–Yau threefold. At the two-derivative level, the moduli space factorizes into vectors and hypermultplets. We will focus on the former, since they control a certain supersymmetry-protected higher-curvature correction appearing in the effective action.

The vector multiplet moduli space of these theories is spanned by the Kähler moduli of the compactification manifold, $X^I$ with $I = 1, \ldots, h^{1,1}$, subject to the constraint of fixed overall volume. Using $\mathcal{K}_{IJK}$ to denote the triple interesection numbers of the Calabi–Yau, this condition reads

$$\mathcal{V} = \frac{1}{6}\mathcal{K}_{IJK}X^I X^J X^K = \text{const.} \qquad \text{(B.14)}$$

In [25] (see also [225]), it was shown that any asymptotic limit within the vector multiplet moduli spaces corresponds to a two- or four-dimensional fiber shrinking to zero size, whilst the base blows up so as to keep the threefold volume fixed. The first one corresponds to a torus fibration and leads to a decompactification to six-dimensions (back to F-theory, cf. Section B.1). The second one involves either an abelian or a K3 fibration, leading to a heterotic or Type IIB emergent string limit, respectively. In both cases, the fibration becomes adiabatic along the asymptotic limit, such that we can approximate the volume of the Calabi–Yau to be given by that of the fiber times the base. Hence, upon assuming —as is customarily done— that the four-dimensional part of the geometry is blowing up or shrinking homogeneously, for any asymptotic limit we can then write

$$\mathcal{V} \sim XY^2 + \cdots, \qquad \text{(B.15)}$$

where we are ignoring some numerical coefficients and the ellipsis represent terms that become irrelevant in the asymptotic limit. The modulus $X$ controls the volume of the two-dimensional subspace, that can be either the fiber or the base depending on the asymptotic limit. Similarly, $Y$ controls the volume of the four-dimensional cycle. In practice, we take $Y^i \sim Y$ (i.e., a homogeneous scaling) for the set of moduli controlling the volumes of the two-cycles that give

the leading contribution to the volume of the 4d subspace in the asymptotic limit. Thus, in order to keep the volume fixed, we need

$$Y^2 \sim X^{-1}. \tag{B.16}$$

In this manner, (B.15) provides a convenient template for any asymptotic limit [176]. Therefore, taking $X \to 0$ models the decompactification limit to six dimensions, whereas $Y \to 0$ corresponds to the emergent string case.

We now turn to the higher-curvature correction of interest, namely the supersymmetry-protected $\mathcal{R}^2$ term,[42] and show that it agrees with the double EFT expansion in any of the asymptotic limits described above. Following [50] (see also [226]), the Wilson coefficient associated to this higher-curvature correction satisfies

$$\alpha_{\mathcal{R}^2} \sim c_{2,I} X^I, \tag{B.17}$$

where $c_{2,I}$ denote the integrated second Chern class numbers of the threefold. Notice that we are ignoring some numerical prefactors that are fixed in the asymptotic limit, including some dependence on the Calabi–Yau volume.

Let us first focus on the decompactification scenario. Since we have $X \sim Y^{-2} \to 0$, the leading term of the Wilson coefficient goes like $Y$. It was shown in [50] that this contribution indeed takes the form in (3) with $n = 4$ and $d = 5$. Being slightly more precise, one finds

$$Y \sim \Lambda_{\text{QG}}^{-2} \sim M_{\text{t}}^{-1/2}. \tag{B.18}$$

In the last step, we have taken into account that for a decompactification from five to six dimensions, the quantum gravity cutoff and the mass of the tower are related by $\Lambda_{\text{QG}} \sim M_{\text{t}}^{1/4}$ [99]. Apart from this, the Wilson coefficient exhibits another term going to zero linearly with $X$. This subleading correction then satisfies

$$X \sim Y^{-2} \sim M_{\text{t}}, \tag{B.19}$$

reproducing precisely the *field-theoretic* contribution (cf. eq. (2) with $n = 4$ and $d = 5$). Let us notice that this term would be absent only if $c_{2,X} = 0$, i.e., when the second Chern class of the compact space vanishes when integrated over the four-dimensional base. For instance, this happens when instead of a Calabi–Yau we have a toroidal compactification. In this case, we end up in a maximally supersymmetry setup, for which this four-derivative, higher-curvature correction is known to be forbidden by supersymmetry.

Turning to the weakly-coupled string limits, the leading contribution to the $\mathcal{R}^2$ operator becomes now linear in $X$. Again, as originally argued in [50], this leads to a term of the *quantum-gravitational* form (cf. (3) with $n = 4$ and $d = 5$). We thus have

$$X \sim \Lambda_{\text{QG}}^{-2} \sim M_{\text{t}}^{-2}, \tag{B.20}$$

where in the last step we have taken into account that $\Lambda_{\text{QG}} \sim M_{\text{t}}$ for this kind of infinite distance limits. As for the previous case, there is another contribution linear with $Y$. This term moreover satisfies

$$Y \sim X^{-1/2} \sim M_{\text{t}}, \tag{B.21}$$

thus recovering again the *field-theoretic* contribution. Let us point out that, more precisely, this term takes the form $c_{2,i} Y^i$ where $Y^i$ are the Kähler moduli controlling the volumes of the two-cycles that give the leading term for the volume of the four-dimensional fiber in the asymptotic

---

[42]For more details about the form of this higher-curvature operator and the connection to the 6d setup discussed in Section B.1 see, e.g., [50, Section 5].

limit. Hence, the field-theoretic contribution is absent if only if all of these integrated second Chern classes of the Calabi–Yau threefold are vanishing. On the other hand, the quantum-gravitational piece does vanish when the fiber is a $\mathbf{T}^4$, which corresponds to an emergent Type IIB limit. In [50], this was related to a supersymmetry enhancement along the asymptotic limit. Indeed, we know that this *quantum-gravitational* contribution must descend from the UV completed theory —i.e., 10d Type IIB string theory— for which this term in indeed absent due to supersymmetry. Finally, we note that in this case the leading contribution to the $\mathcal{R}^2$ Wilson coefficient is actually given by the field-theoretic expansion.

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
