# Peer review of "The Double EFT Expansion in Quantum Gravity"

_SciPost Physics, doi:SciPost Phys. 19, 096 (2025)_

## Round 1 · Referee Report · Anonymous (Referee 1) · 2025-5-22

Report
This manuscript studies the structure of higher-derivative expansions of effective field theories arising from a UV-complete theory of quantum gravity. They argue that there are two relevant scales in this expansion: (1) a field theory scale $M_t$, corresponding to the mass of the lightest (tower of) new degrees of freedom, and (2) a quantum-gravitational cutoff scale $\Lambda_{QG}$, beyond which there is no gravitational EFT description. The authors discuss this double EFT expansion from three perspectives: - Amplitudes: they study the scattering amplitudes that give rise to these gravitational Wilson coefficients, and explain how they fit within their double expansion. In particular, they point out that the higher-derivative expansion breaks down at the field theory scale $M_t$ rather than the quantum gravity scale $\Lambda_{QG}$. - Black holes: the authors discuss how black hole solutions see the UV cutoff scales, and in particular how the quantum-gravitational cutoff $\Lambda_{QG}$ rather than the field theory cutoff $M_t$ provides the dominant contribution to gravitational observables. - Bottom-up: they consider the gravitational Wilson coefficients from a bottom-up perspective, where they impose bounds they obtain from the emergent string conjecture, and compare these bounds with already established S-matrix bootstrap bounds.
I believe this manuscript is well-written and provides an important contribution to the field. It presents a helpful and structured way for organizing the gravitational EFT expansion in terms of the key cutoff scales that have recently received much interest in the swampland program. This double expansion makes clear at what scale and in what way the EFT breaks down, which had not yet been understood up to this point. They thoroughly analyze their double EFT expansion from the perspective of top-down amplitude calculations and black hole solutions. Moreover, they discuss bottom-up bounds they obtained from the emergent string conjecture, and connect in a meaningful way with complementary bounds obtained in the S-matrix bootstrap program.
I strongly recommend this manuscript for publication in SciPost. Below I have included some minor suggestions for improvement.
Requested changes
- I believe it could be helpful if the quantum gravitational cutoff scale/species scale $\Lambda_{\rm QG}$ is explained in somewhat more detail in the introduction. In particular, it would be helpful to comment that the emergent string conjecture predicts that it is either the string scale or a higher-dimensional Planck scale. I believe such clarifications would be especially helpful to bridge the gap with different fields such as S-matrix bootstrap.
- I was wondering if the authors could elaborate on the fact that the gravitational EFT expansion breaks down at the field theory scale $M_t$, while for charged black holes the quantum-gravitational cutoff scale $\Lambda_{QG}$ seems to be most relevant. In the outlook on page 56 they mention supersymmetry in this context, but this slightly confused me: the field-theoretic corrections are also present in the 4d $\mathcal{N}=2$ supergravity setting they considered, so I would appreciate a further clarification on this point. And in any case, I would also be interested to hear if the authors have more to say about why the black hole solutions see the larger cutoff scale of the two, regardless of supersymmetry.
Recommendation
Publish (surpasses expectations and criteria for this Journal; among top 10%)

Author: Alvaro Herraez on 2025-06-05 [id 5549]
(in reply to Report 1 on 2025-05-22)First of all, we would like to thank the referee for their careful report, positive feedback and for highlighting some improvements to be made in the draft. Let us now address the two points raised by the referee:
— We appreciate the suggestion from the referee to improve the clarity of the paper. To that end, we will add a short clarification on how, upon using the Emergent String Conjecture, the QG cutoff indeed recovers the higher dimensional Planck mass or the string scale. This will be included around eq. (1.7), which is when the Emergent String Conjecture is first introduced in the draft.
— Regarding the question of how black hole solutions may be able to detect scales besides the quantum gravity cutoff, we would like to remark the charged black holes that we study do see the breakdown of the lower-dimensional EFT at the KK scale. Indeed, the large radius expansion of their entropy index diverges precisely when the size of the BH is of the order of the KK length-scale. We tried to convey this message in Sections 1.2 and 3.2.2. Interestingly, after resumming the KK-suppressed corrections (or, equivalently, integrating in the extra dimension), the dominant quantum effect is indeed given by the QG scale when the BH radius is much smaller than that of the extra dimension. Concerning the comment we make in page 56, we would like to clarify that it was just an interesting observation beyond the scope of our work. Our goal was to point out that in setups with higher supersymmetry, some of the KK-suppressed corrections can be absent and thus the BH entropy index could be agnostic to this scale. Still, it is likely that other (less protected) observables of the BH are indeed sensitive to the latter.

---

## Round 1 · Referee Report · Anonymous (Referee 2) · 2025-6-18

Report
The analysis takes two complementary directions. In sections 2 and 3 the authors adopt a top-down approach, studying $D^{2\ell}R^4$ operators in type II strings via 4-graviton scattering amplitudes and black-hole entropy in N=2 compactifications. In section 4 the the point of view shifts to a bottom-up one, drawing on bootstrap and swampland arguments to explore constraints on the Wilson coefficients of the double EFT expansion.
While the first two sections are devoted primarily to testing the double expansion, section 4 explores its power in terms of bounding and connecting Wilson coefficients. A particularly interesting result is that imposing the Emergent String Conjecture, together with the scaling relations obtained in section 4.2, leads to a form of string universality.
The manuscript is well-written, though at times the presentation becomes a bit chaotic, owing to the range of diverse topics and viewpoints that the authors adapt. This can be confusing, but my impression is that the authors themselves acknowledged this issue and in fact provided a summary of the results in the introduction.
Given the broad relevance of the content of this work to multiple approaches to the exploration of gravitational EFTs, and given the potential for the double EFT expansion to foster cross-pollination between different subfields, I recommend this manuscript for publication in SciPost.
Recommendation
Publish (surpasses expectations and criteria for this Journal; among top 10%)

---

## Round 1 · Referee Report · Anonymous (Referee 3) · 2025-6-29

Report
In Section 2, the authors support their claims by analyzing four-graviton scattering amplitudes in maximal IIA supergravity in 10d, as well as in its toroidal compactifications. In Section 3, they test their proposal using BPS black hole solutions in 4d N=2 supergravity. In Section 4, they apply the double EFT expansion to derive bottom-up bounds on Wilson coefficients, and they compare these bounds with results obtained from the S-matrix bootstrap approach. I find this direction particularly interesting.
All in all, the manuscript is clearly written, although the presentation can feel lengthy and somewhat non-linear, particularly in Section 2. Despite this, I believe the paper addresses—and potentially answers— important questions in the recent literature on the swampland program. For this reason, I recommend it for publication.
One possible improvement would be to clarify the role of Section 3 within the overall narrative. At first glance, it appears somewhat detached from the main line of argument. Highlighting its purpose more explicitly and emphasizing which novel results it contains—distinct from the review material—, would strengthen the manuscript's coherence.
Recommendation
Publish (surpasses expectations and criteria for this Journal; among top 10%)

---

## Round 1 · Referee Report · Anonymous (Referee 4) · 2025-7-18

Report
After this perspective is convincingly argued for in Section 1, the bulk of the paper can be split into two parts: Sections 2 and 3 provide top-down evidence for the relevance of the double EFT expansion as an organizational principle and Section 4 uses the framework to extract conclusions from a bottom-up perspective instead. The top-down analysis in Section 2 successfully shows that higher-derivative corrections obtained from string theory do indeed align with the double EFT expansion, with the two distinguished types of terms leading to the failure of the EFT in appropriate limits and the remaining terms always providing subleading corrections. It would be interesting to understand how the story unfolds for less protected terms beyond dimensional analysis, but this seems difficult with current technology. Section 3 analyzes how higher-derivative terms affect 4D N=2 black hole gravitational observables under the light of the double EFT expansion. Switching gears, Section 4 uses the organizational principle within a bottom-up context in order to provide bounds for the Wilson coefficients of a gravitational EFT. The section draws connections to the bootstrap and Swampland literature, referencing Wilson coefficient bounds derived in the former and exploiting general features for the involved energy scales found in the latter. Section 5 concludes with a summary and outlook.
The manuscript is well-written and clear. The presentation in the main body of the paper could be made more concise, and the narrative between sections tightened. This makes the excellent summary found in Section 1.2 a welcome addition, highlighting the overarching direction of the paper and laying out the main points of each section for the reader.
The paper is of scientific interest and establishes connections between different areas, e.g., highlighting how the bootstrap and Swampland programs can inform each other. The double EFT expansion is a useful organizational principle, the implications of which would be interesting to understand further. The manuscript is fit for publication in SciPost, and I recommend it.
Recommendation
Publish (surpasses expectations and criteria for this Journal; among top 10%)

---

## Editorial Decision

published